# Empirical Study on Optimizer Selection for Out-of-Distribution Generalization

**Hiroki Naganuma**[1,2] **, Kartik Ahuja**[1,2]**, Shiro Takagi**[3]**, Tetsuya Motokawa**[4]**,**
*{naganuma.hiroki, kartik.ahuja}@mila.quebec, {takagi4646, moto.t.03.td}@gmail.com,*
[1] *Mila - Quebec AI Institute,* [2] *Université de Montréal,* [3] *Independent Researcher,* [4] *University of Tsukuba,*
**Rio Yokota**[5]**, Kohta Ishikawa**[6]**, Ikuro Sato**[5,6]**, Ioannis Mitliagkas**[1,2,7]
*rioyokota@gsic.titech.ac.jp, kishikawa@d-itlab.co.jp, isato@c.titech.ac.jp, ioannis@mila.quebec*
[5] *Tokyo Institute of Technology,* [6] *Denso IT Laboratory Inc.,* [7] *Canada CIFAR AI Chair*

**Reviewed on OpenReview:** *https://openreview.net/forum?id=ipeOIMglFF*

## Abstract

Modern deep learning systems do not generalize well when the test data distribution is slightly different to the training data distribution. While much promising work has been accomplished to address this fragility, a systematic study of the role of optimizers and their out-of-distribution generalization performance has not been undertaken. In this study, we examine the performance of popular first-order optimizers for different classes of distributional shift under empirical risk minimization and invariant risk minimization. We address this question for image and text classification using DomainBed, WILDS, and Backgrounds Challenge as testbeds for studying different types of shifts—namely correlation and diversity shift. We search over a wide range of hyperparameters and examine classification accuracy (in-distribution and out-of-distribution) for over 20,000 models. We arrive at the following findings, which we expect to be helpful for practitioners: i) adaptive optimizers (e.g., Adam) perform worse than non-adaptive optimizers (e.g., SGD, momentum SGD) on out-of-distribution performance. In particular, even though there is no significant difference in in-distribution performance, we show a measurable difference in out-of-distribution performance. ii) in-distribution performance and out-of-distribution performance exhibit three types of behavior depending on the dataset—linear returns, increasing returns, and diminishing returns. For example, in the training of natural language data using Adam, fine-tuning the performance of in-distribution performance does not significantly contribute to the out-of-distribution generalization performance.

## 1 Introduction

The choice of numerical optimization method can make a big difference when it comes to successfully training deep neural networks. In particular, the choice of optimizer influences training speed, stability, and generalization performance. Several studies have compared a variety of optimizers and investigated their influence on trainability and generalization (Wilson et al., 2017; Schneider et al., 2019; Choi et al., 2019). Some concluded that non-adaptive optimizers yield better generalization (Wilson et al., 2017; Balles & Hennig, 2018), while others countered that optimizer selection does not affect generalization performance (Schneider et al., 2019; Schmidt et al., 2021).

The conflicting nature of past reports can be explained by disparities in the methodology used for hyperparameter search. For Adam, in particular, hyperparameter $\epsilon$ controls the degree of adaptation. Low values, which are used by default, lead to a highly adaptive method. Unusually high values lead to less adaptation. In the limit of large $\epsilon$, Adam turns into a non-adaptive momentum method. In other words, when arbitrary tuning is allowed, methods like a Adam can be thought of as a superset of gradient descent with momentum.

The authors in Choi et al. (2019) take this approach, and also consider the less adaptive and non-adaptive regimes of Adam. Not surprisingly, they find that in this full generality Adam never performs worse than gradient descent with momentum in terms of in-distribution generalization, and in some cases, it can perform better.

Although these studies have made substantial progress to improve our understanding of optimizer characteristics, they are based on a common, foundational assumption in learning: training and test data are drawn from the same distribution. In applications, however, it is often the case that test data obey a distribution different from the one for training data. This *distributional shift* violates the typical assumption of independent and identically distributed (i.i.d.) data for learning (Nagarajan et al., 2021). Comparing the generalization performance of different optimizers under this distributional shift, known as out-of-distribution (OOD) generalization (Shen et al., 2021), is of great interest in theory and practice.

In our large suite of experiments, we focus on this OOD generalization question for Natural Language Processing (NLP) and image classification tasks. We train deep neural networks under the principle of Empirical Risk Minimization (ERM) or Invariant Risk Minimization (IRM) (Arjovsky et al., 2019) using a variety of optimizers. Because our objective is to investigate the impact of commonly used optimizers, we target five of the most popular optimizers that have been used and studied in recent years (Schmidt et al., 2021): stochastic gradient descent (SGD), Momentum SGD (Polyak, 1964), and Nesterov accelerated gradient (NAG, also known as Nesterov's momentum) (Nesterov, 2003) in the family of non-adaptive methods as well as RMSProp (Tieleman & Hinton, 2012) and Adam (Kingma & Ba, 2015) in the family of adaptive optimizers. We evaluate the OOD generalization performance of these optimizers on 10 different benchmarks: DomainBed (which includes seven image datasets) (Gulrajani & Lopez-Paz, 2021), the Backgrounds Challenge dataset (Xiao et al., 2021), , and CivilComments-WILDS (Koh et al., 2021).

As discussed above, methods like RMSProp and Adam can be tuned not to be adaptive, in which case they would subsume methods like gradient descent with momentum, and we would trivially get that the more general method can never underperform. Instead, we focus on the more nuanced question of adaptive vs non-adaptive methods: we tune RMSProp and Adam using a range of values for the hyperparameter $\epsilon$, which is strictly wider than ranges used in practice but still keeps the optimizers in their adaptive regime. In this context, we conduct an exhaustive hyperparameter search to select configurations that give good in-distribution validation accuracy for each optimizer. We then test the selected models on the above OOD test sets. Importantly, we demonstrate that our experiments explore more hyperparameter configurations than many existing benchmarks, as betrayed by the fact that we find better-performing models on said benchmarks.

In summary, our contributions are as follows:

- We design and perform a comparison of the effect of optimizers on OOD generalization on a number of OOD benchmarks. To the best of our knowledge, we are the first to consider such a wide variety and scale of datasets. Also, we conduct an empirical study using a wide range of hyperparameter configurations, examining over 20,000 models, evaluating their performance, and measuring the performance gap when moving from the in-distribution test set to the OOD test set.

- We demonstrate on a large number of image classification and NLP tasks that different optimizer choices lead to differences in OOD generalization performance. Furthermore, we show that adaptive optimizers yield more in-distribution overfitting and degrade OOD performance more than non-adaptive optimizers.

- We show that the observed correlation behaviors between in-distribution performance and OOD performance can be categorized into typical patterns: linear return, diminishing return, and increasing return [1]. This categorization should help practitioners better understand and select optimizers.

The following evidence supports the claim that non-adaptive methods outperform adaptive methods in OOD settings: i) There is no significant difference in the in-distribution generalization performance between adaptive

---

[1]These show how much performance can be expected in the out-of-distribution if we increase the in-distribution performance. These terms are explained in detail in Section 4.3.

and non-adaptive optimizers, ii) Adaptive optimizers perform worse in 10 out of 12 tasks regarding best out-of-distribution generalization performance, iii) We match models trained by adaptive optimizers to models trained by non-adaptive optimizers that yield the same in-distribution performance. Using this matching scheme for our comparison, models trained by non-adaptive methods achieve better OOD performance on 11 out of 12 tasks. These results are based on a comprehensive hyperparameter search and validated through soundness checks in the Appendix G.4.

Given the similarity of the results between ERM and IRM training principles, we have opted to focus our analysis on the former. Therefore, in the main section of this paper, we primarily report the results of ERM. Our observations that adaptive optimizers perform worse than non-adaptive methods in OOD performance align with theoretical results (Zou et al., 2022) previously reported in the literature, highlighting the drawbacks of adaptive optimizers such as Adam under the i.i.d. assumption and their tendency to fit noise in the data.

The paper is structured as follows. In Section 2, we discuss optimizer selection under the i.i.d. assumption and outline the problem of OOD generalization in the presence of distributional shifts, which can be encountered in real-world problems. In Section 3, we outline the optimizers and the OOD datasets that are the subject of this study, as well as our model selection method. In Section 4, we present the experimental protocol and the experimental results of 10 different datasets and 5 optimizers in each experimental setting.

## 2 Related Work

### 2.1 Optimizer Selection

Understanding the characteristics of the many optimizers proposed for deep neural network training (Schmidt et al., 2021) is of great importance to the machine learning research community. In terms of convergence, preconditioned optimizers, including Adam, are known to be superior to non-adaptive optimizers (Kingma & Ba, 2015; Liu et al., 2020; Amari, 1998).

While preconditioned optimization methods seem to be better than non-preconditioned ones in terms of convergence, Zhang et al. (2019) argued that there is a trade-off between generalization performance and convergence rate (Zhang et al., 2019). Wadia et al. (2020) also reported that preconditioned methods, especially second-order methods, do not provide great generalization performance either empirically or theoretically. With a simple theoretical and empirical analysis, Wilson et al. (2017) showed that adaptive optimizers are worse at generalization than simple SGD. Balles & Hennig (2018) also reported that Adam generalizes worse than Momentum SGD.

Contrary to these studies, Schneider et al. (2019) found that no single optimizer is the "best" in general. Similarly, Schmidt et al. (2021) claimed that optimizer performance varies from task to task. Choi et al. (2019) have come to a different conclusion from all the studies cited above. As described in Section 1, Choi et al. (2019) tuned the hyperparameter $\epsilon$ of adaptive methods that control the degree of adaptivity, which leads to non-adaptive methods. As a result, they found that well-tuned adaptive optimizers never underperform simple gradient methods.

These studies provided insights into how optimizer selection influences generalization. However, they focused on the classical supervised learning setting, in which the test distribution is assumed to be the same as the training distribution. Our research differs from theirs in that we investigate optimizer selection's influence on OOD generalization.

### 2.2 Out-of-Distribution Generalization

Taming the distributional shift is a big challenge in machine learning research (Sugiyama & Kawanabe, 2012; Ben-David et al., 2010; Pan & Yang, 2009; Szegedy et al., 2014; Arjovsky et al., 2019). Geirhos et al. (2020) argued that many modern deep neural network models sometimes learn shortcut features instead of intended features and overfit to a specific dataset.

Some studies have focused on generalization with adversarial noise to evaluate the impact of optimizer selection on OOD generalization. For instance, the theoretical and empirical analysis by Khoury (2019) showed that SGD is more robust against adversarial noise than adaptive optimizers. Wang et al. (2019) argued that adversarial examples to some methods are not necessarily adversarial to others. Abdelzad et al. (2020) found that the best optimizers for OOD detection vary by experimental setting. Metz et al. (2020) reported that a learned optimizer somehow unexpectedly outperformed a human-designed optimizer in terms of the OOD generalization.

These studies provide valuable insights on optimizer selection for the OOD problem. However, we emphasize that the previous work discussed above explores hyperparameters in a limited range. Khoury (2019) conducted the most exhaustive hyperparameter search but tuned only the learning rate for adaptive optimizers. As we briefly explain in Section 1, an exhaustive hyperparameter search is crucial for the empirical investigation of an optimizer's effect. Thus, we explored more hyperparameters than previous studies, including searching for over 20,000 models.

Here we emphasize that only shifts such as adversarial noise have been studied in previous studies of optimization selection for OOD generalization. Thus, we use a much more diverse set of real OOD datasets, including image classification and NLP tasks where the distributional shift is significant, covariate shift, correlation shift, subpopulation shift, and background shift, not only domain generalization. The set of datasets we explored is the most exhaustive for evaluating the optimizer's role in OOD generalization, as far as we know.

Finally, we mention to some relevant works Kumar et al. (2022a); Wortsman et al. (2022); Chen et al. (2023); Kumar et al. (2022b). Kumar et al. (2022a); Wortsman et al. (2022); Kumar et al. (2022b) have studied the intricate balance between ID and OOD performance when fine-tuning pre-trained models. Chen et al. (2023) theoretically showed that optimizing the ERM with the relaxed OOD penalty is not likely to have a good performance and proposed a better practical solution.

## 3 Preliminaries

### 3.1 Optimizers Subjected to Our Analysis

Similar to previous studies (Wilson et al., 2017; Schneider et al., 2019; Choi et al., 2019), we compare two types of optimizers. The first one is non-adaptive optimizers. The update equation at iteration $t$ of model parameter $\boldsymbol{\theta}_t$ is as follows:

$$\boldsymbol{v}_t \leftarrow \gamma \boldsymbol{v}_{t-1} + \eta_t \tilde{\nabla}_{\boldsymbol{\theta}_{t-1}} \ell(\boldsymbol{\theta}_{t-1}), \quad \boldsymbol{\theta}_t \leftarrow \boldsymbol{\theta}_{t-1} - \boldsymbol{v}_t \tag{1}$$

where $\eta_t$ is the learning rate, $\ell(\boldsymbol{\theta})$ is the loss, and $\tilde{\nabla}_{\boldsymbol{\theta}_{t-1}}$ is the stochastic gradient, in the particular case of stochastic gradient descent, $\gamma = 0$. Optimizers with momentum terms such as Momentum SGD (Polyak, 1964), and Nesterov momentum (Nesterov, 2003) are also classified as non-adaptive optimizers, and $\gamma$ is the parameter for controlling the momentum term. For Nesterov momentum, $\ell(\boldsymbol{\theta}_{t-1})$ should be replaced $\ell(\boldsymbol{\theta}_{t-1} - \gamma \boldsymbol{v}_{t-1})$.

The second type of optimizer is adaptive methods. Adam and RMSprop are adaptive optimizers and they can be written in the form of the generic adaptive optimization method. The generic adaptive optimization method can be written as in Algorithm 1. This is based on what (Liu et al., 2020) and (Reddi et al., 2018) propose.

Our selection of optimizers aligns with prior research on optimizer comparison (Choi et al., 2019) and recent studies on out-of-distribution (OOD) tasks, which have predominantly focused on Adam rather than alternatives such as AdamW. Therefore, we concentrate on the five most commonly employed optimizers in practice (Schmidt et al., 2021). Given their widespread usage, we chose to focus on studying Adam's performance under the adaptive regime, as this would provide more pertinent findings for present-day practices. Experimental results for AdamW (Loshchilov & Hutter, 2017) and SAM (Foret et al., 2020), although not for all data sets, are also discussed in Appendix I.3.

---

**Algorithm 1** Generic adaptive optimization method setup.

---

**Require:** $\{\eta_t\}_{t=1}^T$: step size, $\{\phi_t, \psi_t\}_{t=1}^T$ function to calculate momentum and adaptive rate, $\boldsymbol{\theta}_0$: initial parameter, $\ell(\boldsymbol{\theta})$: objective function
1: **for** $t = 1$ to $T$ **do**
2: $\quad \boldsymbol{g}_t \leftarrow \tilde{\nabla}_{\boldsymbol{\theta}} f_t(\boldsymbol{\theta}_{t-1})$ (Calculate stochastic gradients w.r.t. objective at timestep t)
3: $\quad \boldsymbol{w}_t \leftarrow \phi_t(\boldsymbol{g}_1, ..., \boldsymbol{g}_t)$ (Calculate momentum)
4: $\quad \boldsymbol{l}_t \leftarrow \psi_t(\boldsymbol{g}_1, ..., \boldsymbol{g}_t)$ (Calculate adaptive learning rate)
5: $\quad \boldsymbol{\theta}_t \leftarrow \boldsymbol{\theta}_{t-1} - \eta_t \boldsymbol{w}_t \boldsymbol{l}_t$ (Update parameters)
6: **end for**

---

### 3.2 Out-of-Distribution Generalization Datasets

We use the following datasets to evaluate the optimizer influence on OOD generalization: DomainBed (Gulrajani & Lopez-Paz, 2021), Backgrounds Challenge (Xiao et al., 2021), (Koh et al., 2021), and CivilComments-WILDS (Koh et al., 2021). These datasets include both artificially created and real-world data, and the applications include image classification and NLP tasks. Although we describe the details of these datasets in Appendix C, we summarize them below.

**Image Classification Datasets:** DomainBed consists of a set of benchmark datasets for domain generalization, which includes PACS (Fang et al., 2013), VLCS (Li et al., 2017), Office-Home (Venkateswara et al., 2017), Terra Incognita (Beery et al., 2018) DomainNet (Peng et al., 2019), Rotated MNIST (Ghifary et al., 2015), and Colored MNIST (Arjovsky et al., 2019). These datasets contain a variety of distributional shifts. VLCS is a set of different image datasets, for example, images of *birds* from several datasets. Terra Incognita is a dataset consisting of images taken by cameras at different locations. The difference in the location of the camera corresponds to the difference in the domain. PACS, Office-Home, and DomainNet are image datasets whose style varies by domain. For example, an image of PACS in one domain is photography, while that in another domain is a sketch. Rotated MNIST is an artificially generated dataset of domains that have been given different rotation angles.

Colored MNIST is an *anomaly* in the DomainBed dataset. $P(Y|X)$ remains the same for all datasets (covariate shift) except in Colored MNIST. This dataset is designed to make models fail by exploiting spurious correlations in training environments. In particular, the dataset is such that the model can only exploit the source of spurious correlation (color) and achieve a very high training accuracy without relying on the true invariant source of correlation (shape). In contrast, none of the other datasets have such strong spuriousness. The strength of the spurious correlation flips in the test domain and thus it induces a negative correlation between the validation and the test accuracy.

The Backgrounds Challenge dataset measures a model's robustness against background shift (Xiao et al., 2021). A model is trained on an image and evaluated on the same image with a different background. If the model exploits the background features during training, it will be fooled during evaluation. Therefore, if the model strongly depends on the background, this dataset is a difficult OOD dataset, while if the model does not, this dataset is easy to model. To further strengthen our claim, we also performed experiments on CIFAR10-C and CIFAR10-P which can be casted to image corruption and perturbation shift. The results are shown in Appendix I.1.

**Natural Language Processing (NLP) Datasets:** The CivilComments-WILDS dataset is cast as a subpopulation shift problem. The shift problem tackles a binary classification problem that predicts the toxicity of comments on articles, and domain vectors are assigned according to whether the comment refers to one of eight demographic identities. In the subpopulation shift, the test distribution is a subpopulation of the training distribution.

The dataset has the characteristics of a hybrid shift of a subpopulation shift and domain shift. This dataset is cast as the problem of estimating a rating of 1–5 from the rating comments of each user. In this kind

of dataset, each user corresponds to a domain, and the goal is to produce a high performance for all user comments.

### 3.3 Model Selection Method and Evaluation Metrics

We leverage two distinct model selection methodologies. Our first approach involves partitioning the data from the training domains into a training dataset and a validation dataset, subsequently selecting the model that demonstrates the highest average performance (accuracy) based on the validation data from the training domain. This approach, referred to as **training-domain validation set model selection strategy**, follows the research methodology established by Gulrajani & Lopez-Paz (2021). The second method employs an **Oracle-based model selection strategy** using test domain data. The following shows the model selection methodology when using validation in the training domain.

In the training phase of DomainBed datasets, we do not access the data in the test domain but split data from the training domains into a training dataset and validation dataset. We choose the model with the highest average performance (accuracy) on the validation data in the training domain. As a metric for evaluation, we evaluated the generalization performance in the test domain as the OOD accuracy.

The Backgrounds Challenge uses Imagenet-1k as the training data set and selects models based on their accuracy on the validation data in the training domain. After that, we measure the in-distribution performance with IN9L (Xiao et al., 2021), which aggregates the test data into nine classes. As for the OOD performance, we measure the classification performance on the data where the background image of IN9L is replaced with the background image of other images.

In CivilComments-WILDS, we divide the data into training, validation, and test datasets and maximize worst-group accuracy in the validation data (and by association, maximize the average accuracy over all domains). Then, we perform model selection and evaluate the OOD accuracy on the test data. We adopt the same hyperparameter selection strategy for datasets as that for CivilComments-WILDS. As a metric, we do not evaluate the worst-group performance but rather the 10th-percentile accuracy for the performance of each domain by following the standard federated learning literature (Caldas et al., 2018).

## 4 Experiments

### 4.1 Experimental Overview

Our study aims to elucidate the influence of optimizer selection under distributional shifts. To that end, we perform image classification and NLP tasks and evaluate the trained model accuracy on the benchmark datasets introduced in Section 3.2. By comparing the test accuracy for in-distribution samples with that for OOD samples, we can observe how the solution found by each optimizer is robust to the distributional shift. We investigate both ERM and IRMv1 (Arjovsky et al., 2019), a problem set to solve IRM, to clarify the relationship between the problem formulation and optimizer selection in the OOD problem. For VREx (Krueger et al., 2021) and CORAL (Sun & Saenko, 2016), small-scale experimental results are also provided in Appendix I.7.

We compare five optimizers for all datasets except for the Backgrounds Challenge dataset. For the Backgrounds Challenge dataset, we compare only Momentum SGD and Adam due to the computational cost. We describe the configurations of hyperparameters and protocol for the experiments in further detail in Appendix E and Appendix D respectively. The remaining experimental settings of environment are explained in Appendix B.

**Hyperparameter Tuning:** The hyperparameters are tuned using Bayes optimization functionality of Weights&Biases[2] by evaluating in-distribution validation accuracy. Bayesian optimization sequentially explored the potential hyperparameter candidate points, and we evaluated all the trained models in the search process for comparison. As a confirmation of the soundness of our hyperparameter search, Appendix H.1 shows the histogram that the explored hyperparameters are drawn from the reasonably wide range. In addition, we investigated the relationship between the number of trials to explore hyperparameters and the

---

[2]https://wandb.ai/site

Table 1: Comparison of the best OOD accuracy (%) of ERM between five optimizers. We use **Oracle** (see definition at 3.1 in Gulrajani & Lopez-Paz (2021)) as model selection method in this table. The model selection results in the training-domain validation set are shown in Table 13. Except for a small set of problems, non-adaptive optimizer outperforms adaptive optimizer in all but 10 out of 12 tasks. As a soundness check, we confirm that our Adam results outperform all existing benchmark results using Adam. Details are given in Appendix G.4.

| Model | OOD Dataset | Non-Adaptive Optimizer | | | Adaptive Optimizer | |
|---|---|---|---|---|---|---|
| | | SGD | Momentum | Netsterov | RMSProp | Adam |
| 4-Layer CNN | ColoredMNIST | 34.01% | 34.23% | 40.56% | **89.30%** | 73.92% |
| | RotatedMNIST | 90.00% | 95.41% | 94.06% | 96.27% | **96.40%** |
| ResNet50 | VLCS | 99.43% | **99.43%** | 99.29% | 99.29% | 99.29% |
| | PACS | 88.67% | **89.55%** | 89.25% | 88.81% | 89.30% |
| | OfficeHome | 64.64% | **65.01%** | 63.82% | 62.91% | 63.82% |
| | TerraIncognita | **63.21%** | 62.41% | 62.85% | 62.31% | 61.35% |
| | DomainNet | 58.38% | 61.91% | **62.24%** | 55.74% | 58.48% |
| | BackgroundChallenge | - | 80.09% | - | - | 77.90% |
| DistilBERT | Amazon-WILDS | 52.00% | **54.66%** | 54.66% | 53.33% | 52.00% |
| | CivilComment-WILDS | 51.66% | 57.69% | **60.07%** | 45.39% | 46.82% |
| ResNet-20 | ColoredMNIST | - | **33.50%** | - | - | 31.47% |
| ViT | PACS | - | **91.80%** | - | - | 91.26% |

best OOD performance and show results in Appendix G.2. The impact of initialization strategies during hyperparameter optimization is also confirmed in Appendix H.3. The data shown in the box plot (Figure 2, 3, and 4) are from the evaluation of several trained models.

The number of epochs (the steps budget) used is in line with previous studies (Gulrajani & Lopez-Paz, 2021; Koh et al., 2021; Choi et al., 2019) to ensure the soundness of our experimental design. Since we use the fixed epoch, it might seem to be unfair than when tuning the epoch as well for the optimizer that converges faster. Thus, we studied the effect of early stopping, which corresponds to the tuned epoch in Appendix G.4, and the result confirms that employing a fixed epoch does not impair the fairness of our comparison experiments. We discuss the details in Appendix G.4.

**Boxplot:** We believe that sharing the whole distribution of tuning outcomes is important because: it gives an idea of how sensitive methods are to tuning and how much tuning effort is required. We also share the raw data as scatter plots (Figure 15, 16, and 16) in Appendix F.4.

## 4.2 Experimental Results

For training-domain validation set model selection, Figure 1 shows the relationship between the in-distribution and OOD accuracy in the ERM setting. The x-axis of the plot is the in-distribution accuracy, and the y-axis is the OOD accuracy. To clarify the trend, the in-distribution accuracy corresponding to the x-axis is divided into ten bins, and the average performance of the OOD accuracy in each bin is shown on the y-axis. We compared Momentum SGD, the best non-adaptive optimizer, and Adam, the best adaptive optimizer. Our findings reveal that, in our field of study, where high in-distribution performance is achieved, Momentum SGD outperforms Adam on 11 out of 12 tasks (Table 13).

With respect to the Oracle model selection strategy, best OOD performance, non-adaptive optimizers surpassed adaptive optimizer methods on 10 out of 12 tasks (Table 1). These results suggest that non-adaptive optimizers are more advantageous than adaptive optimizers in OOD, despite their similar performance in the ID environment. For a more detailed explanation of the experimental results for each dataset, please refer to the following section.

**DomainBed:** Figure 2 shows a box plot of the difference between the average in-distribution and OOD accuracy for ERM. In the following discussion, we call this difference a *gap* for convenience. Due to the limitation of the paper length, only the plots of PACS and Office-Home are shown here. All results, including IRM results, are shown in Appendix F.2.1.

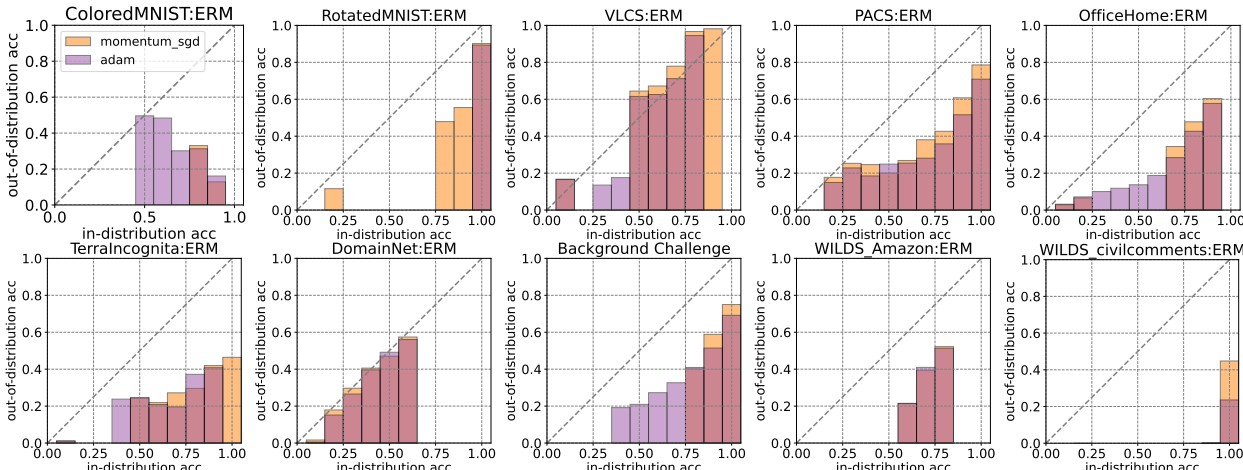

Figure 1. Comparison of the in-distribution accuracy and the OOD accuracy of ERM between Momentum SGD and Adam. We also show the training results in the exhaustive hyperparameter search range as a scatter plot in Appendix F.3. In these ten plots, for each dataset, the in-distribution performance is separated by every ten bins 0.1. The average OOD performance when evaluating the checkpoints in that bin is shown on the vertical axis. We compare which optimizer shows better OOD performance for each model that achieves equivalent in-distribution performance. This approach can be characterized as a model selection strategy, which follows the methodology of utilizing a **training-domain validation set**. In most cases, momentum SGD outperforms Adam in OOD performance in the rightmost region of our interest (the region where high in-distribution performance is achieved).

We found two distinct optimizer effect patterns, depending on the dataset: i) PACS, Office-Home, VLCS, Terra Incognita, Rotated MNIST, and DomainNet, and ii) Colored MNIST. Because these patterns appear in both the results of ERM and IRM, we focus on ERM for the explanation.

For PACS, Office Home, VLCS, TerraIncognita, RotatedMNIST, and DomainNet, the OOD accuracy is greater for non-adaptive optimizers. The gap between the mean OOD accuracy and the mean in-distribution accuracy was smaller for the non-adaptive optimizer except for TerraIncognita. This means the models trained with the non-adaptive optimizer are more robust on average. In TerraIncognita, the non-adaptive optimizer significantly outperforms the adaptive optimizer on the average in-distribution performance and OOD performance. We note, however, that except in this problem, the adaptive optimizer achieves a smaller gap between the two. As shown in Figure 1, when comparing models with the same in-distribution performance, the non-adaptive optimizer showed higher OOD accuracy than the adaptive optimizer. The results in Figures 1 and 2 confirm that the non-adaptive method achieves higher OOD accuracy than the adaptive method, both on average and for models with the same in-distribution performance.

Colored MNIST shows the opposite pattern of these results, where the adaptive optimizer is better than the non-adaptive optimizers. As outlined in Section 3.2, Colored MNIST differs from the other datasets. To comprehend why adaptive optimizers are more effective in OOD generalization for this dataset, we have plotted the average training, validation, and test accuracy over time during training, as illustrated in Figure 33. Our explanation for this outcome can be found in Appendix G.3.

We note our considerations regarding the exceptional behavior of ColoredMNIST. ColoredMNIST is a binary classification task dataset with random labels, where spurious features are positively correlated with invariant features in in-distribution and negatively correlated with OOD. Non-adaptive optimizers learn the spurious features, achieve the oracle performance for in-distribution and perform worse than a random guess for OOD due to inverted correlations. Adaptive optimizers, in contrast, seem to overfit the training data, achieving 100% accuracy in the training set (more than oracle [3]) in this dataset, and failing to learn the spurious

---

[3]Since ColoredMNIST includes label flip as noise, even the best model that correctly learns the data rules will only perform 85%.

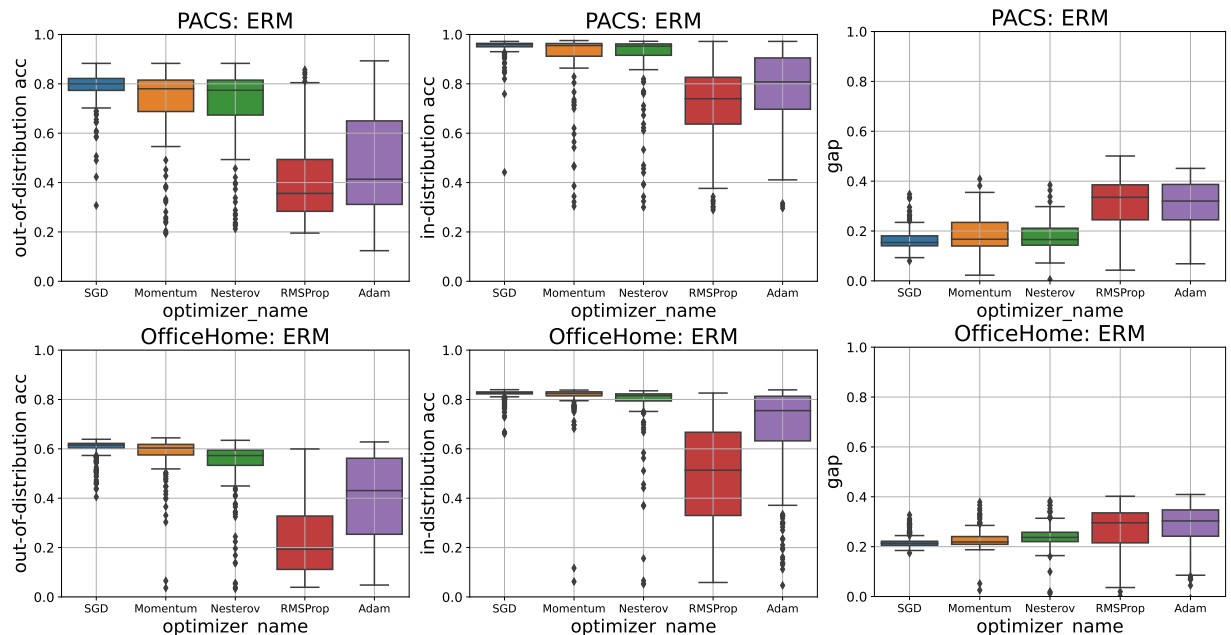

Figure 2. PACS and OfficeHome in DomainBed: Comparison of the in-distribution (validation) accuracy and the out-of-distribution (test) accuracy of ERM across five optimizers. Non-adaptive optimizers outperform the adaptive optimizers in terms of OOD generalization, and the gap between in-distribution performance and OOD performance is small. The details and results of other dataset experiments are described in Appendix F.2.

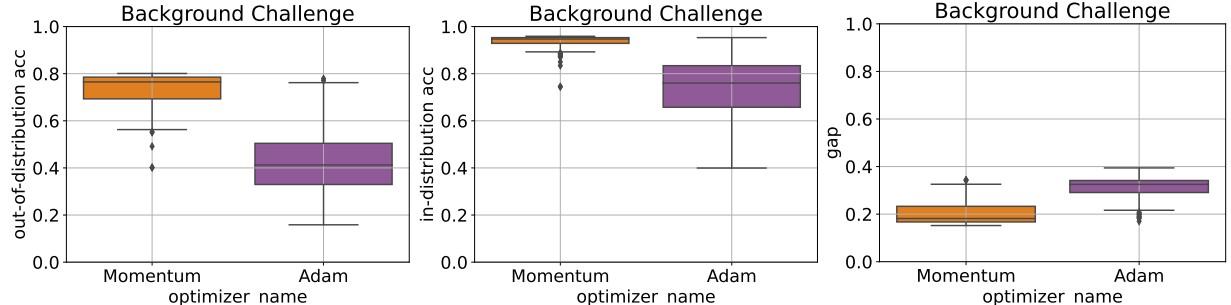

Figure 3. Backgrounds Challenge: Comparison of the in-distribution (validation) accuracy and the out-of-distribution (test) accuracy of ERM across two optimizers. In terms of OOD generalization, Momentum SGD outperforms Adam in both average and best OOD performance. The highest value of in-distribution accuracy remains the same, but Momentum SGD shows higher performance on average.

features. Due the aforementioned (synthetic) inverted correlations in the dataset, this overfitting behaviour in-distribution happens to favor adaptive optimizers. This behavior happens exceptionally on ColoredMNIST due to these synthetic flips in correlation. This exploitation unexpectedly enables Adam to avert being trapped in the training domain and produces better OOD generalization.

**Backgrounds Challenge:** Backgrounds Challenge requires training of ImageNet-1k on ResNet50 from scratch, and it takes 256 GPU hours to obtain a single trained model, so we only compared Momentum SGD with Adam. Because Momentum SGD achieved competitive performance among non-adaptive optimization methods in the DomainBed and WILDS experiments, we adopted Momentum SGD to represent non-adaptive optimization methods. In the same way, Adam outperformed RMSProp in almost all the benchmarks, so we adopted Adam as a representative of the adaptive optimizers.

Figure 3 compares the accuracy for ORIGINAL (in-distribution) and Mixed-Rand (out-of-distribution). The best in-distribution performance is the same for each optimizer, but concerning the best OOD performance,

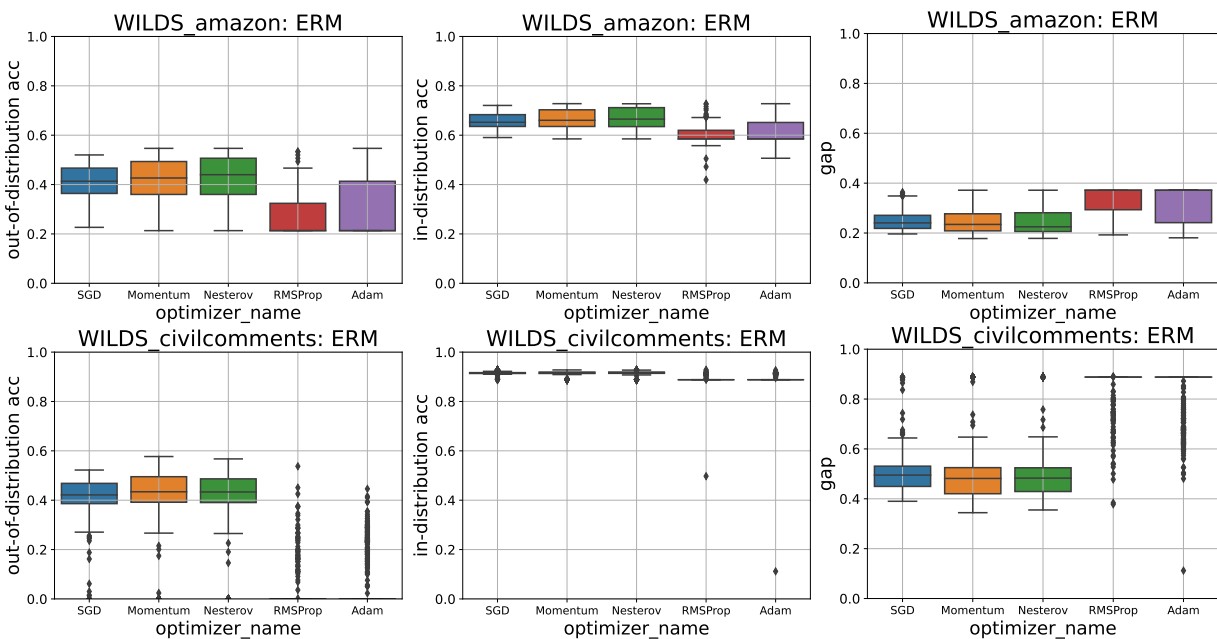

Figure 4. and CivilComments-WILDS: Comparison of the in-distribution (validation) accuracy and the out-of-distribution (test) accuracy of ERM across five optimizers. Both non-adaptive and adaptive optimizers have similar in-distribution accuracy, but the adaptive method significantly degrades the performance of the OOD generalization.

the non-adaptive optimizer outperformed the adaptive optimizer. As can be seen from Figure 1 (second row, middle column), particularly in the region of high in-distribution performance on the right, the OOD performance of Momentum SGD exceeds that of Adam.

**WILDS:** A comparison of in-distribution and OOD averages for WILDS is shown in Figure 4. It can be clearly seen that the adaptive optimizer is fit too well to the in-distribution in the WILDS problem setting. For and CivilComment-WILDS, as in DomainBed and Backgrounds Challenge, the non-adaptive optimizer outperforms the adaptive optimizer in terms of OOD generalization. The gap between in-distribution accuracy and OOD accuracy is also tiny for non-adaptive optimizers. In particular, the CivilComments-WILDS experimental result is remarkable, as both non-adaptive and adaptive optimizers show similar high in-distribution performance, but in the OOD environment, adaptive methods significantly fail to make inferences.

### 4.3 Correlation Behaviors

Our results show that three typical types of behavior are observed in terms of the correlation between in-distribution performance and OOD performance for different datasets. Detailed results of the experiments on all datasets are shown in Appendix F.4. We follow Miller et al. (2021), who used a probit transform to show the relationship. The three types are increasing return, linear return, and diminishing return. These show how much performance in OOD can be expected if we increase the in-distribution performance.

The increasing return is an example, as shown in the leftmost part of Figure 5. The increasing returns in large regions of the in-distribution generalization significantly affect the OOD generalization. This suggests that the last tuning is significant for the OOD generalization, as seen in all domain generalization datasets except for DomainNet.

The linear return is as shown in the middle of Figure 5. The OOD accuracy increases linearly with the in-distribution accuracy. This is generally the same result for in-distribution validation and test.

Conversely, diminishing return behavior, illustrated in the rightmost part of Figure 5, indicates that substantial in-distribution improvement leads to a saturation of OOD generalization with only a marginal effect. This

observation implies that the effort invested in fine-tuning might not always yield significant enhancements in OOD generalization. We have observed similar trends in settings with subpopulation shifts, such as CivilComments-WILDS.

In Appendix F.4, we present our experimental results without a probit transform, confirming that diminishing returns are not necessarily linear before probit transformation. This finding aligns with recent studies by Wenzel et al. (2022); Teney et al. (2022), stating that the accuracy of ID and OOD varies across datasets. Moreover, as Baek et al. (2022) suggests, the occurrence of linear return depends on the problem set. Our results support these claims, providing a comprehensive analysis of datasets and problem sets that Miller et al. (2021) did not address and uncovering trends they did not reveal.

For practitioners, our findings offer valuable insights into adjusting their expectations and strategies based on the observed behavior. For instance, if they work with a dataset similar to CivilComments-WILDS, they can anticipate one of the identified types of behavior. Should their dataset exhibit saturating behavior, they can adjust their expectations accordingly; conversely, if their dataset demonstrates a regime where every slight improvement in in-distribution accuracy aids, they should pursue enhanced in-distribution optimization.

## 5 Discussion and Conclusion

We conduct an exhaustive empirical comparison of the generalization performance of various optimizers under different practical distributional shifts. Notably, ten state-of-the-art OOD datasets were used to study the environment of broad shifts in correlation and diversity shifts. The investigation elucidates how optimizer selection affects OOD generalization. As the main claim, the answer to our research objective is that the non-adaptive optimizer is superior to the adaptive optimizer in terms of OOD generalization.

The following evidence supports this: i) when comparing in-distribution accuracy is the same, OOD accuracy of non-adaptive optimizers is greater than that of adaptive optimizers (Figure 1); ii) on average and top performance, the OOD accuracy of the non-adaptive optimizer is higher (Figures 2, 3, and 4). All these points support our main claim that non-adaptive optimizers are superior in OOD generalization within our exhaustive experiments. We have tuned Adam to the range that it is used in practice and have updated Adam's scores on all OOD datasets, which use Adam optimizer as default for benchmarks. This supports the soundness of our experiments.

Overall, we can conclude that optimizer selection influences OOD generalization in the cases we are interested in. Future research should consider the algorithm or loss function and optimizers in the OOD problem. The results of IRM show a trend similar to ERM's, but a more detailed analysis is needed to consider the differences in loss landscapes. All these points support our main claim that non-adaptive optimizers are superior in OOD generalization within our exhaustive experiments.

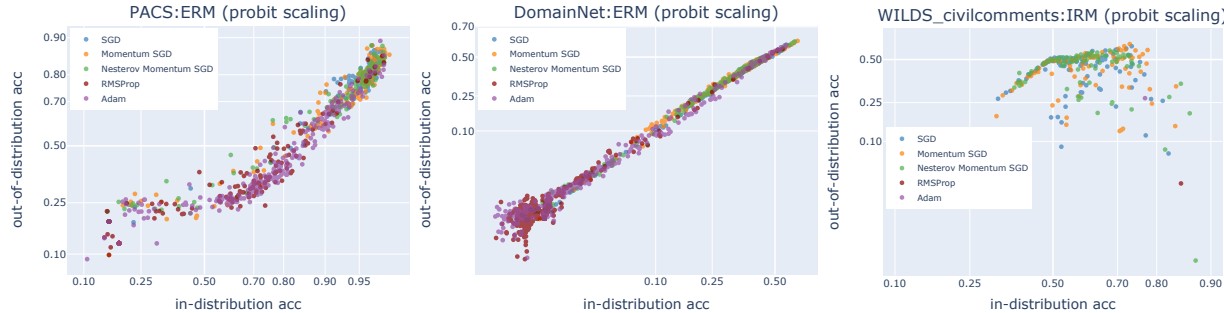

Figure 5. Three types of correlation behavior: increasing return (PACS), linear return (DomainNet), and diminishing return (CivilComments-WILDS). The legend circles on the right side of each figure show, in order, the SGD, Momentum SGD, Nesterov Momentum, RMProp, and Adam. The details and results of other dataset experiments are described in Appendixes 15, 16, and 16
.

Finally, we would like to mention the limitations of our work. One limitation is that we did not study recently proposed and less popular optimizers.

The choice of optimizers we study is in line with previous work (Choi et al., 2019), (Schmidt et al., 2021) on optimizer comparison and with most recent OOD work; those studies overwhelmingly focused on Adam rather than e.g., AdamW (Loshchilov & Hutter, 2017). Similarly, other less popular optimizers such as SWA (Izmailov et al., 2018), SWAD (Cha et al., 2021), SAM (Foret et al., 2020), have been omitted to allow for a more extensive study of the chosen methods [4].

Another limitation is that we employed a total of six models used in the DomainBed (ConvNet, ResNet20, ResNet50 and Vision Transformer (Dosovitskiy et al., 2020)), Backgrounds Challenge (ResNet50), and WILDS (DistilBERT (Sanh et al., 2019)) benchmarks.

## Acknowledgments

Our deepest gratitude goes out to the anonymous reviewers and the Action Editor whose invaluable insights substantially enhanced the quality of this manuscript. We express our heartfelt thanks to Kilian Fatras (Mila, McGill), Divyat Mahajan (Mila, UdeM), and Mehrnaz Mofakhami (Mila, UdeM) for their constructive feedback that significantly contributed to the research process. Our sincere appreciation extends to the Masason Foundation Fellowship, which generously supported this work and made an award to Hiroki Naganuma.

The computational resources instrumental to this study were provided under the auspices of the "ABCI Grand Challenge" Program, National Institute of Advanced Industrial Science and Technology (AIST), and the TSUBAME Grand Challenge Program, Tokyo Institute of Technology. Special thanks to the AI Bridging Cloud Infrastructure (ABCI) and the TSUBAME 3.0. Moreover, we acknowledge the generous allocation of computational resources from the TSUBAME3.0 supercomputer, facilitated by Tokyo Institute of Technology. This assistance came through the Exploratory Joint Research Project Support Program from JHPCN (EX23401) and the TSUBAME Encouragement Program for Young / Female Users, whose support was instrumental to this research.

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

# Contents (Appendix)

# A  Optimizers

As we explained in Section 4.1, we compared SGD, momentum-SGD, Nesterov momentum, RMSprop, and Adam. We write the algorithm of these optimizers below.

## A.1  SGD

$$\boldsymbol{\theta}_t \leftarrow \boldsymbol{\theta}_{t-1} - \eta_t \tilde{\nabla}_{\boldsymbol{\theta}_{t-1}} \ell\left(\boldsymbol{\theta}_{t-1}\right) \tag{2}$$

## A.2  Momentum-SGD

$$\boldsymbol{v}_t \leftarrow \gamma \boldsymbol{v}_{t-1} + \eta_t \tilde{\nabla}_{\boldsymbol{\theta}_{t-1}} \ell\left(\boldsymbol{\theta}_{t-1}\right) \tag{3}$$
$$\boldsymbol{\theta}_t \leftarrow \boldsymbol{\theta}_{t-1} - \boldsymbol{v}_t \tag{4}$$

## A.3  Nesterov Momentum

$$\boldsymbol{v}_t \leftarrow \gamma \boldsymbol{v}_{t-1} + \eta_t \tilde{\nabla}_{\boldsymbol{\theta}_{t-1}} \ell\left(\boldsymbol{\theta}_{t-1} - \gamma \boldsymbol{v}_{t-1}\right) \tag{5}$$
$$\boldsymbol{\theta}_t \leftarrow \boldsymbol{\theta}_{t-1} - \boldsymbol{v}_t \tag{6}$$

## A.4  RMSprop

$$\boldsymbol{v}_t \leftarrow \alpha \boldsymbol{v}_{t-1} + (1-\alpha) \tilde{\nabla}_{\boldsymbol{\theta}_{t-1}} \ell\left(\boldsymbol{\theta}_t\right)^2 \tag{7}$$
$$\boldsymbol{m}_t \leftarrow \gamma \boldsymbol{m}_{t-1} + \frac{\eta_t}{\sqrt{\boldsymbol{v}_t + \epsilon}} \tilde{\nabla}_{\boldsymbol{\theta}_{t-1}} \ell\left(\boldsymbol{\theta}_t\right) \tag{8}$$
$$\boldsymbol{\theta}_t \leftarrow \boldsymbol{\theta}_{t-1} - \boldsymbol{m}_t \tag{9}$$

## A.5  Adam

$$\boldsymbol{m}_t \leftarrow \beta_1 \boldsymbol{m}_{t-1} + (1-\beta_1) \tilde{\nabla}_{\boldsymbol{\theta}_{t-1}} \ell\left(\boldsymbol{\theta}_t\right) \tag{10}$$
$$\boldsymbol{v}_t \leftarrow \beta_2 \boldsymbol{v}_{t-1} + (1-\beta_2) \tilde{\nabla}_{\boldsymbol{\theta}_{t-1}} \ell\left(\boldsymbol{\theta}_t\right)^2 \tag{11}$$
$$b_t \leftarrow \frac{\sqrt{1-\beta_2^t}}{1-\beta_1^t} \tag{12}$$
$$\boldsymbol{\theta}_t \leftarrow \boldsymbol{\theta}_{t-1} - \eta_t \frac{\boldsymbol{m}_t}{\sqrt{\boldsymbol{v}_t + \epsilon}} b_t. \tag{13}$$

# B  Implementation and Environment for Experiments

We perform our experiment with ABCI (AI Bridging Cloud Infrastructure), a supercomputer owned by the National Institute of Advanced Industrial Science and Technology, and TSUBAME, a supercomputer owned by the Tokyo Institute of Technology.

All codes for experiments are modifications of the codes provided by the authors who introduced the datasets Gulrajani & Lopez-Paz (2021); Koh et al. (2021); Xiao et al. (2021). Licenses of the codes are MIT license for DomainBed Gulrajani & Lopez-Paz (2021) and WILDS Koh et al. (2021). The code of Backgrounds Challenge does not indicate the license. Our code can be found at the link below.
https://github.com/Hiroki11x/Optimizer_Comparison_OOD

# C  Datasets

## C.1  DomainBed

DomainBed consists of sub-datasets shown in Table 2, where we exclude Terra Incognita and Rotated MNIST as stated in Section 3.2. We summarize the dataset information in the Table by referring to (Gulrajani & Lopez-Paz, 2021).

Table 2: DomainBed: Dataset Information

|  | domain | examples | class |
|---|---|---|---|
| Colored MNIST | [0.1, 0.2, 0.9] | 70000 | 2 |
| Rotated MNIST | [0, 15, 30, 45, 60, 75] | 70000 | 10 |
| PACS | [art, cartoons, photos, sketches] | 9991 | 7 |
| VLCS | [Caltech101, LabelMe,415SUN09, VOC2007] | 10729 | 5 |
| Office-Home | [art, clipart, product, real] | 15588 | 65 |
| TerraIncognita | [L100, L38, L43, L46] | 24788 | 10 |
| DomainNet | [clipart, infograph, painting, quickdraw,420real, sketch] | 586575 | 345 |

Colored MNIST is a dataset for binary classification of MNIST dataset (Arjovsky et al., 2019). The digits from 0 to 4 are labeled 0, and those greater than 5 are labeled 1, and the task is to classify these classes successfully. However, each digit is also colored by either red or green. This is for having models to confuse the important feature for classification. The domain $d$ indicates the correlation of the label with color. For example, if the domain is 0.1, the correlation between, say red, with the number smaller than 5 is 0.1. Furthermore, the label is flipped at a constant rate: in this paper, 15 % label is flipped. Therefore, the correlation between color and digit is $d$, while between label and digits is 0.85. That is, what models should learn is the correlation between label and noise, resulting in classification accuracy of 0.85. However, if the model exploits spurious correlation of the domain, it will learn the correlation between label and color, resulting in training accuracy being 0.9 but test accuracy being 0.1 in this case.

PACS and Office-Home are image datasets whose domain determines the style of the image. These are benchmark datasets for domain generalization.

VLCS is a set of different photographic datasets, PASCAL VOC (Everingham et al., 2010), LabelMe (Russell et al., 2008), Caltech101 (Fei-Fei et al., 2004), and SUN09 (Choi et al., 2010). PASCAL VOC, LabelMe, and SUN09 are benchmark datasets for object detection. Caltech101 is 101 classes image datasets, where each class has 40 - 80 samples.

DomainNet is a large dataset proposed for the study of domain generalization. The number of classes, domains, and dataset size is the largest in the DomainBed dataset.

TerraIncognita is a dataset consisting of images taken by cameras at different locations. The difference in the camera's location corresponds to the difference in the domain.

RotatedMNIST is a dataset that artificially rotates MNIST and divides the domain according to the rotation angle. The number in the domain corresponds to the rotation angle.

We leave one domain for the final evaluation and use the remaining domains for training. To evaluate the performance during training, we split the data of each domain into training data and validation data. The split ratio is 80 % for training and 20 % for validation. We take an average of test accuracies and validation accuracies across domains, respectively, and use them to evaluate the OOD generalization.

## C.2  Backgrounds Challenge Dataset

In Section D.2, we explained that we use the subset of ImageNet (ORIGINAL). ORIGINAL consists of nine classes displayed in table 3. These classes are synthetically created from ImageNet classes based on WordNet ID. This table is a copy of a table in the original paper (Xiao et al., 2021).

Table 3: Backgrounds Challenge: Dataset information originally created in (Xiao et al., 2021)

| classes | WordNet ID | num sub-classes |
|---|---|---|
| Dog | n02084071 | 116 |
| Bird | n01503061 | 52 |
| Vehicle | n04576211 | 42 |
| Reptile | n01661091 | 36 |
| Carnivore | n02075296 | 35 |
| Insect | n02159955 | 27 |
| Instrument | n03800933 | 26 |
| Primate | n02469914 | 20 |
| Fish | n02512053 | 16 |

This dataset is filtered to balance samples across classes. We follow the same filtering procedure as the original paper. For further details, please refer to the original paper (Xiao et al., 2021).

### C.3 Amazon-WILDS and CivilComments-WILDS Dataset

WILDS is a set ob benchmark datasets with distributional shift and their variants: iWildCam (Beery et al., 2020), Camelyon17 (Bandi et al., 2018), RxRx1 (Taylor et al., 2019), OGB-MolPCBA (Hu et al., 2020), GlobalWheat (David et al., 2020; 2021), CivilComments (Borkan et al., 2019), FMoW (Christie et al., 2018), PovertyMap (Yeh et al., 2020), Amazon (Ni et al., 2019), and Py150 (Lu et al., 2021; Raychev et al., 2016). We use CivilComments and Amazon for our experiment as NLP tasks.

### C.4 CIFAR10-C and CIFAR10-P Dataset

First, for the corrupted or perturbed data generalization studies, we use CIFAR-10-C, CIFAR-10-P to evaluate the generalization performance Hendrycks & Dietterich (2019); Mu & Gilmer (2019). CIFAR-10-C consist of CIFAR image data corrupted by 19 noise patterns. CIFAR-10-P is similar, but with ten perturbations. Similar corruptions and perturbations frequently occur in real-world imaging applications. Thus, evaluating the robustness against this noise is essential for improving practical applicability. Because noises changes the input samples, we can regard the corrupted and perturbed data as being sampled from a different distribution $P(\boldsymbol{x}, y)'$ than the original distribution: $P(\boldsymbol{x}, y)' \neq P(\boldsymbol{x}, y)$. If a classifier $f : \mathcal{X} \to \mathcal{Y}$ is robust to corruption $c : \mathcal{X} \to \mathcal{X}$, or $\mathbb{E}_{c \sim C}\left[P(f(c(\boldsymbol{x}) = y))\right]$ with corruption distribution $C$, we can say that the classifier can generalize for the corrupted samples. The same is true for perturbation.

# D   Experimental Protocol

We employ a Bayesian hyperparameter search strategy in the sweep functionality of Weights&Biases[5]. In this strategy, the relationship between parameters and evaluation metrics is modeled as a Gaussian process, and the parameters are selected in such a way as to optimize the improvement probability. Table 4 shows the number of hyperparameter optimizations performed for each task. The transition of the best ood accuracy in these hyperparameter optimizations is shown in Appendix G.2.

In line with previous studies (Gulrajani & Lopez-Paz, 2021; Xiao et al., 2021; Koh et al., 2021) different benchmarks use different evaluation metrics, each of which is outlined in the following sections.

Table 4: Number of Experiments for Each Dataset

|  | SGD | Momentum | Nesterov | RMSprop | Adam |
|---|---|---|---|---|---|
| RotatedMNIST | 200 | 200 | 200 | 200 | 200 |
| ColoredMNIST | 200 | 342 | 200 | 200 | 200 |
| PACS | 200 | 200 | 200 | 200 | 412 |
| VLCS | 200 | 200 | 200 | 200 | 324 |
| OfficeHome | 399 | 200 | 200 | 200 | 699 |
| TerraIncognita | 200 | 200 | 200 | 451 | 202 |
| DomainNet | 490 | 515 | 857 | 1160 | 1137 |
| Amazon-WILDS | 440 | 438 | 449 | 489 | 466 |
| Amazon-CivilComments | 594 | 543 | 588 | 575 | 554 |
| Background Challenge | - | 347 | - | - | 567 |

## D.1   DomainBed

We follow the setting that is employed in the original paper (Gulrajani & Lopez-Paz, 2021). We train models with training domains and evaluate their performance on the test domain, which is the domain not used for training. We use ResNet-50 for PACS, VLCS, Office-Home, DomainNet, TerraIncognita, and MNIST ConvNet (Gulrajani & Lopez-Paz, 2021) for RotatedMNIST and Colored MNIST.

In our experiments, one domain is used as the test domain (OOD: out-of-distribution) and the other domain as the training domain (ID: in-distribution). More precisely, the test domain is *"Art"* for PACS, *"Caltech101"* for VLSC, *"Art"* for Office-Home, *"Clipart"* for DomainNet, *"L100"* for TerraIncognita, *"30°"* for RotatedMNIST, and *"-90%"* for ColoredMNIST. We explain the experimental configurations in Section E.1.

## D.2   Backgrounds Challenge Dataset

We follow Xiao et al. (2021) for using Backgrounds Challenge dataset for evaluation. Thus, we use the following data-generating procedure proposed by Xiao et al. (2021). We train ResNet-50 on ImageNet-1k with two popular optimizers, Momentum SGD, and Adam. The test datasets to evaluate the trained model are derivations of ImageNet dataset. First, we construct a subset of the ImageNet which has nine coarse-grained classes, e.g. insect (*ORIGINAL*). Especially, we refer to ORIGINAL with all images from ImageNet as *IN9L*. Then, we create a dataset by changing the background of the images of IN9L. In particular, we change the background of each image into a random background cropped from another image in ORIGINAL. We call this dataset *Mixed-Random*, following Xiao et al. (2021). By comparing the accuracy of Mixed-Same with that of ORIGINAL, we can measure the dependence of the model on the spurious correlation of background information. Thus, we investigate the relations between these two accuracies. The search range for hyperparameters is shown in Section E.2.

---

[5]https://wandb.ai/site

### D.3 WILDS

We also follow the setting that is employed in the original paper (Koh et al., 2021). First, we divide the dataset into train, validation, and test and train a model using the train data. For model selection, we use the performance evaluation of validation data. In the test dataset, considering the subpopulation shift, we measure the performance of the OOD in the worst group for CivilComments-WILDS and in the domain of 10-percentile for Amazon-WILDS. In both Amazon-WILDS and CivilComments-WILDS, DistilBERT (Sanh et al., 2019) is used as the deep neural network model architecture. We explain the further details of the experimental configurations in Section E.3.

### D.4 CIFAR10-C and CIFAR10-P

First, we trained deep neural networks with CIFAR-10[6]. Then we evaluated the trained models with CIFAR-10-C, CIFAR-10-P[7]. For the corruption datasets (CIFAR-10-C), we compared the accuracy for the corruption or perturbation test samples (samples from CIFAR-10-C) with that for the training domain test samples (samples from CIFAR-10). For the perturbation dataset, we measured the perturbation robustness introduced by Hendrycks & Dietterich (2019).

---

[6]https://www.cs.toronto.edu/~kriz/cifar.html
[7]https://zenodo.org/record/2535967

# E   Hyperparameters and Detailed Configurations

We report the hyperparameter's search space. For vanilla SGD, we search learning rate $\eta$, learning rate decay rate $\rho$ and the timing to decay learning rate $\delta$, and regularization coefficient of weight decay $\lambda$. When $\delta = 0.7$, it means that the learning rate decays when training passes 70 % of the total iterations. We do not search $\rho$ and $\delta$ for DomainBed because we do not employ a learning rate schedule.

For non-adaptive momentum methods, a parameter to control momentum $\gamma$ is added to the hyperparameters. For RMSprop, we further add parameters $\alpha$ and $\epsilon$, which control the second-order momentum and numerical stability, respectively. Although $\epsilon$ is originally introduced for numerical stability, this parameter is found to play a crucial role in generalization performance (Choi et al., 2019). Thus, we follow Choi et al. and vary this parameter as well.

For Adam, we add $\epsilon$, $\beta_1$, $\beta_2$ to vanilla SGD's configuration. The parameter $\epsilon$ is the same as that for RMSprop and $\beta_1$ and $\beta_2$ control first and second-order momentum terms, respectively.

## E.1   DomainBed

We conduct Bayesian optimization for hyperparameter search of DomainBed. First, we sampled hyperparameters from uniform distribution whose minimum and maximum values are shown in the table 6 and 7 Then, we conducted Bayesian optimization on these sampled candidate hyperparameters and selected some of the hyperparameters among them. For reference, the scatter plot of Adam's learning rate for the OfficeHome dataset is shown in Appendix H.1. Note that the values for batch size $B$ in the tables do not indicate minimum and maximum for Bayesian optimization but those for grid search. Unlike other datasets, we implement IRMv1 in addition to ERM. IRMv1 is a heuristic optimization problem to solve IRM, introduced by Arjovsky et al. (2019). Thus, we search the hyperparameters of IRMv1 as well.

IRMv1 is the following constrained optimization problem (Arjovsky et al., 2019):

$$\min_{\Phi:\mathcal{X}\to\mathcal{Y}} \sum_{e\in\mathcal{E}_{\mathrm{tr}}} R^e(\Phi) + \lambda_{\mathrm{IRM}} \left\| \nabla_{w|w=1.0} R^e(w \circ \Phi) \right\|^2, \tag{14}$$

where $\Phi$ is representation function and $w$ is the weight on top of the function of a model $f = w \circ \Phi$. $\mathcal{X}$ is the input domain and $\mathcal{Y}$ is the output domain. The character $e$ indicates an environment in training environment set $\mathcal{E}_{\mathrm{tr}}$ and $R^e$ is the risk of the environment. Because this is the optimization with regularization term, we search coefficient $\lambda_{\mathrm{IRM}}$. In addition, we implement annealing of this coefficient and so we try various penalty annealing iterations $N_{\mathrm{IRM}}$ too. The basic workload is summarized in table 5.

We made two changes to the experimental setup in the original paper, as well as to the optimizer. We added regularization to the training of ColoredMNIST and RotatedMNIST to stabilize the learning. In addition, we increased the steps budget for RotatedMNIST because we observed that the training loss of RotatedMNIST was not sufficiently reduced.

We also experimented with two additional model architectures (ResNet-20 and Vision Transformer) for the benchmarks proposed in the existing DomainBed.

Table 5: DomainBed: Workloads

| Model | Dataset | Batch size | Step Budget |
|---|---|---|---|
| MNIST ConvNet (Gulrajani & Lopez-Paz, 2021) | Colored MNIST | [128, 512, 2048] | 5000 |
| MNIST ConvNet (Gulrajani & Lopez-Paz, 2021) | Rotated MNIST | [128, 512, 2048] | 100K |
| ResNet-50 | VLCS | [64, 128] | 5000 |
| ResNet-50 | PACS | [64, 128] | 5000 |
| ResNet-50 | Office Home | [64, 128] | 5000 |
| ResNet-50 | DomainNet | [64, 128] | 5000 |
| ResNet-50 | TerraIncognita | [64, 128] | 5000 |
| Vision Transformer (for Additional Study) | PACS | [64] | 5000 |
| ResNet-20 (for Additional Study) | Colored MNIST | [64] | 5000 |

Table 6: DomainBed: ResNet-50

| | $B$ | $\eta$ | $\lambda$ | $\gamma$ | $\alpha$ | $\epsilon$ | $\lambda_{\mathrm{IRM}}$ | $N_{\mathrm{IRM}}$ |
|---|---|---|---|---|---|---|---|---|
| SGD | [64, 128] | [1e-5, 1e-2] | [1e-6, 1e-2] | - | - | - | [1e-1, 1e+5] | [1e+0, 1e+4] |
| Momentum | [64, 128] | [1e-5, 1e-2] | [1e-6, 1e-2] | [0, 0.99999] | - | - | [1e-1, 1e+5] | [1e+0, 1e+4] |
| Nesterov | [64, 128] | [1e-5, 1e-2] | [1e-6, 1e-2] | [0, 0.99999] | - | - | [1e-1, 1e+5] | [1e+0, 1e+4] |
| RMSprop | [64, 128] | [1e-5, 1e-2] | [1e-6, 1e-2] | [0, 0.99999] | [0, 0.999] | [1e-8, 1e-3] | [1e-1, 1e+5] | [1e+0, 1e+4] |

| | $B$ | $\eta$ | $\lambda$ | $\beta_1$ | $\beta_2$ | $\epsilon$ | $\lambda_{\mathrm{IRM}}$ | $N_{\mathrm{IRM}}$ |
|---|---|---|---|---|---|---|---|---|
| Adam | [64, 128] | [1e-5, 1e-2] | [1e-6, 1e-2] | [0, 0.999] | [0, 0.999] | [1e-8, 1e-3] | [1e-1, 1e+5] | [1e+0, 1e+4] |

| | $B$ | $\eta$ | $\gamma$ | $\epsilon$ | $\rho$ |
|---|---|---|---|---|---|
| SAM | [64] | [1e-5, 1e-2] | [0.9] | [1e-4] | [5e-2] |

Table 7: DomainBed: MNIST ConvNet (Gulrajani & Lopez-Paz, 2021)

| | $B$ | $\eta$ | $\lambda$ | $\gamma$ | $\alpha$ | $\epsilon$ | $\lambda_{\mathrm{IRM}}$ | $N_{\mathrm{IRM}}$ |
|---|---|---|---|---|---|---|---|---|
| SGD | [128, 521, 2048] | [1e-5, 1e-2] | [1e-6, 1e-2] | - | - | - | | |
| Momentum | [128, 521, 2048] | [1e-5, 1e-2] | [1e-6, 1e-2] | [0, 0.99999] | - | - | [1e-1, 1e+5] | [1e+0, 1e+4] |
| Nesterov | [128, 521, 2048] | [1e-5, 1e-2] | [1e-6, 1e-2] | [0, 0.99999] | - | - | [1e-1, 1e+5] | [1e+0, 1e+4] |
| RMSprop | [128, 521, 2048] | [1e-5, 1e-2] | [1e-6, 1e-2] | [0, 0.99999] | [0, 0.999] | [1e-8, 1e-3] | [1e-1, 1e+5] | [1e+0, 1e+4] |

| | $B$ | $\eta$ | $\lambda$ | $\beta_1$ | $\beta_2$ | $\epsilon$ | $\lambda_{\mathrm{IRM}}$ | $N_{\mathrm{IRM}}$ |
|---|---|---|---|---|---|---|---|---|
| Adam | [128, 521, 2048] | [1e-5, 1e-2] | - | [0, 0.999] | [0, 0.999] | [1e-8, 1e-3] | [1e-1, 1e+5] | [1e+0, 1e+4] |

### E.2  Backgrounds Challenge Dataset

We conduct Bayesian optimization for the Backgrounds Challenge dataset as well. We use 4096 as the batch size. For all configurations other than batch size, as we follow Choi et al. (2019).

We conduct the hyperparameter search and restart training from scratch. Of the trained models, we evaluate trained model performance in the OOD dataset.

Table 8: Backgrounds Challenge: ResNet-50

|          | $\eta$ | $\lambda$ | $\gamma$ |
|----------|--------|-----------|----------|
| Momentum | [1e-5, 1e-2] | [1e-6, 1e-2] | [0, 0.99999] |

|      | $\eta$ | $\lambda$ | $\beta_1$ | $\beta_2$ | $\epsilon$ |
|------|--------|-----------|-----------|-----------|------------|
| Adam | [1e-5, 1e-2] | [1e-6, 1e-2] | [0, 0.999] | [0, 0.999] | [1e-8, 1e-3] |

### E.3  WILDS

We conduct Bayesian optimization for the hyperparameter search of WILDS. The hyperparameters are sampled by uniform distribution whose minimum and maximum values are shown in the table 9 and 10. Note that the values for batch size $B$ is fixed in these experiments due to computational efficiency. We implement IRMv1 as well as DomainBed experiments.

For training the Amazon-WILDS and CivilComments-WILDS datasets that we used as NLP datasets, we followed the deep neural network model and training configuration proposed in the original paper. DistilBERT, a distillation of BERT Base, is used as the deep neural network model.

Table 9: WILDS-Amazon: DistilBERT

|          | $B$ | $\eta$ | $\lambda$ | $\gamma$ | $\alpha$ | $\epsilon$ | $\lambda_{\text{IRM}}$ | $N_{\text{IRM}}$ |
|----------|-----|--------|-----------|----------|----------|------------|------------------------|------------------|
| SGD      | [8] | [1e-5, 1e-2] | [1e-6, 1e-2] | - | - | - | [1e-1, 1e+5] | [1e+0, 1e+4] |
| Momentum | [8] | [1e-5, 1e-2] | [1e-6, 1e-2] | [0, 0.99999] | - | - | [1e-1, 1e+5] | [1e+0, 1e+4] |
| Nesterov | [8] | [1e-5, 1e-2] | [1e-6, 1e-2] | [0, 0.99999] | - | - | [1e-1, 1e+5] | [1e+0, 1e+4] |
| RMSprop  | [8] | [1e-5, 1e-2] | [1e-6, 1e-2] | [0, 0.99999] | [0, 0.999] | [1e-8, 1e-3] | [1e-1, 1e+5] | [1e+0, 1e+4] |

|      | $B$ | $\eta$ | $\lambda$ | $\beta_1$ | $\beta_2$ | $\epsilon$ | $\lambda_{\text{IRM}}$ | $N_{\text{IRM}}$ |
|------|-----|--------|-----------|-----------|-----------|------------|------------------------|------------------|
| Adam | [8] | [1e-5, 1e-2] | [1e-6, 1e-2] | [0, 0.999] | [0, 0.999] | [1e-8, 1e-3] | [1e-1, 1e+5] | [1e+0, 1e+4] |

|       | $B$ | $\eta$ | $\beta_1$ | $\beta_2$ | $\epsilon$ |
|-------|-----|--------|-----------|-----------|------------|
| AdamW | [8] | [1e-5, 1e-2] | [0.9] | [0.999] | [1e-8] |

|     | $B$ | $\eta$ | $\gamma$ | $\epsilon$ | $\rho$ |
|-----|-----|--------|----------|------------|--------|
| SAM | [8] | [1e-5, 1e-2] | [0.9] | [1e-4] | [5e-2] |

Table 10: WILDS-CivilComments: DistilBERT

|  | $B$ | $\eta$ | $\lambda$ | $\gamma$ | $\alpha$ | $\epsilon$ | $\lambda_{\text{IRM}}$ | $N_{\text{IRM}}$ |
|---|---|---|---|---|---|---|---|---|
| SGD | [16] | [1e-5, 1e-2] | [1e-6, 1e-2] | - | - | - | [1e-1, 1e+5] | [1e+0, 1e+4] |
| Momentum | [16] | [1e-5, 1e-2] | [1e-6, 1e-2] | [0, 0.99999] | - | - | [1e-1, 1e+5] | [1e+0, 1e+4] |
| Nesterov | [16] | [1e-5, 1e-2] | [1e-6, 1e-2] | [0, 0.99999] | - | - | [1e-1, 1e+5] | [1e+0, 1e+4] |
| RMSprop | [16] | [1e-5, 1e-2] | [1e-6, 1e-2] | [0, 0.99999] | [0, 0.999] | [1e-8, 1e-3] | [1e-1, 1e+5] | [1e+0, 1e+4] |

|  | $B$ | $\eta$ | $\lambda$ | $\beta_1$ | $\beta_2$ | $\epsilon$ | $\lambda_{\text{IRM}}$ | $N_{\text{IRM}}$ |
|---|---|---|---|---|---|---|---|---|
| Adam | [16] | [1e-5, 1e-2] | [1e-6, 1e-2] | [0, 0.999] | [0, 0.999] | [1e-8, 1e-3] | [1e-1, 1e+5] | [1e+0, 1e+4] |

### E.4 CIFAR10-C and CIFAR10-P

**Corruption:** The corruption dataset we consider contains 19 corruption patterns. These are *Gaussian Noise*, *Shot Noise*, *Impulse Noise*, *Defocus Blur*, *Frosted Glass Blur*, *Motion Blur*, *Zoom Blue*, *Show*, *Frost*, *Fog*, *Brightness*, *Contrast*, *Elastic*, *Pixelate*, and *JPEG*. Sample corrupted images are shown in the original paper by Hendrycks & Dietterich (2019). Following Hendrycks & Dietterich (2019), we compute the classification error $E_{s,c}$ for each corruption $c$ with a five-point scale for the noise severity $s$. Averaging over a five-point scale, we measure the classification accuracy for each corruption $\text{Acc}_c$ as follows:

$$\text{Acc}_c = 1 - \frac{\sum_{s=1}^{5} E_{s,c}}{5} \tag{15}$$

This accuracy is the test accuracy (corruption) we defined in the main body of this paper. We evaluate the generalization performance by comparing this quantity with the accuracy calculated using the original test dataset $\text{Acc} = 1 - E$. Note that $E$ is the classification error for the test (training domain) samples.

**Perturbation:** The perturbation dataset has six corruption patterns that are the same as in the corruption dataset and four additional digital transformations: *Gaussian Noise*, *Shot Noise*, *Motion Blur*, *Zoom Blur*, *Show*, *Brightness*, *Translate*, *Rotate*, *Tilt*, and *Scale*. The perturbation dataset differs from the corruption dataset in that each corruption generates more than thirty perturbation sequences. Hence, we use not a simple measure of accuracy but the following metric to determine the perturbation robustness:

$$d\left(\tau(\boldsymbol{x}), \tau\left(\boldsymbol{x}'\right)\right) = \sum_{i=1}^{5} \sum_{j=\min\{i,\sigma(i)\}+1}^{\max\{i,\sigma(i)\}} \mathbf{1}(1 \le j - 1 \le 5), \tag{16}$$

where $\tau(\boldsymbol{x})$ is the ranked prediction of the classifier on image $\boldsymbol{x}$ and $\sigma(i) = \tau(\boldsymbol{x}')/\tau(\boldsymbol{x})$. This criterion measures how the top-5 prediction on two different images differs. By using this quantity, top-5 robustness perturbation can be written as follows:

$$\text{uT5D}_p = \frac{1}{m(n-1)} \sum_{i=1}^{m} \sum_{j=2}^{n} d\left(\tau\left(\boldsymbol{x}_j\right), \tau\left(\boldsymbol{x}_{j-1}\right)\right). \tag{17}$$

Table 11: Hyperparameter Search Range: ResNet-8 / CIFAR10

|  | $B$ | $\eta$ | $\lambda$ | $\gamma$ |
|---|---|---|---|---|
| Momentum | 256 | [1e-6, 1e+1] | [1e-5, 1e-4] | [1e-4 0.9999] |

|  | $B$ | $\eta$ | $\lambda$ | $\beta_1$ | $\beta_2$ | $\epsilon$ |
|---|---|---|---|---|---|---|
| Adam | 256 | [1e-4, 1e-1] | [1e-5, 1e-4] | [0.9, 0.999] | [0.99, 0.9999] | [1e-4, 1e-2] |

## F    Full Results of Experiments

We conducted an evaluation of optimizers using two distinct model selection strategies. One approach is Oracle-based, while the other employs a method of training-domain validation set (Gulrajani & Lopez-Paz, 2021). In Section F.1, we compare the OOD accuracy of models obtained through each strategy, as displayed in a tabular format. Particularly for the latter strategy, we present both the average and variance of the Top-10 models, as well as the average and variance for models surpassing a certain threshold.

The rationale behind providing the average and variance of the Top-10 models lies in our adoption of Bayesian optimization for hyperparameters, with the aim of illustrating the error bars of high-performing models obtained through this method. The results for models exceeding a certain threshold (in the rightmost bin) are shared to facilitate comparison of average OOD accuracy performance when non-adaptive and adaptive optimization methods are comparable in terms of ID accuracy.

In Section F.2, we present the results for all hyperparameters we explored (except for the model inferior to random guess), along with their distributions in a box plot. Finally, in Section F.4, we present the OOD accuracy and ID accuracy results for all the results from which these results are derived as a scatter plot.

### F.1    Full Results of Table

**Oracle:**
In this section, we provide additional information on Table 1 in the paper. Table 12 shows the mean and standard deviation for the top-10 models, including those selected by the Oracle model selection method. In eight of the ten datasets, the non-adaptive method outperforms the adaptive method in out-of-distribution accuracy.

Table 12:  Top-10 Trials: Mean and Standard Deviation for Each Dataset (Model Selection by Oracle). The variance is relatively suppressed in many tasks because it is the mean and variance of the model that achieves Top-10 OOD accuracy.

| Model | OOD Dataset | Non-Adaptive Optimizer | | | Adaptive Optimizer | |
|---|---|---|---|---|---|---|
| | | SGD | Momentum | Netsterov | RMSProp | Adam |
| 4-Layer CNN | RotatedMNIST | $89.62_{\pm0.18}$ | $93.23_{\pm0.69}$ | $92.89_{\pm0.2}$ | $95.81_{\pm0.16}$ | $\mathbf{96.05_{\pm0.16}}$ |
| | ColoredMNIST | $14.64_{\pm0.07}$ | $29.80_{\pm1.84}$ | $28.85_{\pm2.0}$ | $\mathbf{52.82_{\pm6.21}}$ | $52.39_{\pm6.89}$ |
| ResNet50 | VLCS | $\mathbf{98.98_{\pm0.1}}$ | $98.70_{\pm0.16}$ | $98.49_{\pm0.12}$ | $96.23_{\pm1.39}$ | $97.09_{\pm1.61}$ |
| | PACS | $86.56_{\pm0.83}$ | $\mathbf{86.90_{\pm0.82}}$ | $86.01_{\pm1.36}$ | $81.10_{\pm2.93}$ | $82.59_{\pm2.93}$ |
| | OfficeHome | $63.68_{\pm0.12}$ | $63.52_{\pm0.44}$ | $62.40_{\pm0.47}$ | $55.34_{\pm3.39}$ | $62.12_{\pm0.35}$ |
| | TerraIncognita | $52.22_{\pm1.62}$ | $\mathbf{56.05_{\pm1.96}}$ | $52.98_{\pm1.61}$ | $53.67_{\pm2.62}$ | $48.78_{\pm3.53}$ |
| | DomainNet | $55.31_{\pm2.38}$ | $56.12_{\pm3.85}$ | $\mathbf{58.19_{\pm2.2}}$ | $52.71_{\pm2.74}$ | $55.74_{\pm1.65}$ |
| | BackgroundChallenge | - | $\mathbf{78.99_{\pm0.48}}$ | - | - | $75.55_{\pm1.5}$ |
| DistilBERT | WILDS_amazon | $52.00_{\pm0.0}$ | $\mathbf{54.67_{\pm0.0}}$ | $54.67_{\pm0.0}$ | $48.67_{\pm3.46}$ | $52.44_{\pm1.5}$ |
| | WILDS_civilcomments | $51.81_{\pm0.29}$ | $\mathbf{56.40_{\pm0.61}}$ | $55.98_{\pm0.47}$ | $38.42_{\pm7.04}$ | $37.71_{\pm4.0}$ |
| ResNet-20 | ColoredMNIST | - | $28.93_{\pm2.63}$ | - | - | $\mathbf{29.84_{\pm1.64}}$ |
| ViT | PACS | - | $\mathbf{90.28_{\pm0.54}}$ | - | - | $90.05_{\pm0.29}$ |

**Training-Domain Validation Set:**    Table 13 shows the results of top-10 models with training-domain validation set (Gulrajani & Lopez-Paz, 2021) as the model selection method. It shows that 11 out of 12 tasks showed that the non-adaptive optimization method is superior.    As exhibited in Table 14, we present the out-of-distribution accuracy along with the mean and standard deviation of the subset exhibiting high

validation performance within our targeted training domain, as graphically depicted in Figure 1. This data specifically corresponds to the apex of the rightmost bin in the bar chart encapsulated in Figure 1. As noted, the relatively elevated standard deviations observed can be primarily attributed to our calculation methodology. This approach calculates the mean and standard deviation values of the out-of-distribution accuracy, specifically focusing on the rightmost bins indicative of high validation accuracy as portrayed in Figure 1. It is crucial to acknowledge that this unique approach could potentially yield a broader range of variances.

Table 13: Top-10 Trials: Mean and Standard Deviation for Each Dataset (Model Selection by Training-Domain Validation Set). The variance is relatively suppressed in many tasks because it is the mean and variance of the model that achieves Top-10 OOD accuracy.

| Model | OOD Dataset | Non-Adaptive Optimizer | | | Adaptive Optimizer | |
|---|---|---|---|---|---|---|
| | | SGD | Momentum | Netsterov | RMSProp | Adam |
| 4-Layer CNN | RotatedMNIST | $89.13_{\pm 0.56}$ | $93.09_{\pm 0.82}$ | $91.36_{\pm 1.4}$ | $95.51_{\pm 0.48}$ | $\mathbf{95.76_{\pm 0.43}}$ |
| | ColoredMNIST | $10.4_{\pm 0.3}$ | $10.04_{\pm 0.07}$ | $\mathbf{10.09_{\pm 0.44}}$ | $9.84_{\pm 0.18}$ | $10.08_{\pm 0.4}$ |
| ResNet50 | VLCS | $\mathbf{98.32_{\pm 0.34}}$ | $98.15_{\pm 0.57}$ | $97.93_{\pm 0.28}$ | $94.49_{\pm 3.9}$ | $96.76_{\pm 2.56}$ |
| | PACS | $85.01_{\pm 1.27}$ | $\mathbf{85.60_{\pm 1.21}}$ | $84.91_{\pm 1.53}$ | $81.10_{\pm 2.93}$ | $82.36_{\pm 3.11}$ |
| | OfficeHome | $62.62_{\pm 0.74}$ | $\mathbf{62.65_{\pm 1.02}}$ | $60.98_{\pm 1.46}$ | $55.02_{\pm 3.94}$ | $60.09_{\pm 1.08}$ |
| | TerraIncognita | $\mathbf{46.84_{\pm 4.15}}$ | $44.67_{\pm 9.2}$ | $45.21_{\pm 4.88}$ | $44.48_{\pm 7.96}$ | $41.82_{\pm 7.65}$ |
| | DomainNet | $55.25_{\pm 2.51}$ | $55.96_{\pm 4.1}$ | $\mathbf{58.19_{\pm 2.2}}$ | $52.71_{\pm 2.74}$ | $55.67_{\pm 1.78}$ |
| | BackgroundChallenge | - | $\mathbf{78.99_{\pm 0.48}}$ | - | - | $75.55_{\pm 1.5}$ |
| DistilBERT | WILDS_amazon | $52.00_{\pm 0.0}$ | $\mathbf{54.13_{\pm 0.69}}$ | $53.77_{\pm 0.63}$ | $50.93_{\pm 3.25}$ | $52.44_{\pm 1.5}$ |
| | WILDS_civilcomments | $49.17_{\pm 1.09}$ | $50.09_{\pm 1.91}$ | $\mathbf{50.40_{\pm 1.54}}$ | $31.65_{\pm 12.56}$ | $33.13_{\pm 5.89}$ |
| ResNet-20 | ColoredMNIST | - | $\mathbf{11.00_{\pm 0.52}}$ | - | - | $10.09_{\pm 0.15}$ |
| ViT | PACS | - | $\mathbf{90.28_{\pm 0.54}}$ | - | - | $90.05_{\pm 0.29}$ |

Table 14: Trials in Rightmost Bin: Mean and Standard Deviation for Each Dataset (Model Selection by Training-Domain Validation Set). The variance is relatively large because we compute the mean and variance of models that exceed a specific threshold (the rightmost bin in Figure 1) in ID accuracy.

| Model | OOD Dataset | Non-Adaptive Optimizer | | | Adaptive Optimizer | |
|---|---|---|---|---|---|---|
| | | SGD | Momentum | Netsterov | RMSProp | Adam |
| 4-Layer CNN | RotatedMNIST | $82.83_{\pm 6.15}$ | $87.84_{\pm 5.27}$ | $87.43_{\pm 5.58}$ | $94.29_{\pm 1.22}$ | $\mathbf{94.43_{\pm 1.91}}$ |
| | ColoredMNIST | $10.93_{\pm 1.61}$ | $12.86_{\pm 4.66}$ | $10.44_{\pm 2.53}$ | $\mathbf{22.84_{\pm 6.45}}$ | $16.12_{\pm 7.98}$ |
| ResNet50 | VLCS | $\mathbf{97.95_{\pm 1.02}}$ | $96.81_{\pm 2.96}$ | $96.54_{\pm 2.38}$ | $94.07_{\pm 3.4}$ | $94.49_{\pm 6.59}$ |
| | PACS | $\mathbf{79.67_{\pm 4.35}}$ | $78.58_{\pm 6.07}$ | $78.11_{\pm 5.9}$ | $74.98_{\pm 8.59}$ | $70.86_{\pm 6.96}$ |
| | OfficeHome | $\mathbf{61.11_{\pm 2.07}}$ | $60.38_{\pm 2.35}$ | $58.29_{\pm 2.58}$ | $55.98_{\pm 3.39}$ | $57.74_{\pm 3.1}$ |
| | TerraIncognita | $\mathbf{42.56_{\pm 5.31}}$ | $42.22_{\pm 8.33}$ | $41.44_{\pm 6.3}$ | $41.96_{\pm 10.36}$ | $40.76_{\pm 8.06}$ |
| | DomainNet | $55.79_{\pm 1.94}$ | $57.40_{\pm 3.1}$ | $\mathbf{57.51_{\pm 2.55}}$ | $55.09_{\pm 0.65}$ | $56.05_{\pm 1.41}$ |
| | BackgroundChallenge | - | $\mathbf{74.98_{\pm 4.7}}$ | - | - | $69.14_{\pm 5.61}$ |
| DistilBERT | Amazon-WILDS | $50.58_{\pm 1.07}$ | $\mathbf{52.20_{\pm 1.49}}$ | $52.00_{\pm 1.56}$ | $51.73_{\pm 1.55}$ | $51.32_{\pm 1.83}$ |
| | CivilComments-WILDS | $42.24_{\pm 5.8}$ | $\mathbf{44.79_{\pm 6.57}}$ | $44.15_{\pm 6.37}$ | $21.79_{\pm 9.59}$ | $23.55_{\pm 7.43}$ |
| ResNet-20 | ColoredMNIST | - | $19.62_{\pm 7.35}$ | - | - | $\mathbf{21.70_{\pm 7.66}}$ |
| ViT | PACS | - | $\mathbf{87.07_{\pm 5.52}}$ | - | - | $85.17_{\pm 7.67}$ |

## F.2 Full Results of Boxplot

Since we could not include all the results in the main paper, we report all the OOD accuracy, ID accuracy and their difference (we denote as gap) results including ERM and IRM results in a box plot.

### F.2.1 Full Results of Filtered Boxplot (ERM)

What we want to find out is the OOD generalization performance for models that perform better than random guess in the ID test set, i.e., models that have been trained. As Filtered Results, we define random guess = 1/num of classes in each problem setting, and show the results of eliminating those models whose ID accuracy does not exceed this threshold.

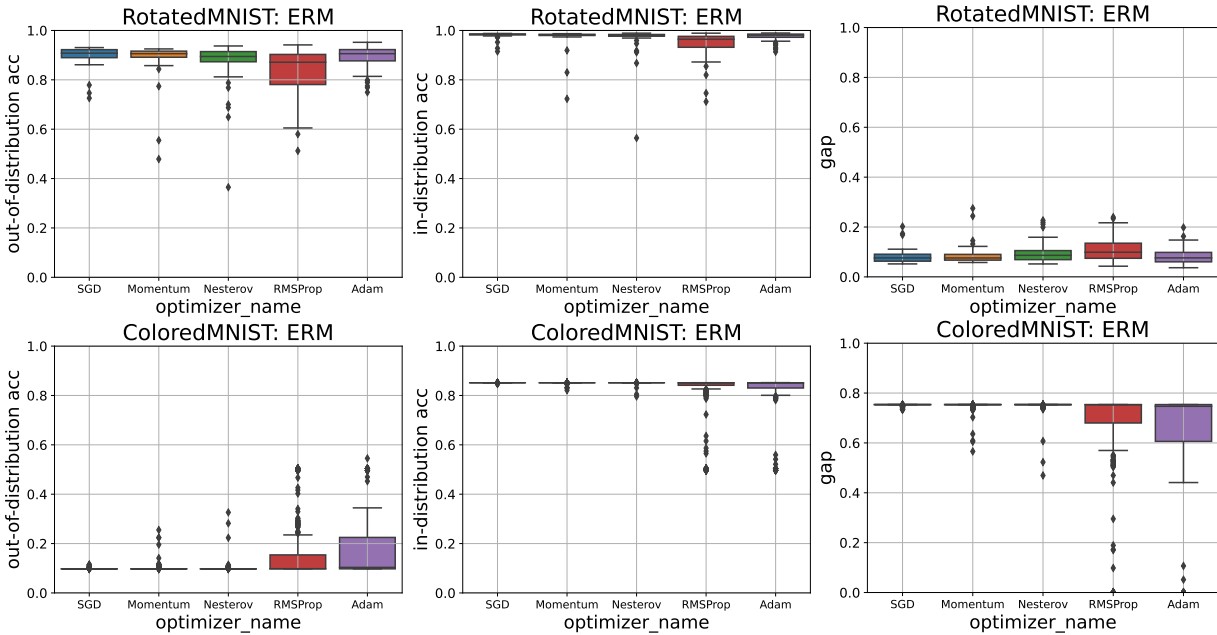

Figure 6. Filtered Results of RotatedMNIST and ColoredMNIST in DomainBed: Comparison of the in-distribution (validation) accuracy and the out-of-distribution (test) accuracy of ERM across five optimizers.

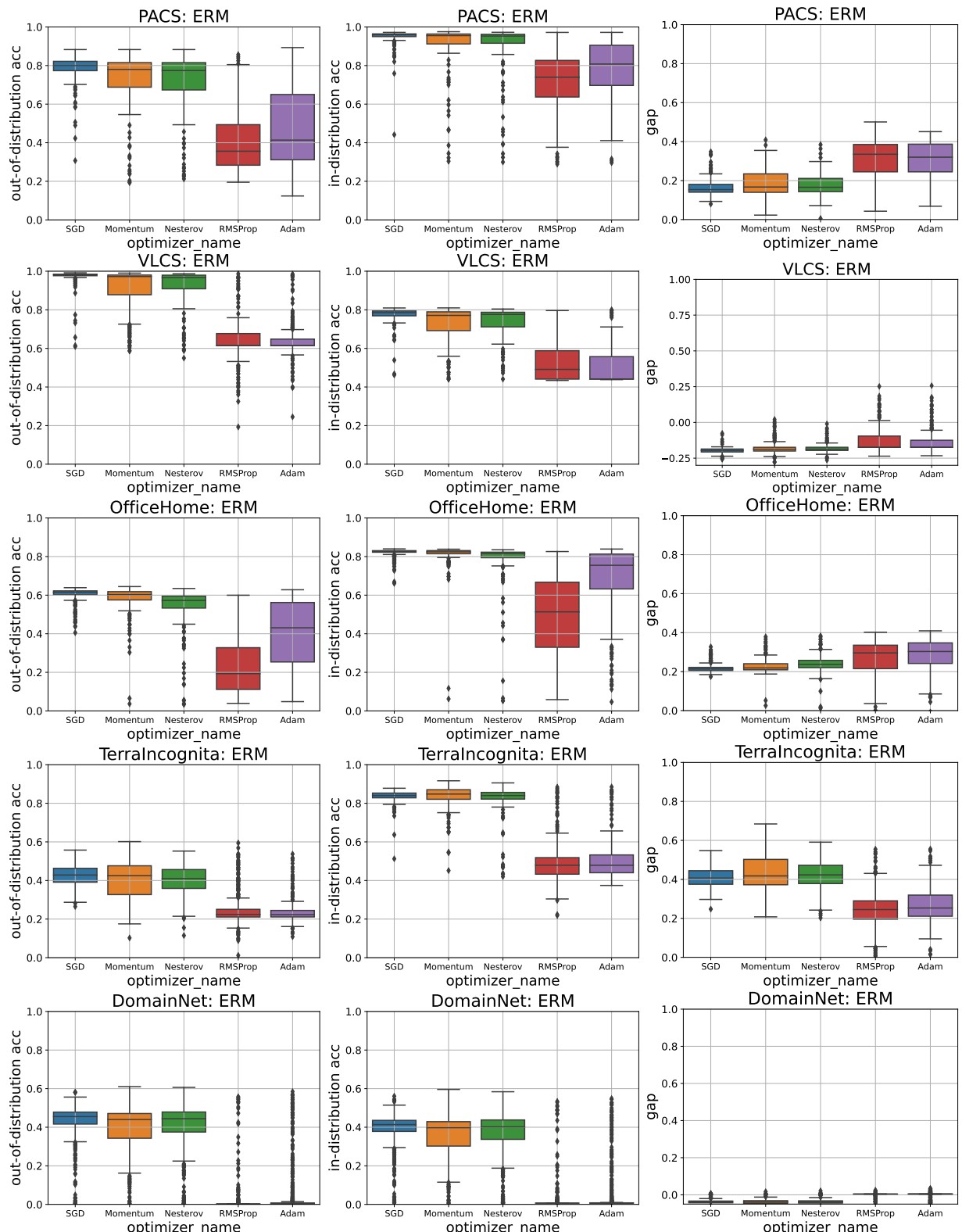

Figure 7. Filtered Results of PACS, VLCS, OfficeHome, TerraIncognita, and DomainNet in DomainBed: Comparison of the in-distribution (validation) accuracy and the out-of-distribution (test) accuracy of ERM across five optimizers.

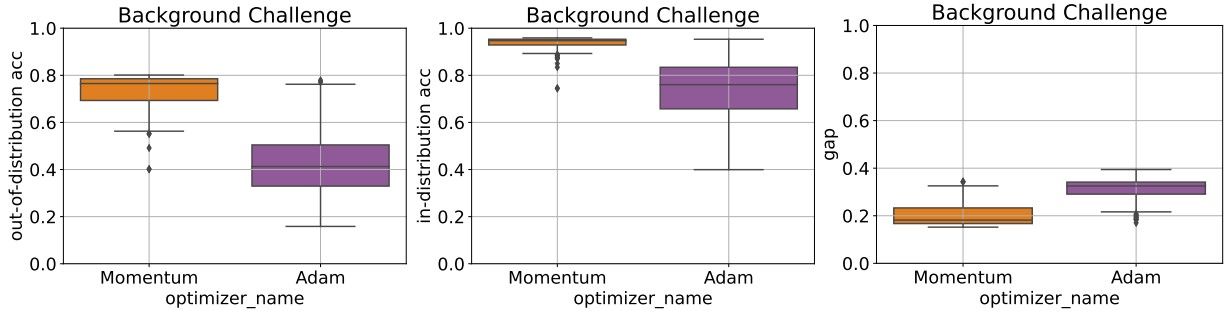

Figure 8. Filtered Results of Backgrounds Challenge: Comparison of the in-distribution (validation) accuracy and the out-of-distribution (test) accuracy of ERM across five optimizers.

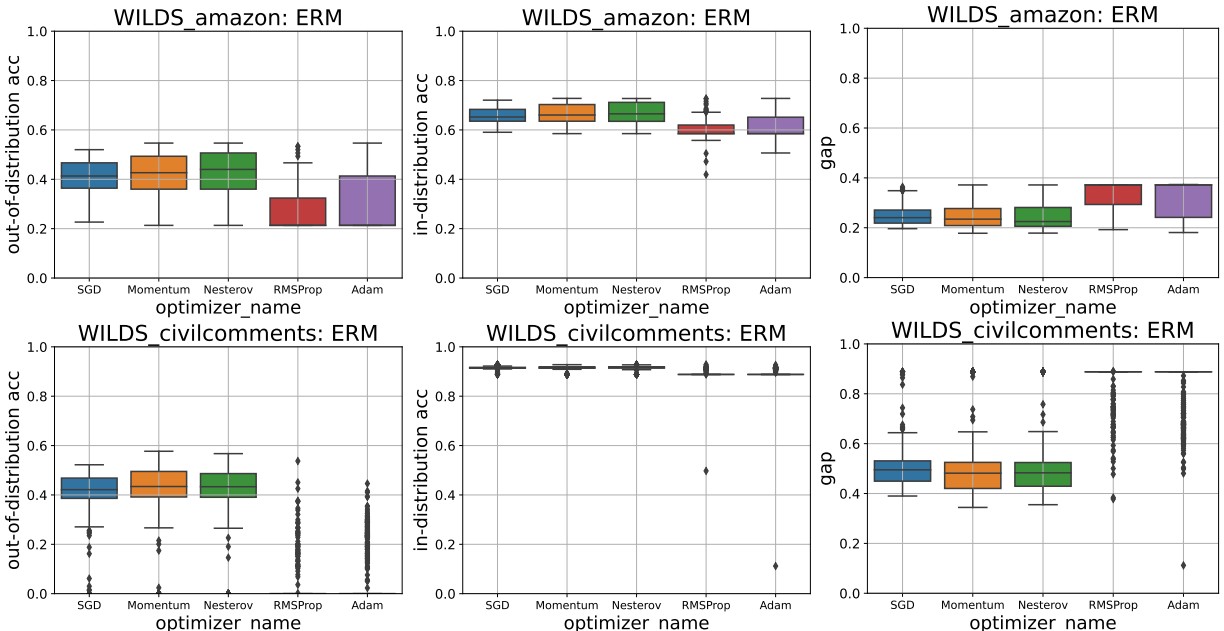

Figure 9. Filtered Results of Amazon-WILDS and CivilComments-WILDS: Comparison of the in-distribution (validation) accuracy and the out-of-distribution (test) accuracy of ERM across five optimizers.

### F.2.2 Full Results of Filtered Boxplot (IRM)

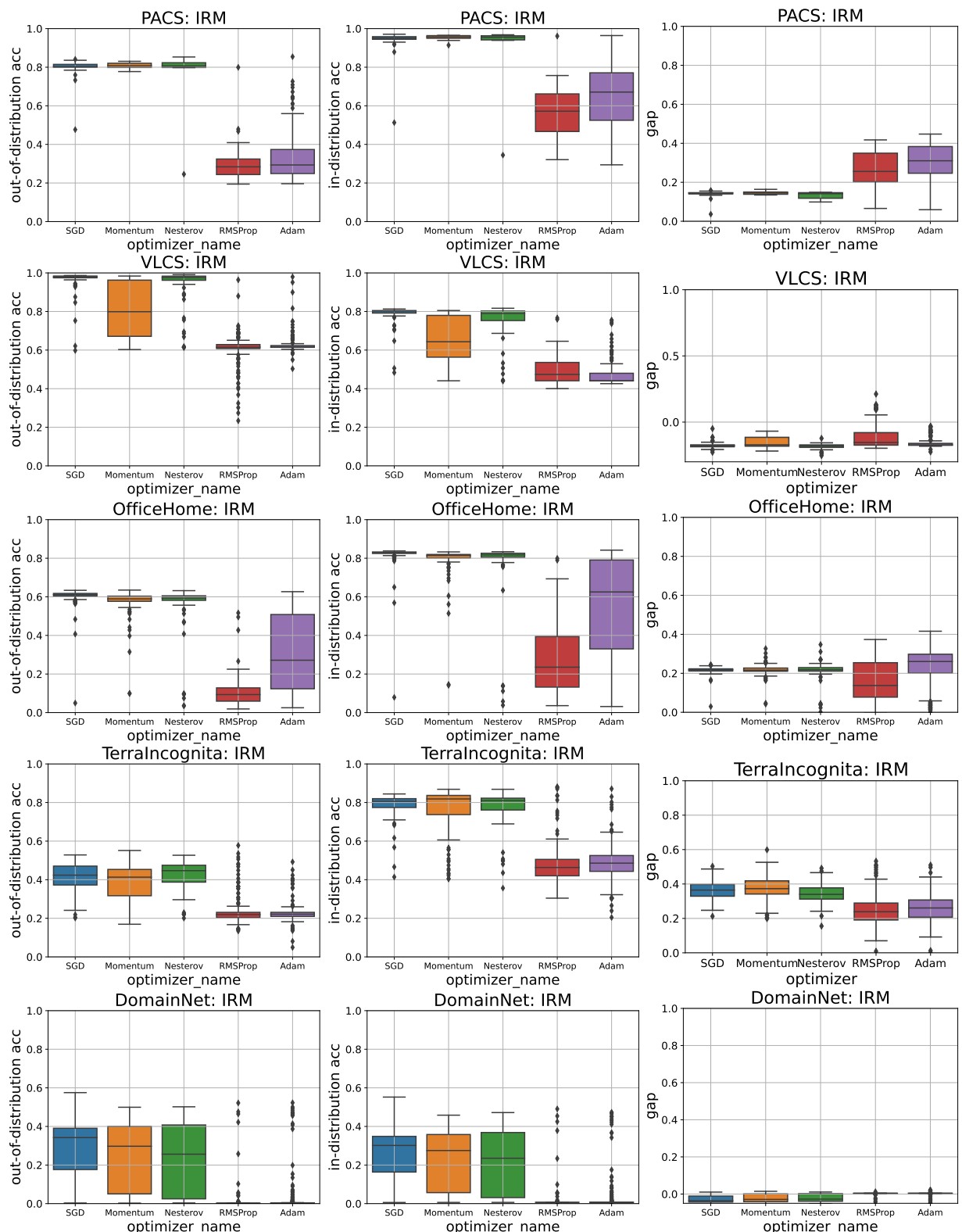

Figure 10. Filtered Results of PACS, VLCS, OfficeHome, TerraIncognita and DomainNet in DomainBed: Comparison of the in-distribution (validation) accuracy and the out-of-distribution (test) accuracy of IRM across five optimizers.

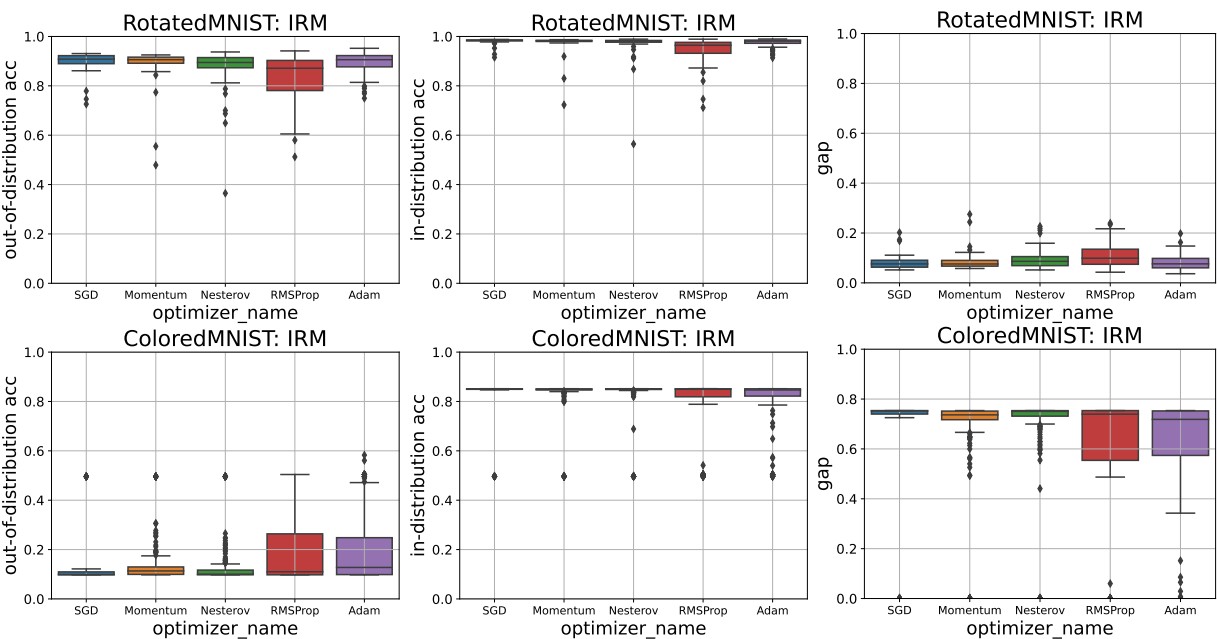

Figure 11. Filtered Results of RotatedMNIST and ColoredMNIST in DomainBed: Comparison of the in-distribution (validation) accuracy and the out-of-distribution (test) accuracy of IRM across five optimizers.

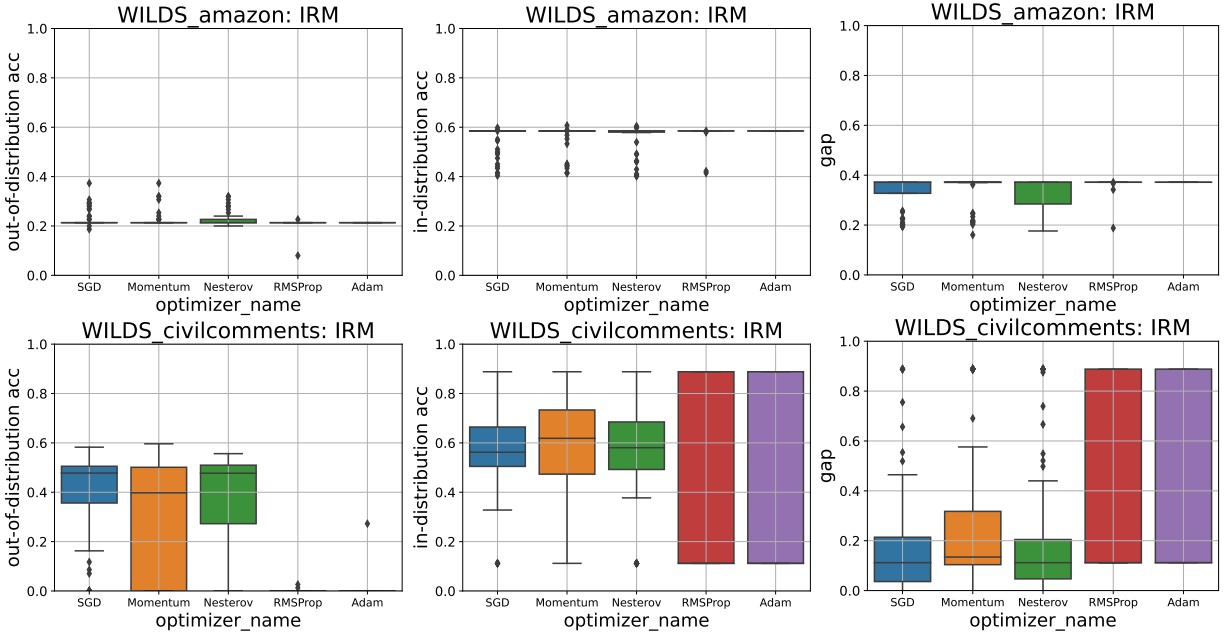

Figure 12. Filtered Results of Amazon-WILDS and CivilComments-WILDS: Comparison of the in-distribution (validation) accuracy and the out-of-distribution (test) accuracy of IRM across five optimizers.

## F.3 Full Results of Bin-Diagram Plots

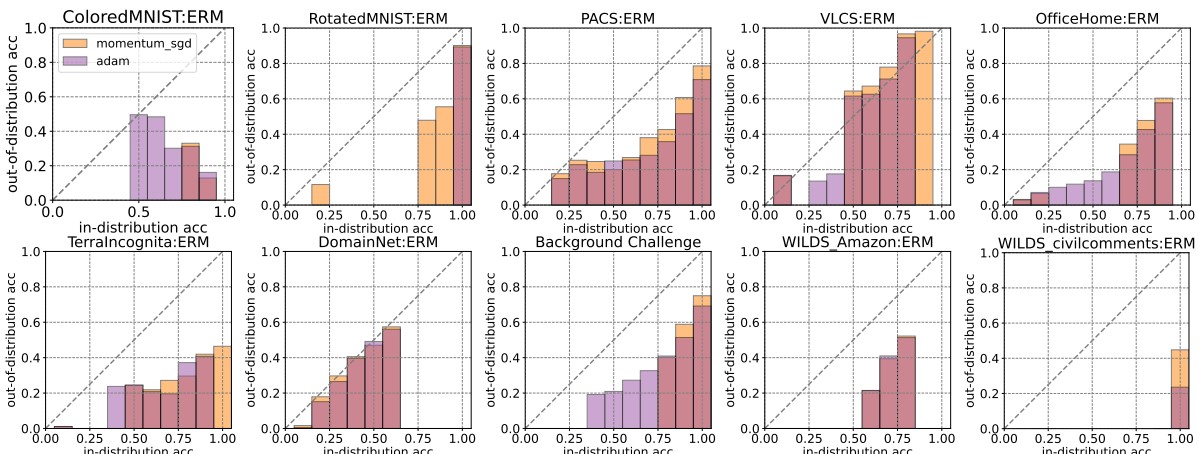

Figure 13. DomainBed, Backgrounds Challenge, Amazon-WILDS, and CivilComments-WILDS: Comparison of the in-distribution accuracy and the out-of-distribution accuracy of ERM between Momentum SGD and Adam. Since Adam showed better OOD performance than RMSProp, Adam is presented as a representative of adaptive methods. Momentum SGD shows competitive performance in OOD with Vanilla SGD and Nesterov Momentum and is a representative of non-adaptive methods.

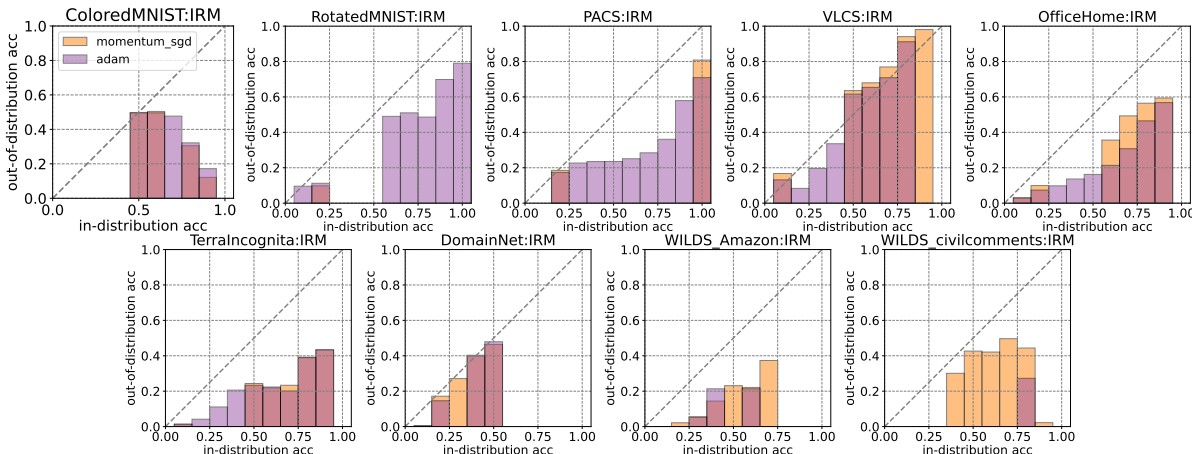

Figure 14. DomainBed, Amazon-WILDS, and CivilComments-WILDS: Comparison of the in-distribution accuracy and the out-of-distribution accuracy of IRM between Momentum SGD and Adam. Since Adam showed better OOD performance than RMSProp, Adam is presented as a representative of adaptive methods. Momentum SGD shows competitive performance in OOD with Vanilla SGD and Nesterov Momentum and is a representative of non-adaptive methods.

### F.4    Full Results of Scatter Plots

The results of the comparison of OOD accuracy and in-distribution accuracy shown in Section 4.3 are shown below for all benchmarks.

The box plots shown in Section F.2 and the Reliability Diagram Like Plot shown in Section F.3 are based on the data shown in the Scatter Plot shown below.

### F.4.1    ERM

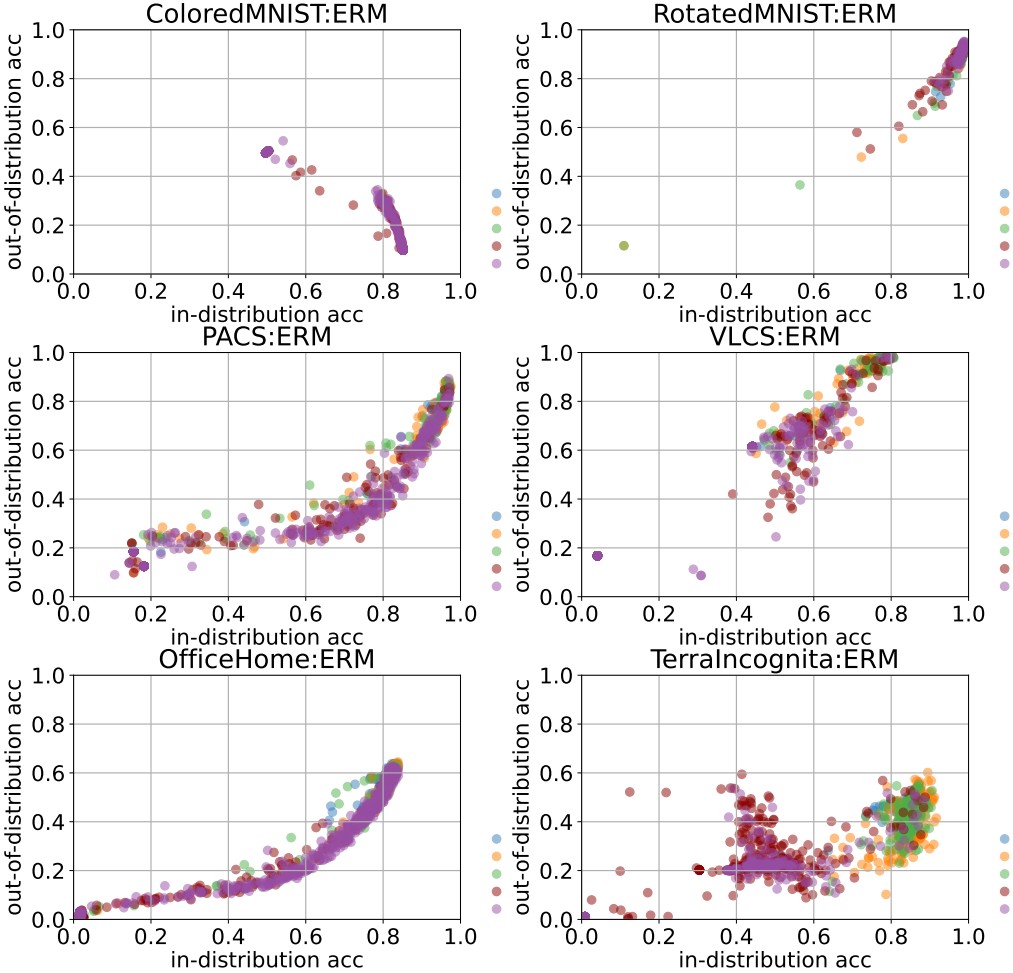

Figure 15. DomainBed (ColoredMNIST, RotatedMNIST, PACS, VLCS, OfficeHome and TerraIncognita): Comparison of the in-distribution accuracy and the out-of-distribution accuracy of ERM across optimizers. The legend circles on the right side of each figure show, in order, VanillaSGD, Momentum SGD, Nesterov Momentum, RMProp, and Adam. The difference in each data point indicates the difference in hyperparameter configuration.

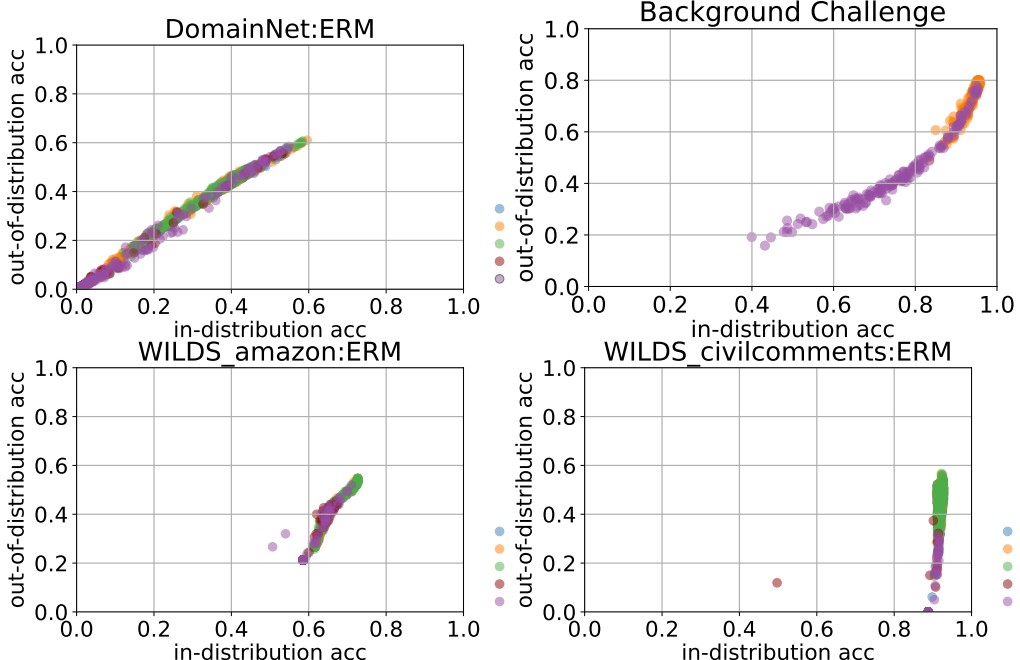

Figure 16. DomainBed (DomainNet), Backgrounds Challenge and WILDS: Comparison of the in-distribution accuracy and the out-of-distribution accuracy of ERM across optimizers. The legend circles on the right side of each figure show, in order, VanillaSGD, Momentum SGD, Nesterov Momentum, RMProp, and Adam. The difference in each data point indicates the difference in hyperparameter configuration.

### F.4.2   IRM

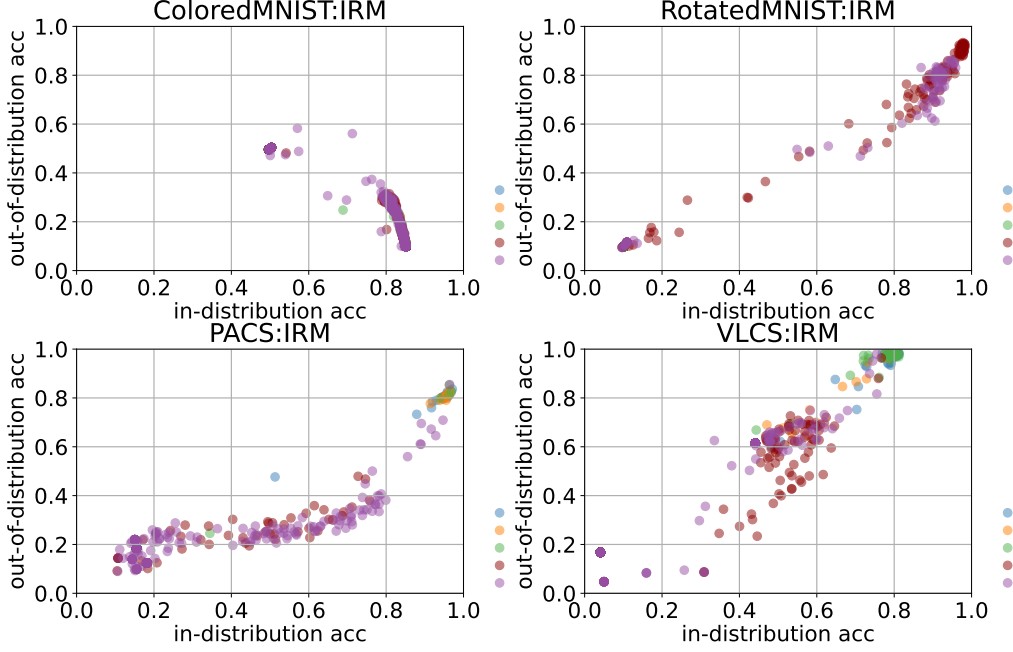

Figure 17. DomainBed (ColoredMNIST, RotatedMNIST, PACS and VLCS): Comparison of the in-distribution accuracy and the out-of-distribution accuracy of IRM across optimizers. The legend circles on the right side of each figure show, in order, VanillaSGD, Momentum SGD, Nesterov Momentum, RMProp, and Adam. The difference in each data point indicates the difference in hyperparameter configuration.

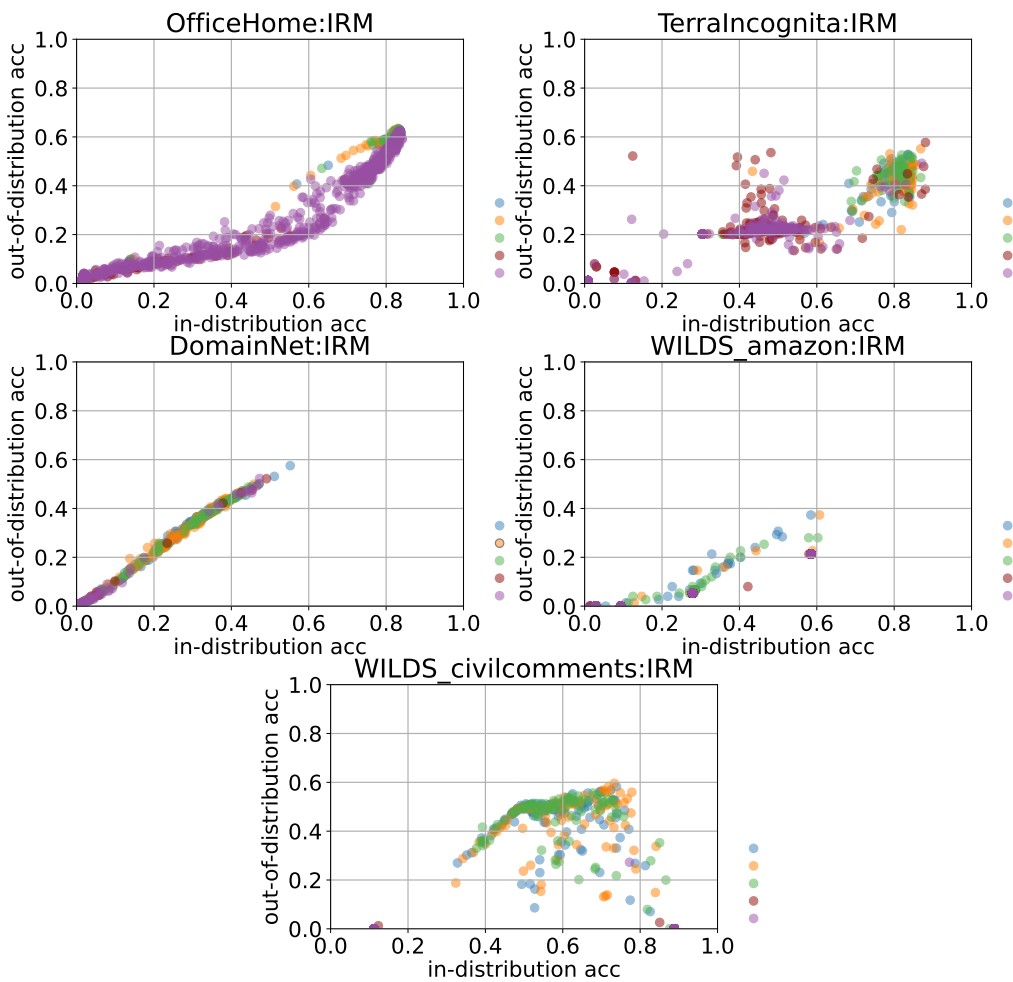

Figure 18. DomainBed (OfficeHome, TerraIncognita, DomainNet) and WILDS: Comparison of the in-distribution accuracy and the out-of-distribution accuracy of IRM across optimizers. The legend circles on the right side of each figure show, in order, VanillaSGD, Momentum SGD, Nesterov Momentum, RMProp, and Adam. The difference in each data point indicates the difference in hyperparameter configuration.

# G  Ablation Study

In addition to the experimental results presented in the main paper, we provide a more in-depth analysis, which is shown in this chapter.

First of all, We show that the pattern of linear correlation of accuracy claimed by (Miller et al., 2021) does not necessarily occur even when the correlation patterns we observed, such as diminishing returns, are probit transformed. The results of the probit transform of the scatter plots shown in Appendix F.4 are shown, following the method of (Miller et al., 2021). This may be due to the fact that our experimental setup uses a larger number of data sets and takes IRM and other factors into account.

In the second section, we present results comparing the performance improvement of OOD with a larger trial budget in the search for hyperparameters by Bayesian optimization.

Thirdly, We investigate why Adam shows better OOD performance than SGD in the negatively correlated case seen in Figure 6 for ColoredMNIST by considering the learning curve.

Finally, we examine the effectiveness of the early stopping for each optimizer to study if adaptive optimizers overfit because of their speed of convergence.

## G.1  Probit Transformed Scatter Plot

As shown in Section 4.3, the work of (Miller et al., 2021) compares the accuracy of OOD with the accuracy of in-distribution by showing a plot on a probit scale. Their comparison shows that the correlation of accuracy is linear.

In this section, we convert the scatter plots identified in Section F.4 to probit scale and confirm that they do not necessarily show a linear correlation (Figure 19, 21, 22, 23, 23, and 24).

### G.1.1  ERM

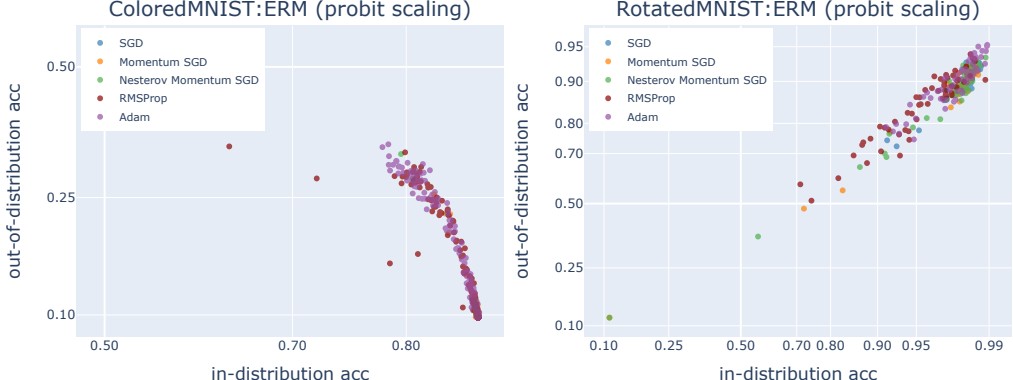

Figure 19. Probit Scale / DomainBed (ColoredMNIST and RotatedMNIST): Comparison of the in-distribution accuracy and the out-of-distribution accuracy of ERM across optimizers. The difference in each data point indicates the difference in hyperparameter configuration.

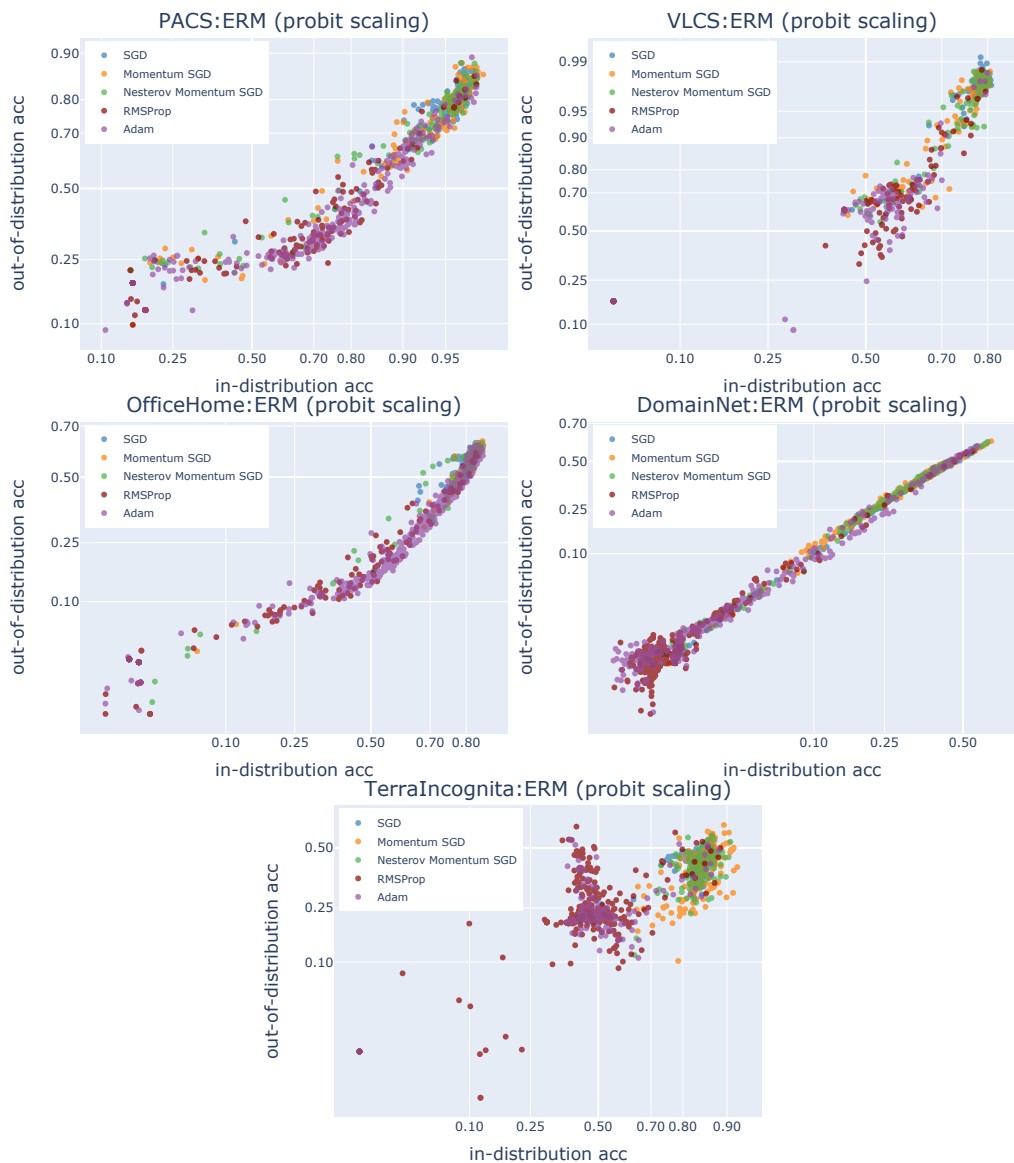

Figure 20. Probit Scale / DomainBed (PACS, VLCS, OfficeHome, DomainNet and TerraIncognita): Comparison of the in-distribution accuracy and the out-of-distribution accuracy of ERM across optimizers. The difference in each data point indicates the difference in hyperparameter configuration.

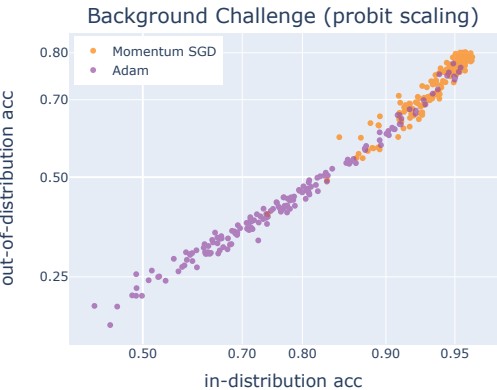

Figure 21. Probit Scale / Backgrounds Challenge: Comparison of the in-distribution accuracy and the out-of-distribution accuracy of ERM across optimizers. The difference in each data point indicates the difference in hyperparameter configuration.

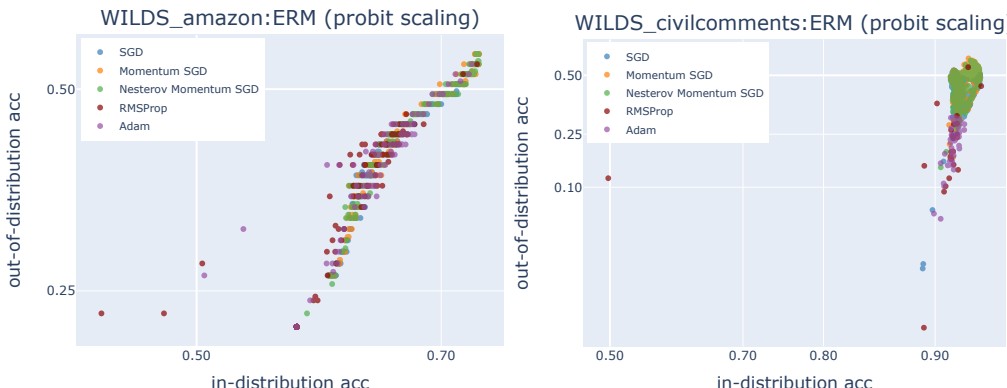

Figure 22. Probit Scale / WILDS: Comparison of the in-distribution accuracy and the out-of-distribution accuracy of ERM across optimizers. The difference in each data point indicates the difference in hyperparameter configuration.

### G.1.2 IRM

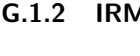

Figure 23. Probit Scale / DomainBed (ColoredMNIST, RotatedMNIST, PACS, VLCS, OfficeHome, DomainNed and TerraIncognita): Comparison of the in-distribution accuracy and the out-of-distribution accuracy of IRM across optimizers. The difference in each data point indicates the difference in hyperparameter configuration.

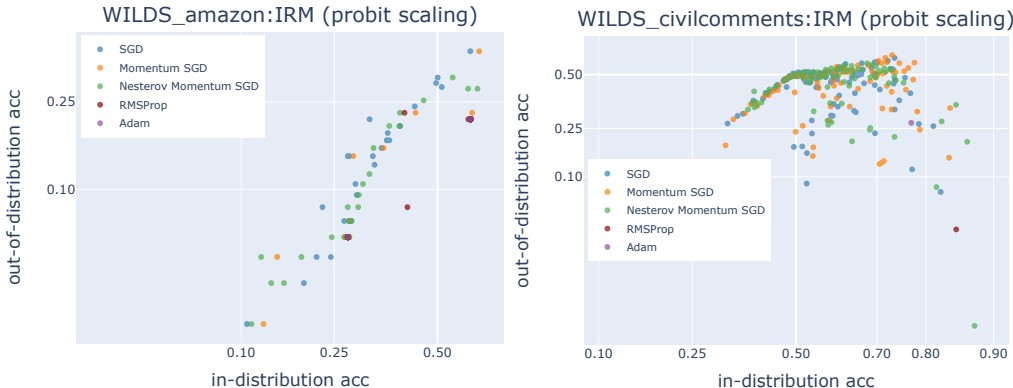

Figure 24. Probit Scale / WILDS: Comparison of the in-distribution accuracy and the out-of-distribution accuracy of IRM across optimizers. The difference in each data point indicates the difference in hyperparameter configuration.

## G.2 Model Performance Transition throught Hyperparameter Search

For practitioners, we show how much the trial budget for hyperparameter optimization affects OOD generalization.

### G.2.1 Hyperparameter Trial Budget vs OOD Accuracy

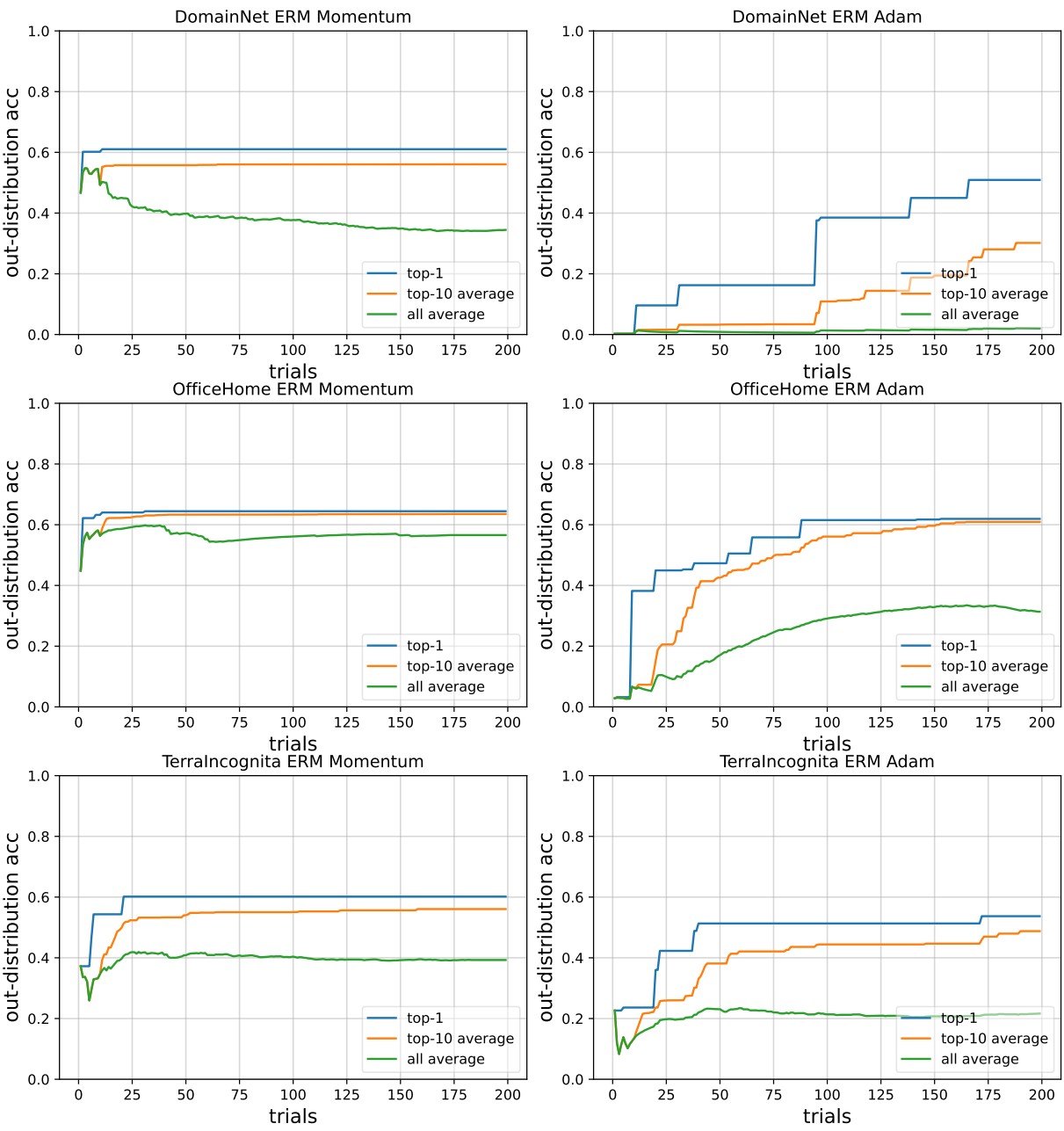

Figure 25. ERM DomainNet, OfficeHome and TerraIncognita / OOD accuracy when horizontal axis is the trial budget

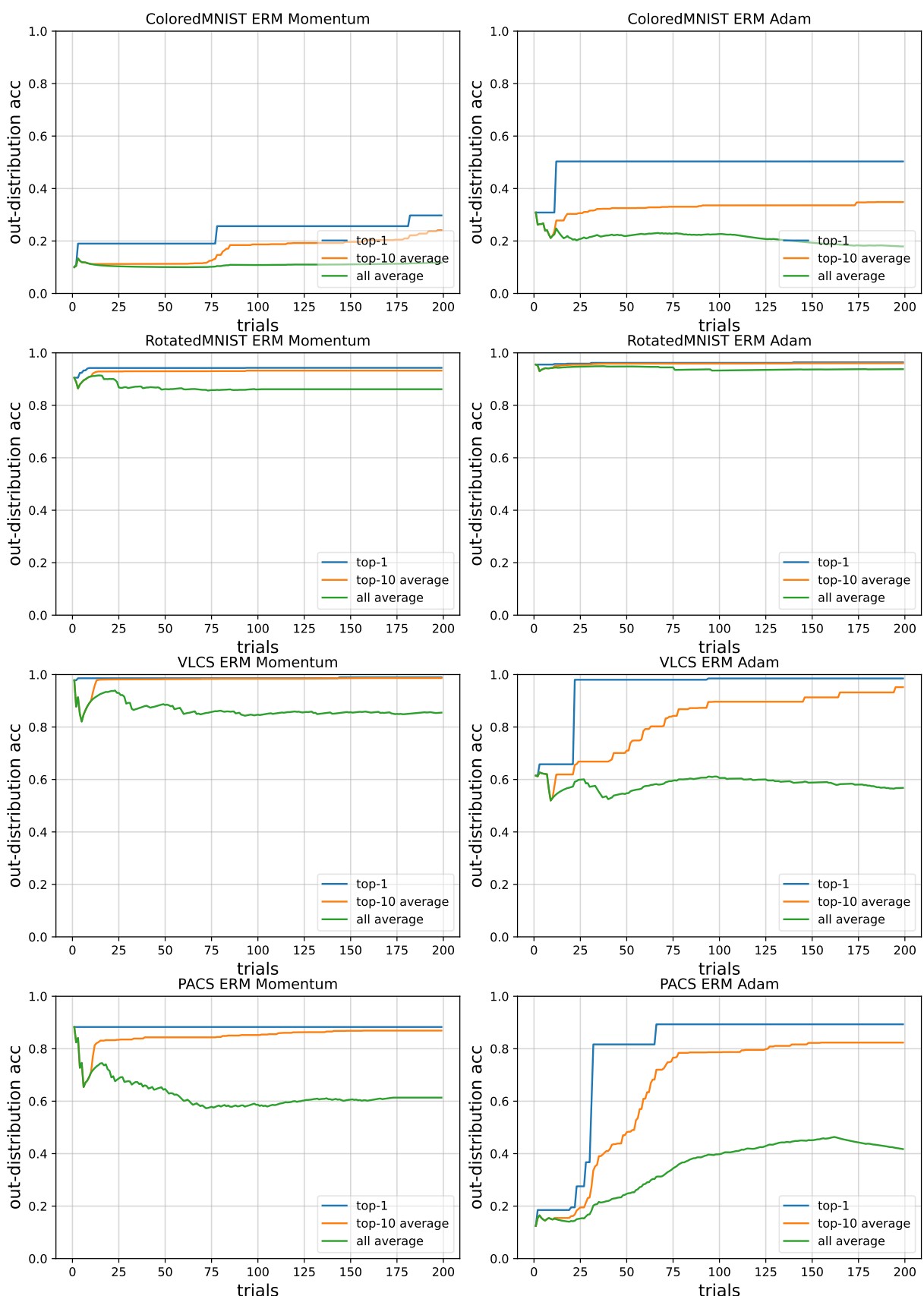

Figure 26. ERM ColoredMNIST, RotatedMNIST, VLCS and PACS / OOD accuracy when horizontal axis is the trial budget

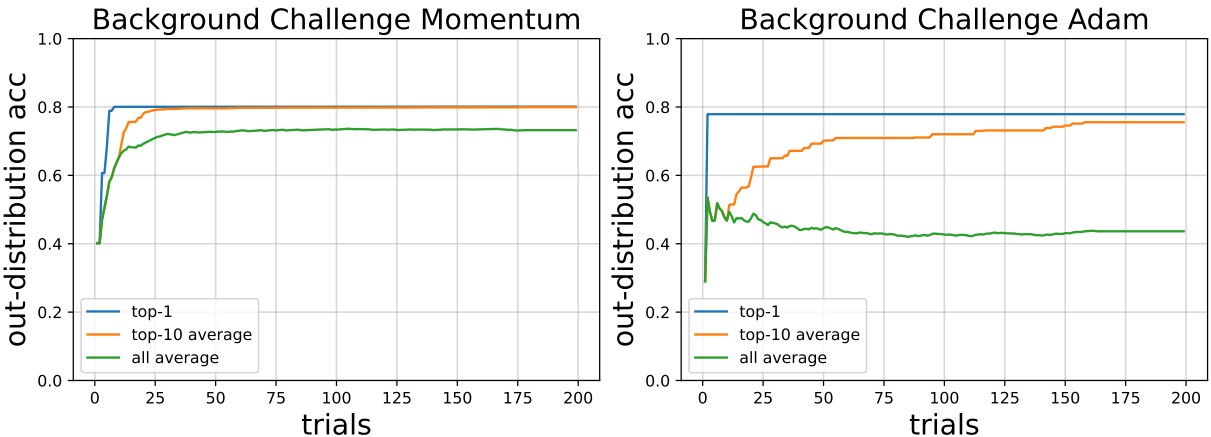

Figure 27. ERM Background Challenge / OOD accuracy when horizontal axis is the trial budget

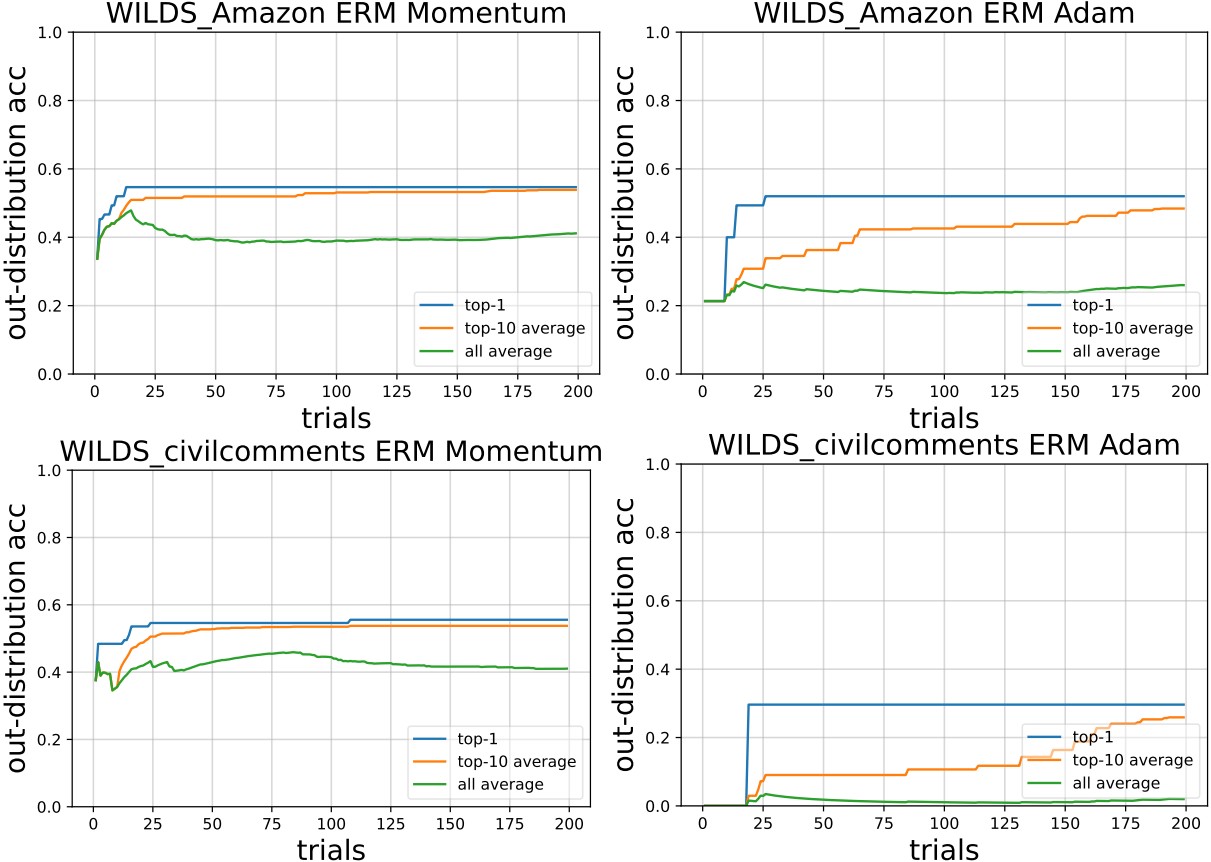

Figure 28. ERM WILDS Amazon and WILDS CivilComment / OOD accuracy when horizontal axis is the trial budget

### G.2.2 Hyperparameter Trial Budget vs OOD Error

To visualize the slight increase at the end of OOD accuracy more clearly, we visualized it in log scale as OOD error.

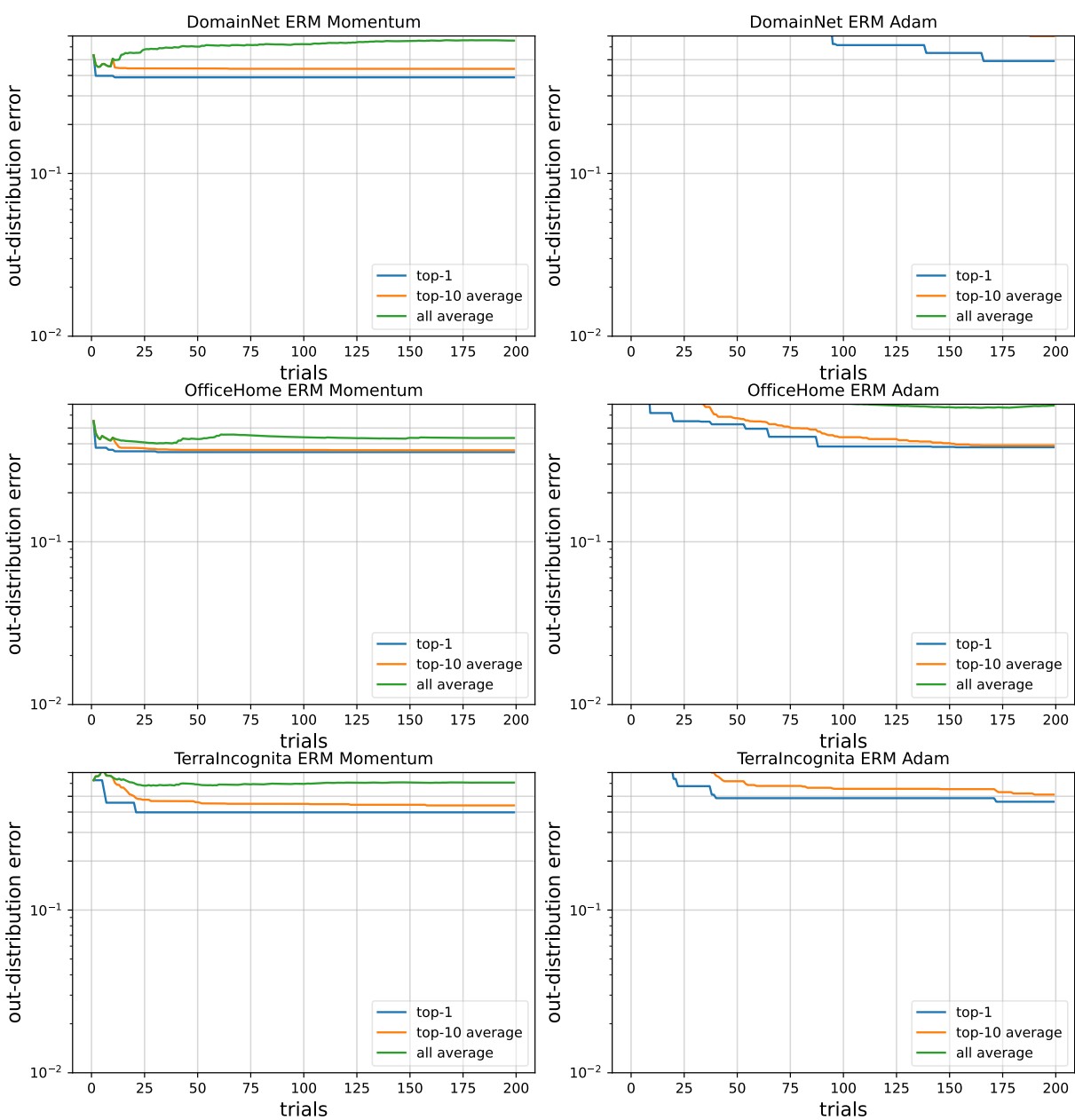

Figure 29. ERM DomainNet, OfficeHome and TerraIncognita / OOD error when horizontal axis is the trial budget

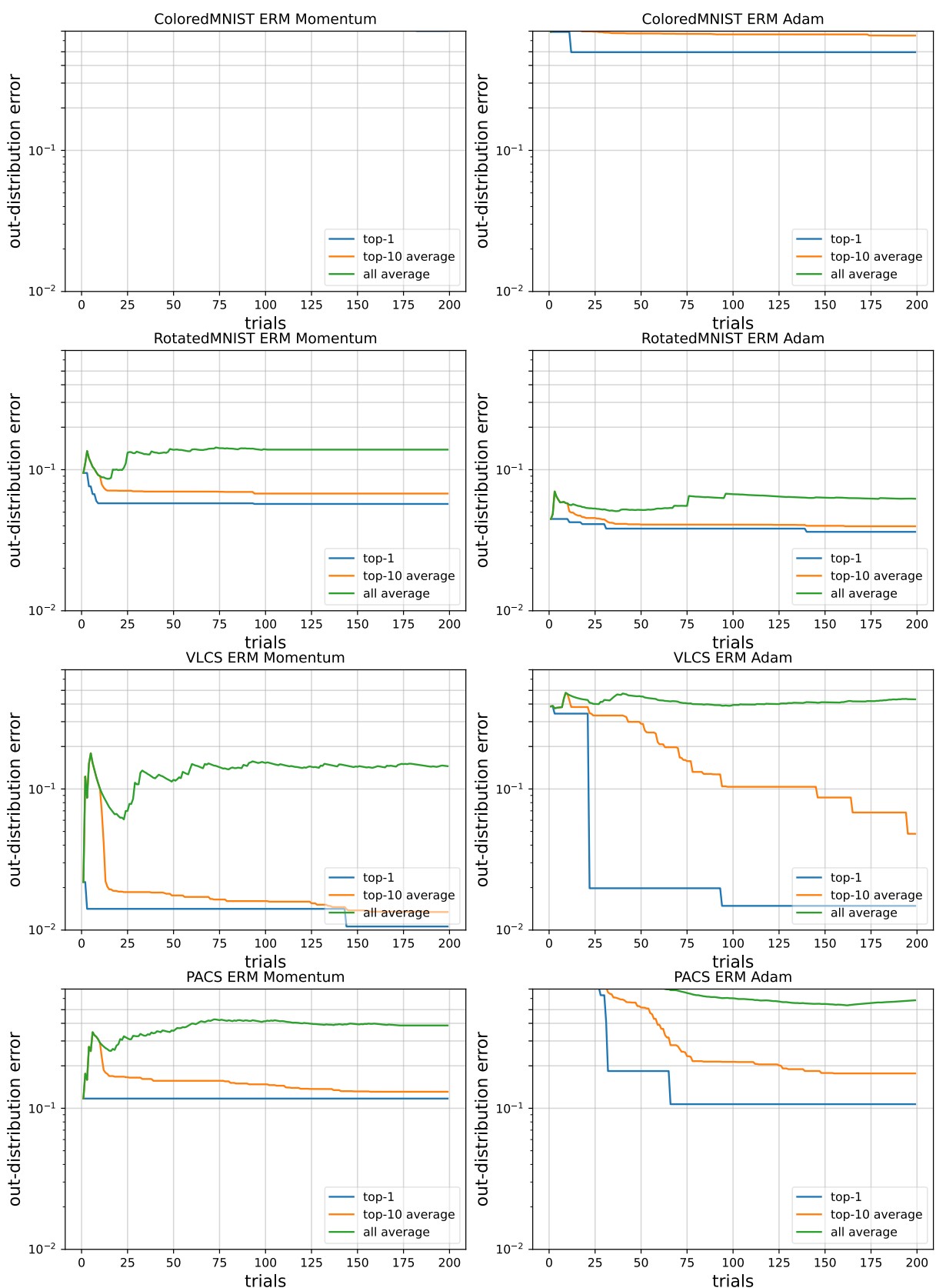

Figure 30. ERM ColoredMNIST, RotatedMNIST, VLCS and PACS / OOD error when horizontal axis is the trial budget

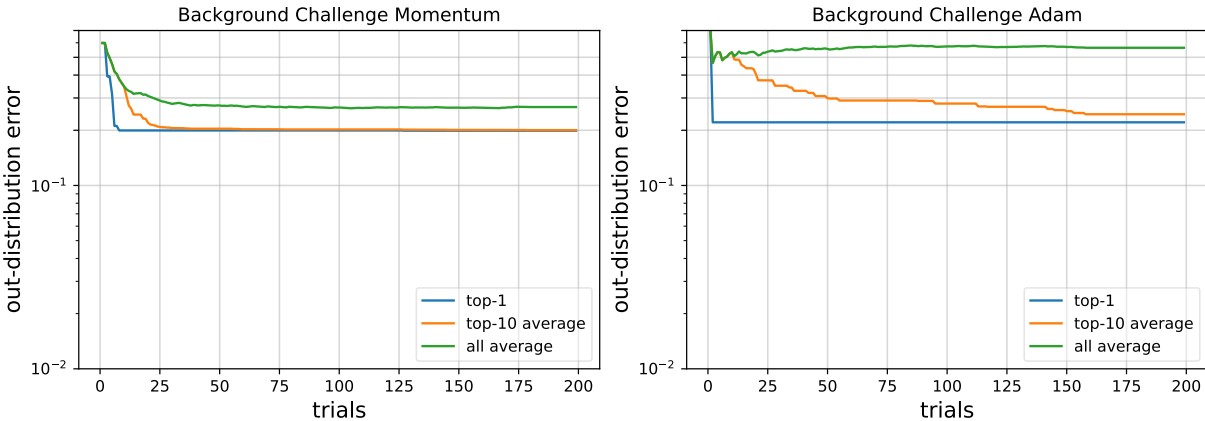

Figure 31. ERM Background Challenge / OOD accuracy when horizontal axis is the trial budget

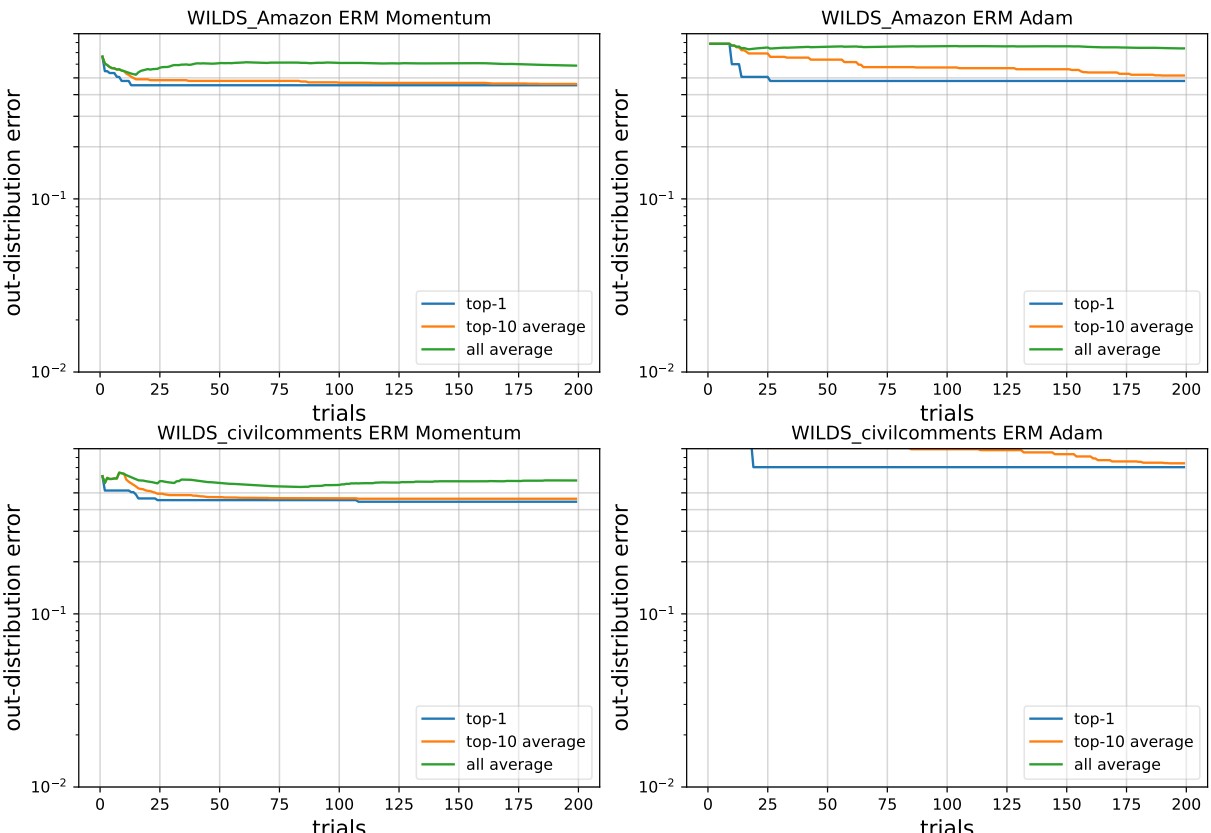

Figure 32. ERM WILDS Amazon and WILDS CivilComment / OOD accuracy when horizontal axis is the trial budget

### G.3 Learning Curve of ColoredMNIST

In Section 4.2, we conjecture that the better performance of Adam in Colored MNIST classification may come from overfitting to training data. To confirm this hypothesis, we plot the averaged training accuracy, the averaged validation accuracy, and the averaged test accuracy throughout the training. We pick the top-14 results in terms of test accuracy and use them for the plot. We show the result in Figure 33.

As is evident from Figure 33 (a) and Figure 33 (b), we observe that training accuracy increases while the validation accuracy keeps unchanged. This indicates that overfitting occurs in the training. However, Figure 33 (c) indicates that test accuracy gradually improves as the model is overfitting. On the other hand, SGD shows no such overfitting and OOD generalization improvement, as shown in Figure 34. Thus, we can empirically support our conjecture that Adam produces the better OOD generalization performance by overfitting training data.

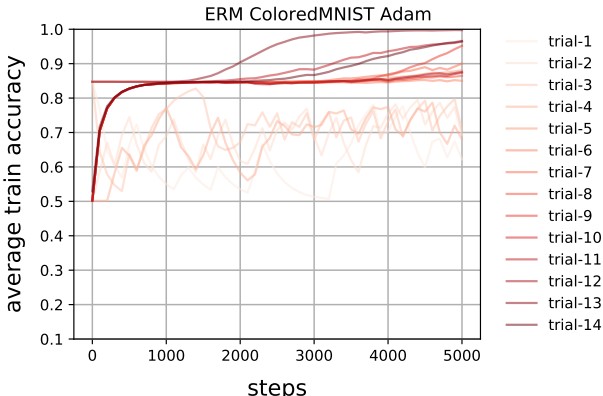

(a) ERM: Averaged training accuracy of Adam on Colored MNIST throughout the training.

(b) ERM: Averaged validation accuracy of Adam on Colored MNIST throughout the training.

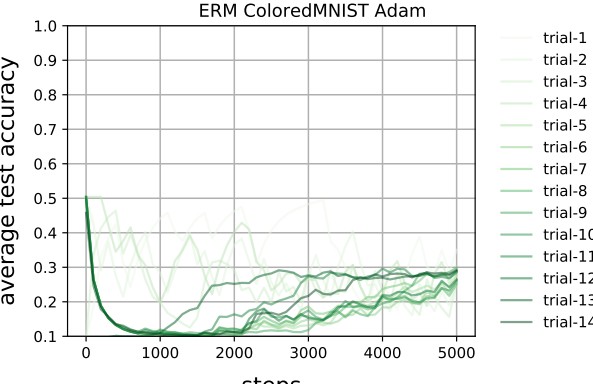

(c) ERM: Averaged test accuracy of Adam on Colored MNIST throughout the training.

Figure 33. Adam: Comparison of averaged training accuracy, averaged validation accuracy, and averaged test accuracy throughout the training. We plot these values of check points whose train accuracy is in the top-14.

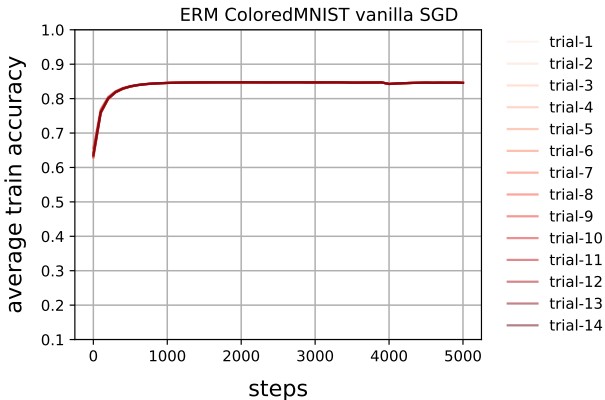

(a) ERM: Averaged training accuracy of SGD on Colored MNIST throughout the training.

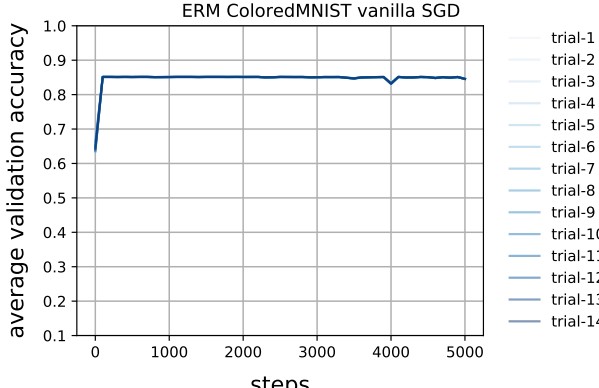

(b) ERM: Averaged validation accuracy of SGD on Colored MNIST throughout the training.

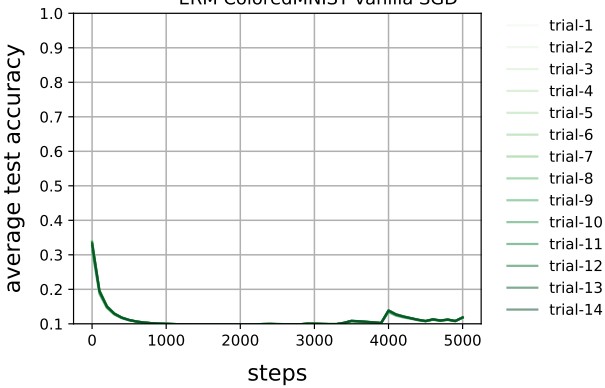

(c) ERM: Averaged test accuracy of SGD on Colored MNIST throughout the training.

Figure 34. SGD: Comparison of averaged training accuracy, averaged validation accuracy, and averaged test accuracy of SGD throughout the training. We plot these values of checkpoints whose train accuracy is in the top-14.

### G.4 Early Stopping

In Section 4.2, figures 1 shows that adaptive optimizers tend to overfit to training domain. A possible reason is that the training speed of adaptive methods is faster than non-adaptive ones. That is, in the same steps budget, adaptive optimizers converge faster in effect. To validate if this is the case, we investigate whether early stopping improves the OOD generalization of adaptive optimizers.

In particular, we compute the difference between averaged accuracy at early stopping and at last epoch for test accuracy and validation accuracy, respectively:

$$Acc_{\text{diff}} = \frac{1}{N_{\text{es}}} \sum_{i=1}^{N_{\text{es}}} Acc_{\text{es-i}} - \frac{1}{N_{le}} \sum_{i=1}^{N_{\text{le}}} Acc_{\text{le-i}} \tag{18}$$

where $Acc_{\text{diff}}$ represents the difference in accuracy between early stopping and the last epoch, $N_{\text{es}}$ is the number of trials at early stopping, $N_{\text{le}}$ is the number of trials at the last epoch, $Acc_{\text{es-i}}$ denotes the accuracy at the early stopping for the $i-th$ trial, $Acc_{\text{le-i}}$ denotes the accuracy at the last epoch for the $i-th$ trial.

If the average accuracy at early stopping is larger, the difference is positive and vice versa.

Figures 35 to 40 are the results of this comparison for each dataset and algorithm. The y-axis is the difference of out-of-distribution (test) accuracy and the x-axis is the difference of in-distribution (validation) accuracy. The color indicates the epoch of early stopping. The darker color indicates that early stopping is conducted in relatively earlier epochs.

For PACS, both differences are positive and lighter colors are concentrated at points of small difference of the validation accuracy, as indicated in Figures 35 and 36. Therefore, we can conclude that when the validation accuracy deteriorates by further training, the test accuracy also gets worse. However, the effect of further training is less evident for Adam. In other words, early stopping does not influence Adam so much but keeps test accuracy from decreasing for SGD.

The result of VLCS shows a similar pattern as PACS (Figures 37 and 38). If anything, Further training after early stopping makes adaptive optimizes be likely to result in better test accuracy than SGD, though they have a large variance. With respect to SGD, additional training degrades the test accuracy as much as it degrades validation accuracy.

Office-Home shows similar results as PACS, as presented in Figures 39 and 40.

In summary, we find that early stopping does not influence adaptive optimizers. Therefore, we can conclude that adaptive optimizer overfits not because it trains faster than non-adaptive methods.

The fact that early stopping does not affect the adaptive optimizer means that adjusting the number of epochs instead of a fixed number of epochs will not change the result. We have followed previous studies and experimented with a fixed number of epochs in the present study, and our results suggest that this does not have a serious impact on the comparison of adaptive and non-adaptive optimizers. Thus, we can see that using a fixed epoch number does not undermine the validity of our optimizer comparison experiment in this sense.

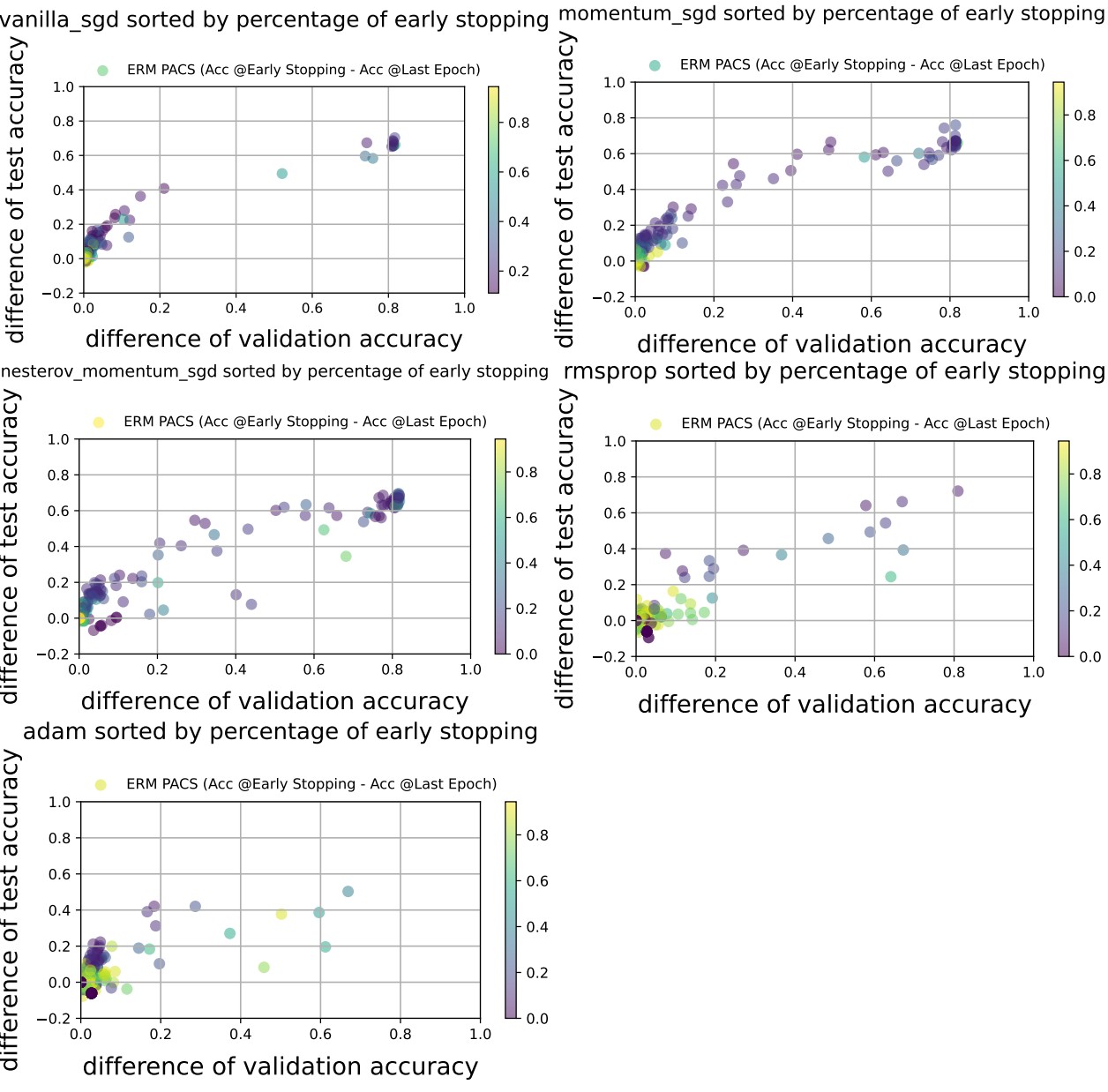

Figure 35. ERM/PACS: Difference between accuracy at early stopping and at last epoch. The y-axis is the difference of out-of-distribution (test) accuracy and the x-axis is the difference of in-distribution (validation) accuracy. The color indicates the epoch of early stopping. The darker color indicates that early stopping is conducted in relatively earlier epochs.

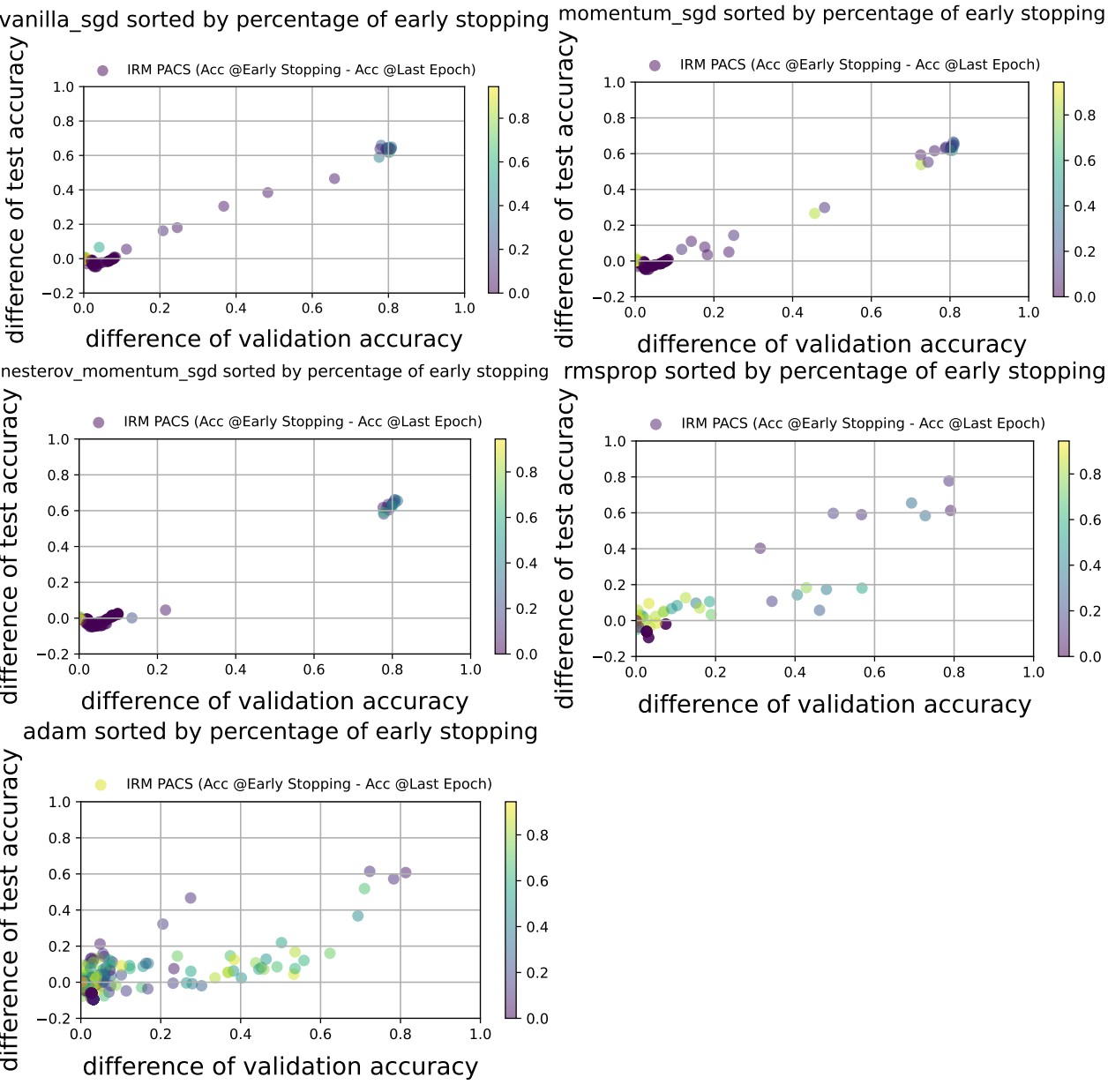

Figure 36. IRM/PACS: Difference between accuracy at early stopping and at last epoch. The y-axis is the difference of out-of-distribution (test) accuracy and the x-axis is the difference of in-distribution (validation) accuracy. The color indicates the epoch of early stopping. The darker color indicates that early stopping is conducted in relatively earlier epochs.

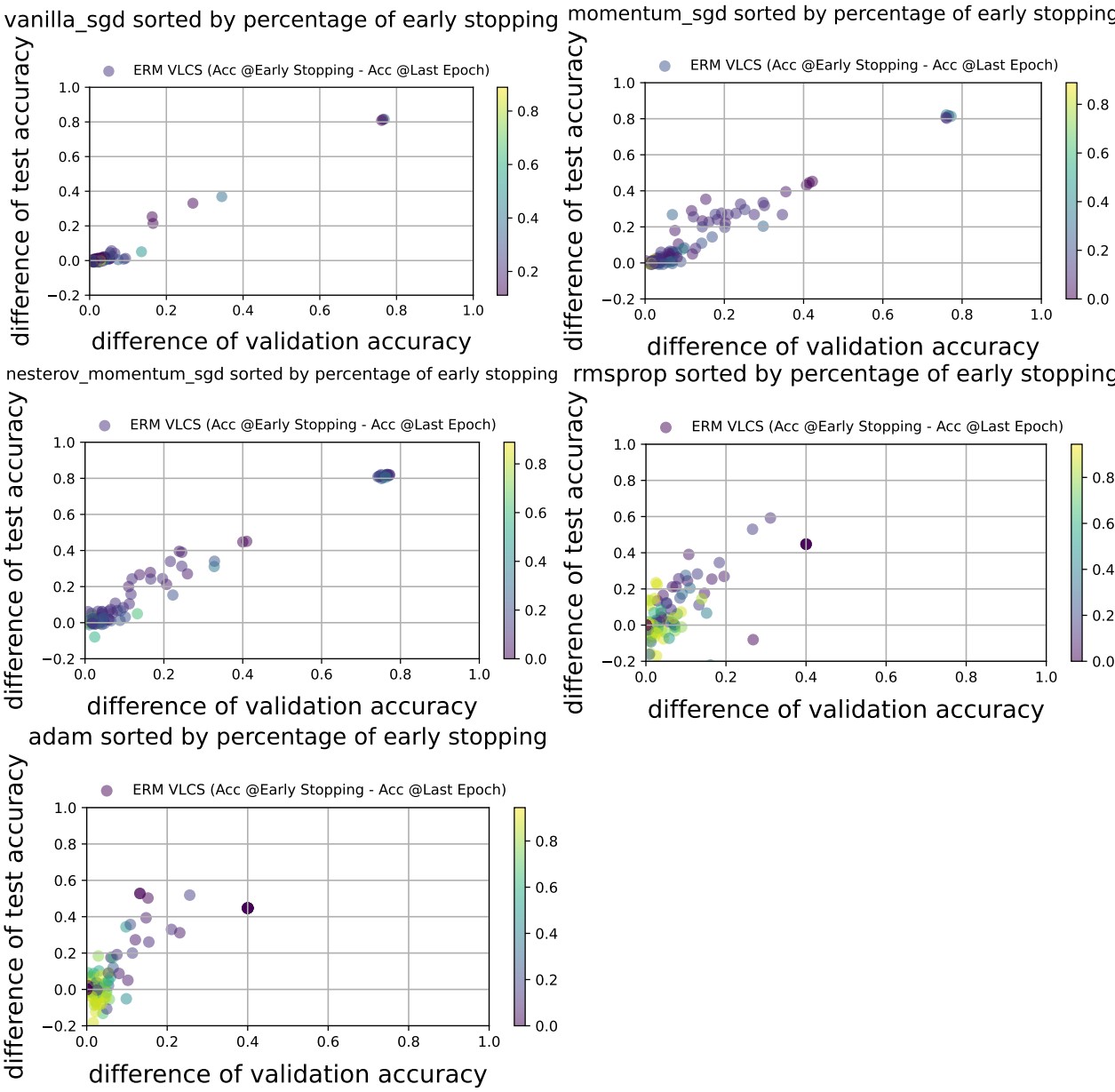

Figure 37. ERM/VLCS: Difference between accuracy at early stopping and at last epoch. The y-axis is the difference of out-of-distribution (test) accuracy and the x-axis is the difference of in-distribution (validation) accuracy. The color indicates the epoch of early stopping. The darker color indicates that early stopping is conducted in relatively earlier epochs.

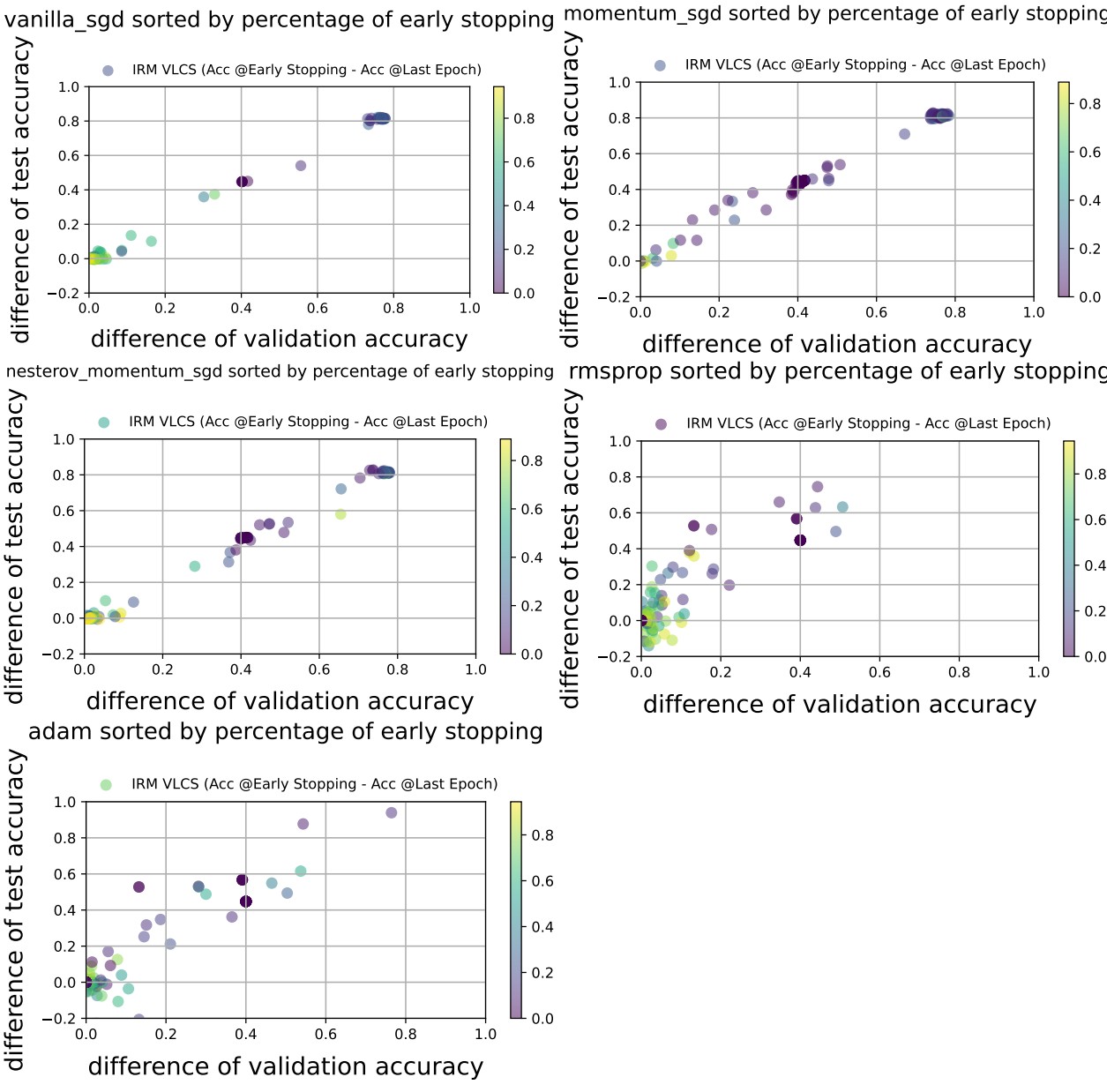

Figure 38. IRM/VLCS: Difference between accuracy at early stopping and at last epoch. The y-axis is the difference of out-of-distribution (test) accuracy and the x-axis is the difference of in-distribution (validation) accuracy. The color indicates the epoch of early stopping. The darker color indicates that early stopping is conducted in relatively earlier epochs.

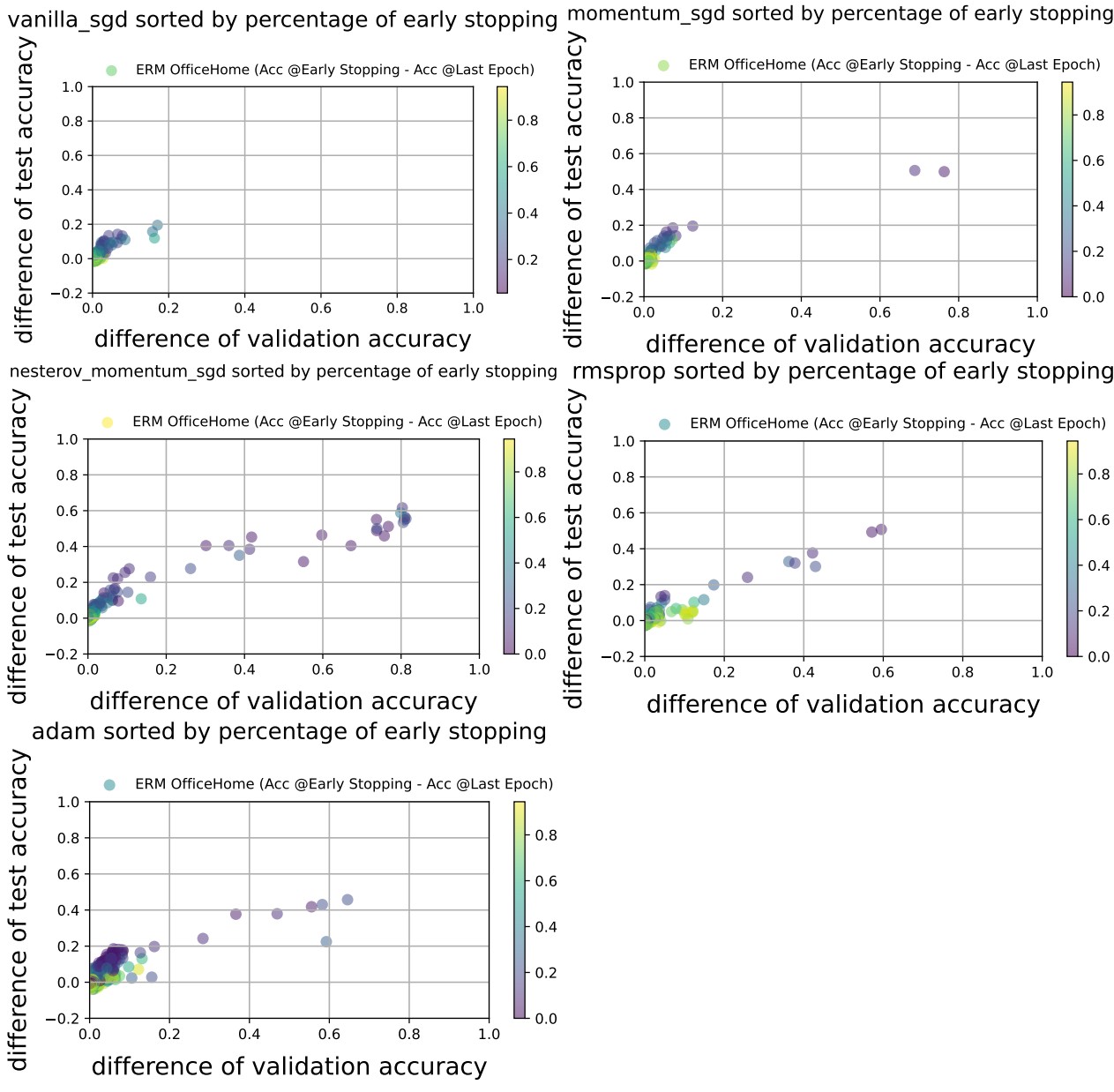

Figure 39. ERM/Office-Home: Difference between accuracy at early stopping and at last epoch. The y-axis is the difference of out-of-distribution (test) accuracy and the x-axis is the difference of in-distribution (validation) accuracy. The color indicates the epoch of early stopping. The darker color indicates that early stopping is conducted in relatively earlier epochs.

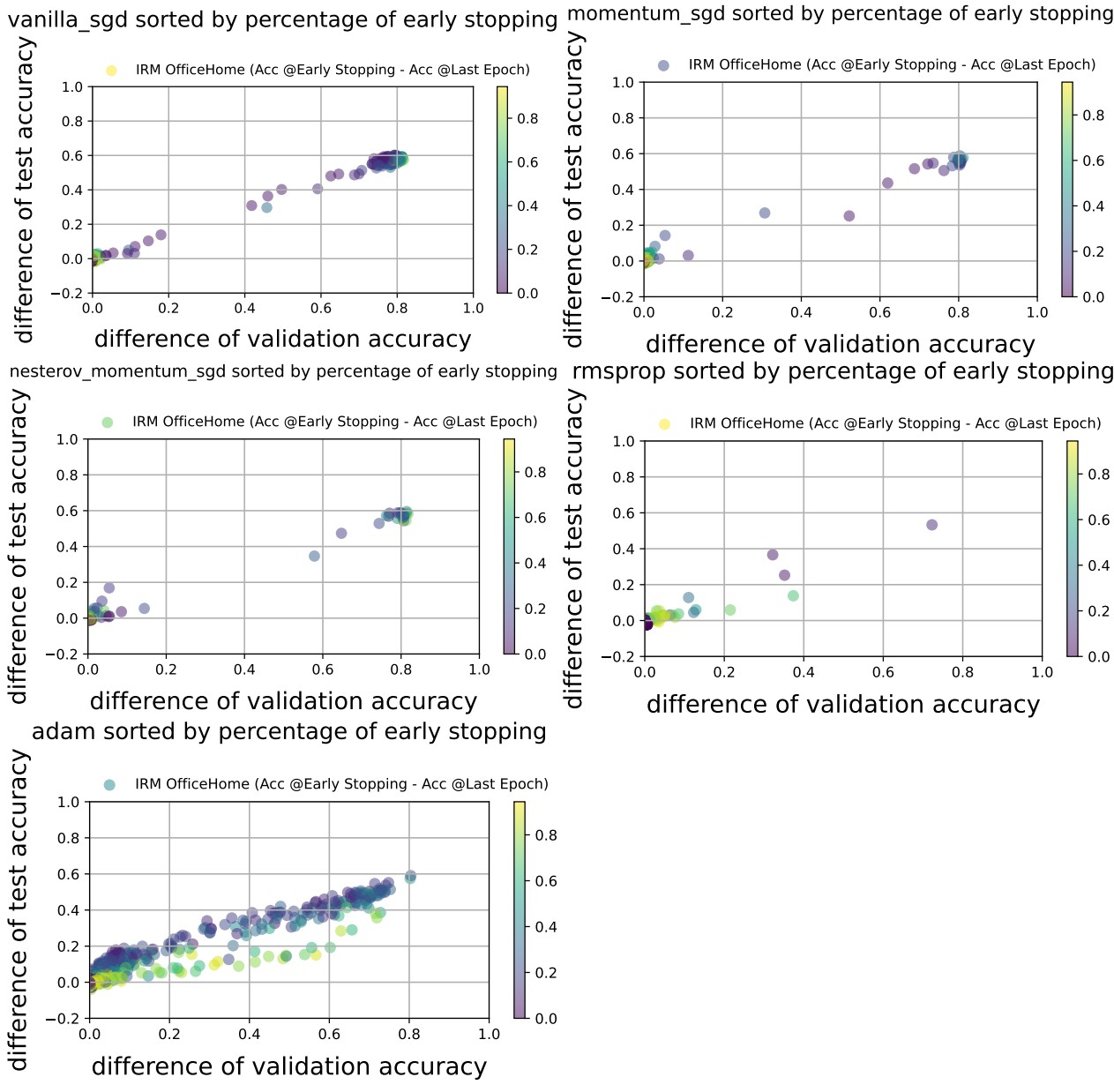

Figure 40. IRM/Office-Home: Difference between accuracy at early stopping and at last epoch. The y-axis is the difference of out-of-distribution (test) accuracy and the x-axis is the difference of in-distribution (validation) accuracy. The color indicates the epoch of early stopping. The darker color indicates that early stopping is conducted in relatively earlier epochs.

However, we find different results from Colored MNIST. As shown in Figures 41 and 42, we can observe that the difference of validation accuracy is positive and that of test accuracy is negative. That is further training after early stopping decreases validation accuracy but increases test accuracy on Colored MNIST, while there is no difference for SGD. This is consistent with the result of Appendix G.3.

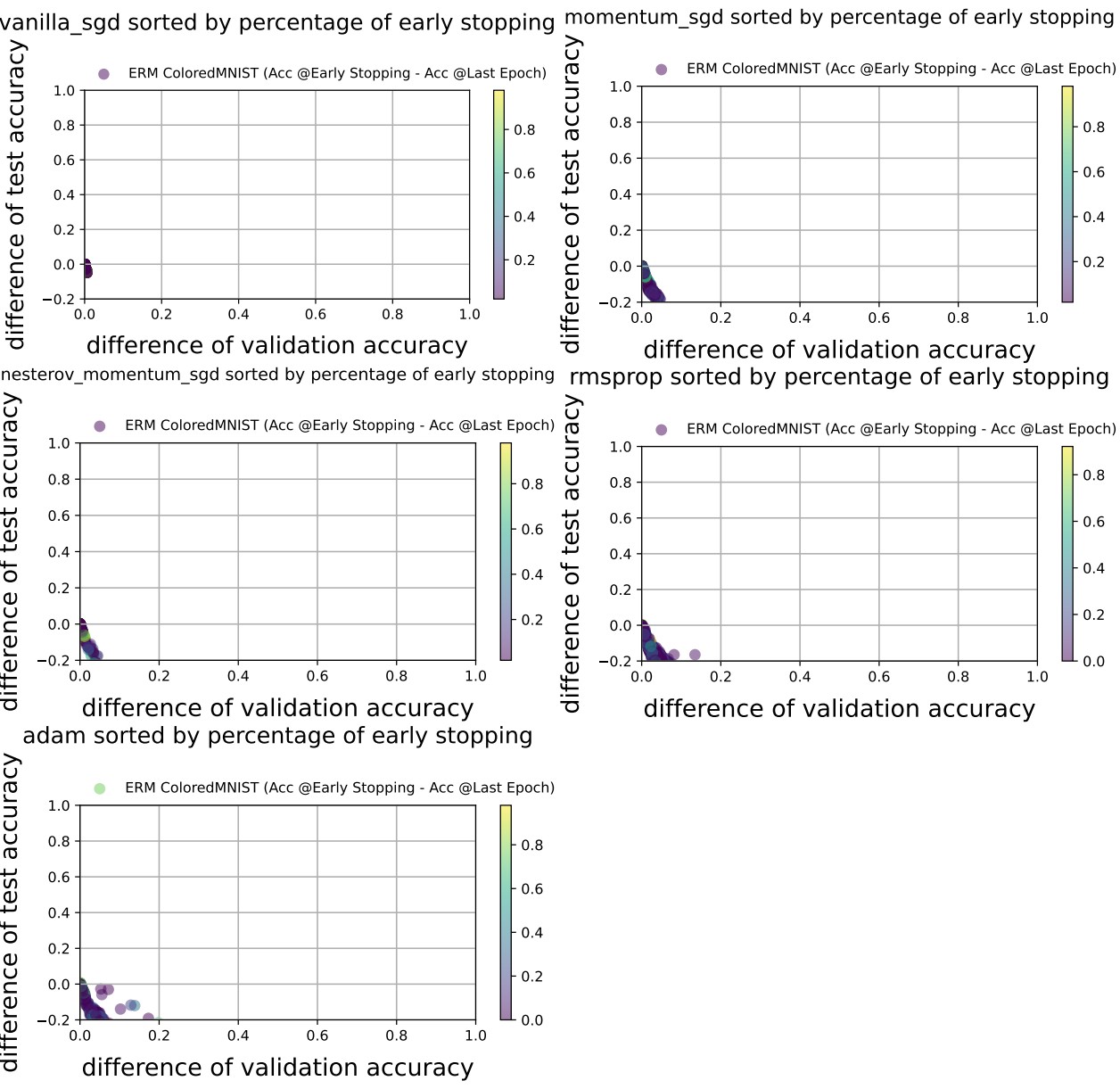

Figure 41. ERM/Colored MNIST: Difference between accuracy at early stopping and at last epoch. The y-axis is the difference of out-of-distribution (test) accuracy and the x-axis is the difference of in-distribution (validation) accuracy. The color indicates the epoch of early stopping. The darker color indicates that early stopping is conducted in relatively earlier epochs.

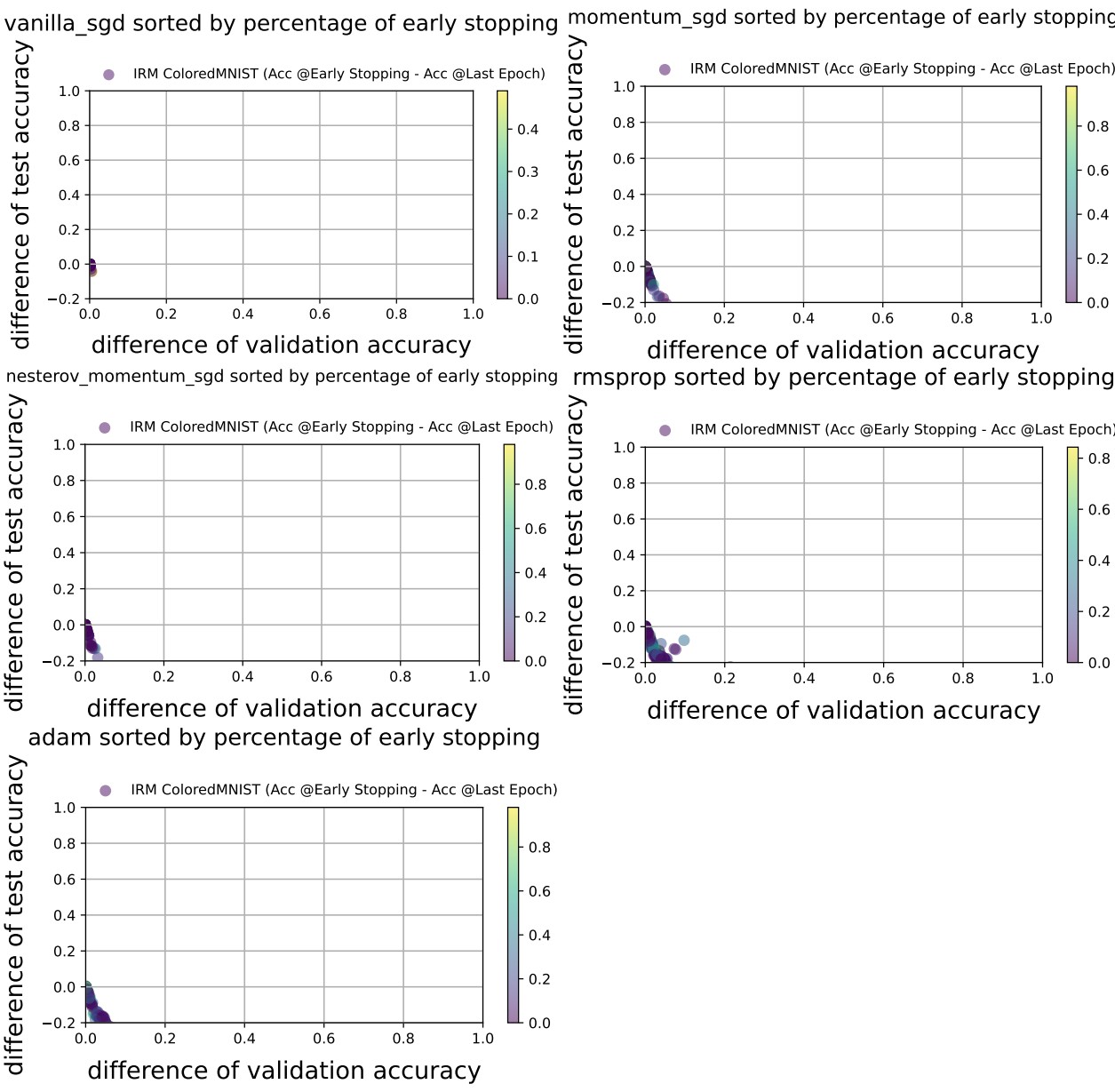

Figure 42. IRM/Colored MNIST: Difference between accuracy at early stopping and at last epoch. The y-axis is the difference of out-of-distribution (test) accuracy and x-axis is the difference of in-distribution (validation) accuracy. The color indicates the epoch of early stopping. The darker color indicates that early stopping is conducted in relatively earlier epochs.

# H   Soundness Check of Our Experiments

## H.1   Histgram of Hyperparameters

We have shown the histgram of the hyperparameters to be used for training just for reference. In particular, we display a result for learning rate of Momentum SGD and Adam for the each dataset. We observe that we could sample hyperparameters from a reasonably wide range.

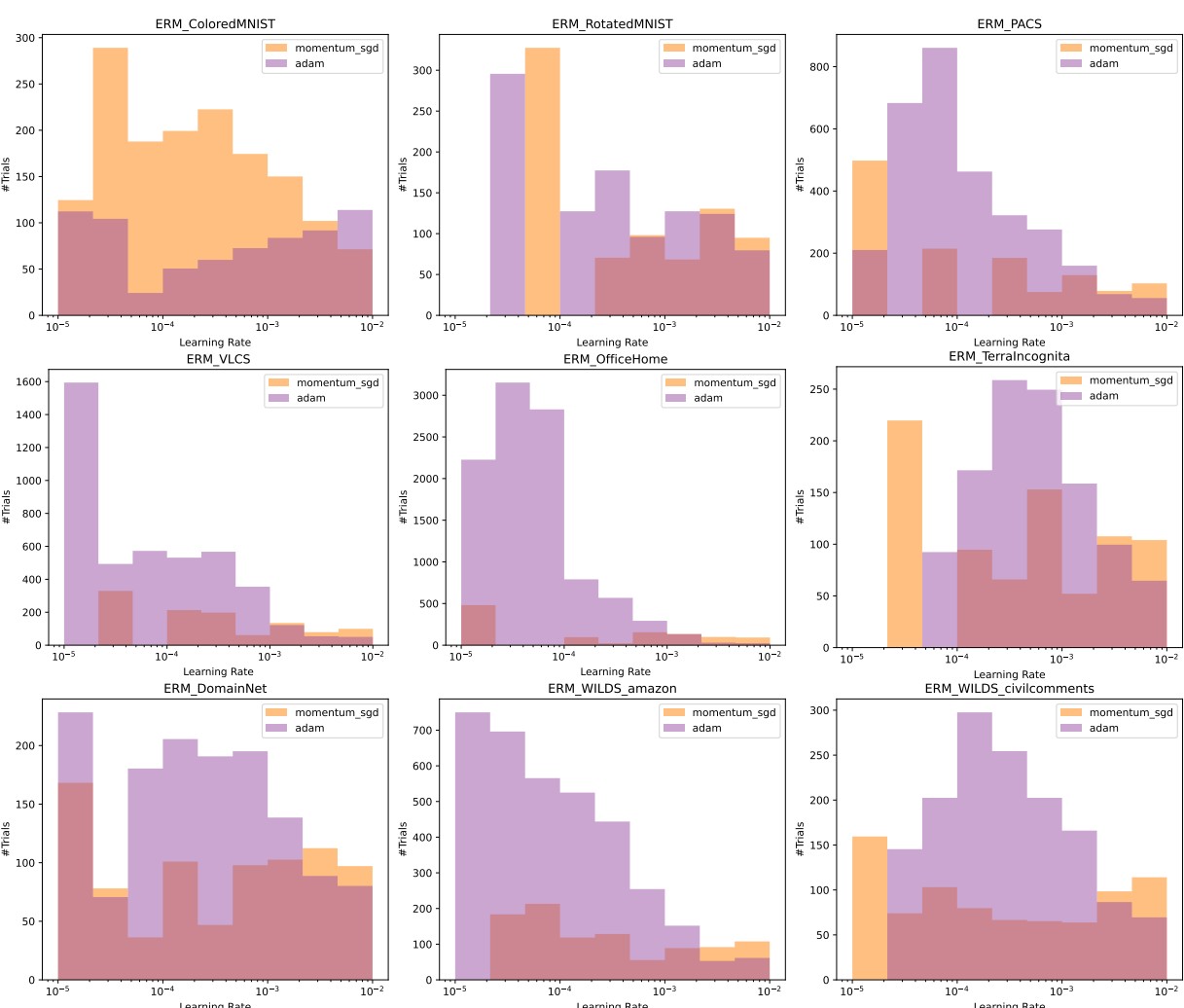

Figure 43. Histogram of explored hyperparameters (learning rate) in training of DomainBed, WILDS dataset in ERM. Momentum SGD and Adam results are included. Although uniform distribution is used as the prior distribution for hyperparameter optimization, the histogram results do not match the uniform distribution because Bayesian optimization is used.

## H.2   Hyperparameters and OOD Accuracy Box-Plot

The previous section provided information on the hyperparameter search range. In this section, we share the results of the out-of-distribution performance for a specific hyperparameter range as a box plot with the hyperparameters separating the bin as shown in Figure 44, 45 and 46.

From these results, we observe that lr, which achieves high out-of-distribution accuracy, is within the search range of learning rate and thus has sufficient range to perform the search.

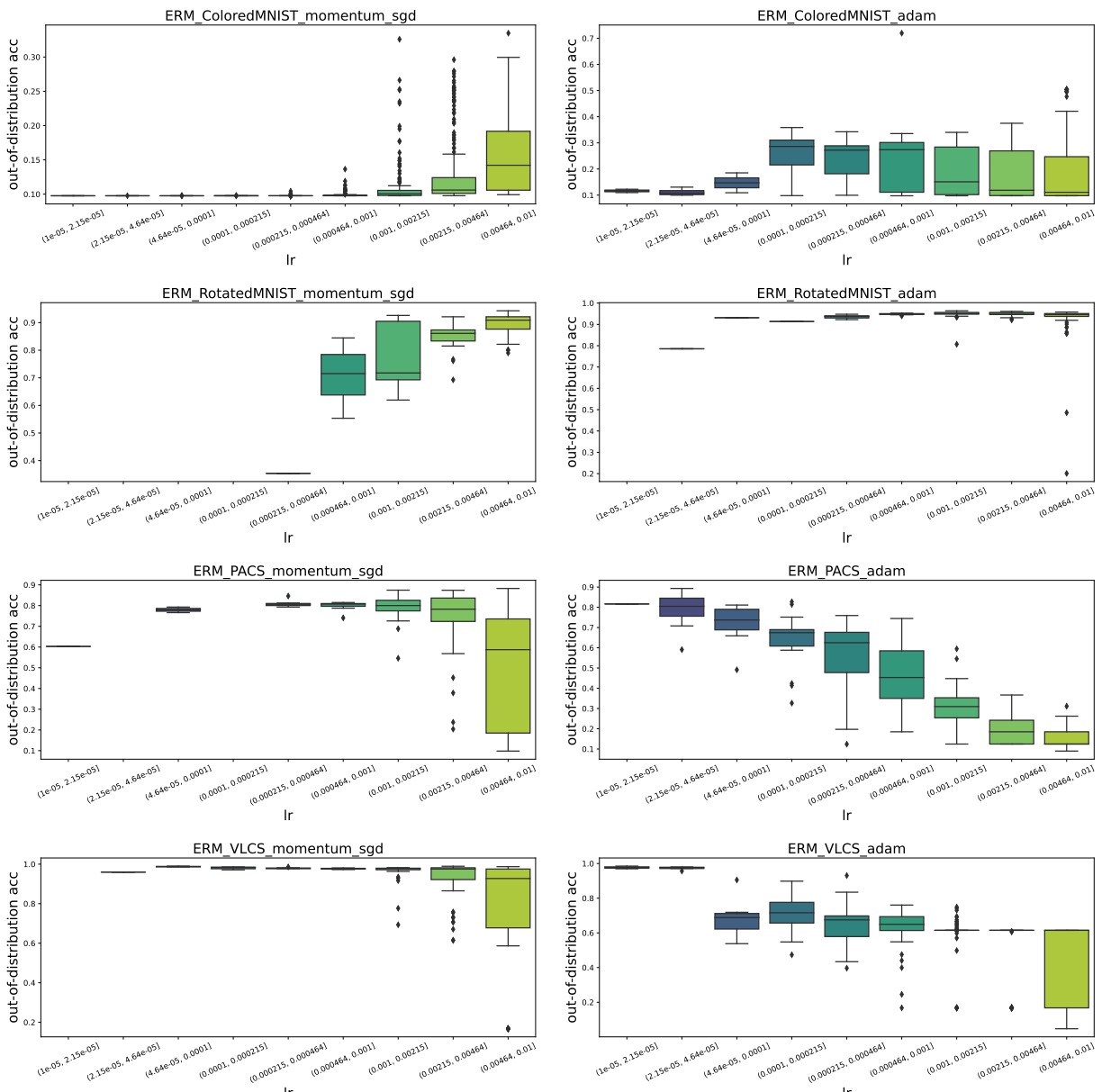

Figure 44. Box-Plot of Out-of-Distribution Accuracy per Log-Scale of Learning Rate: ColoredMNIST, RotatedMNIST, PACS and VLCS

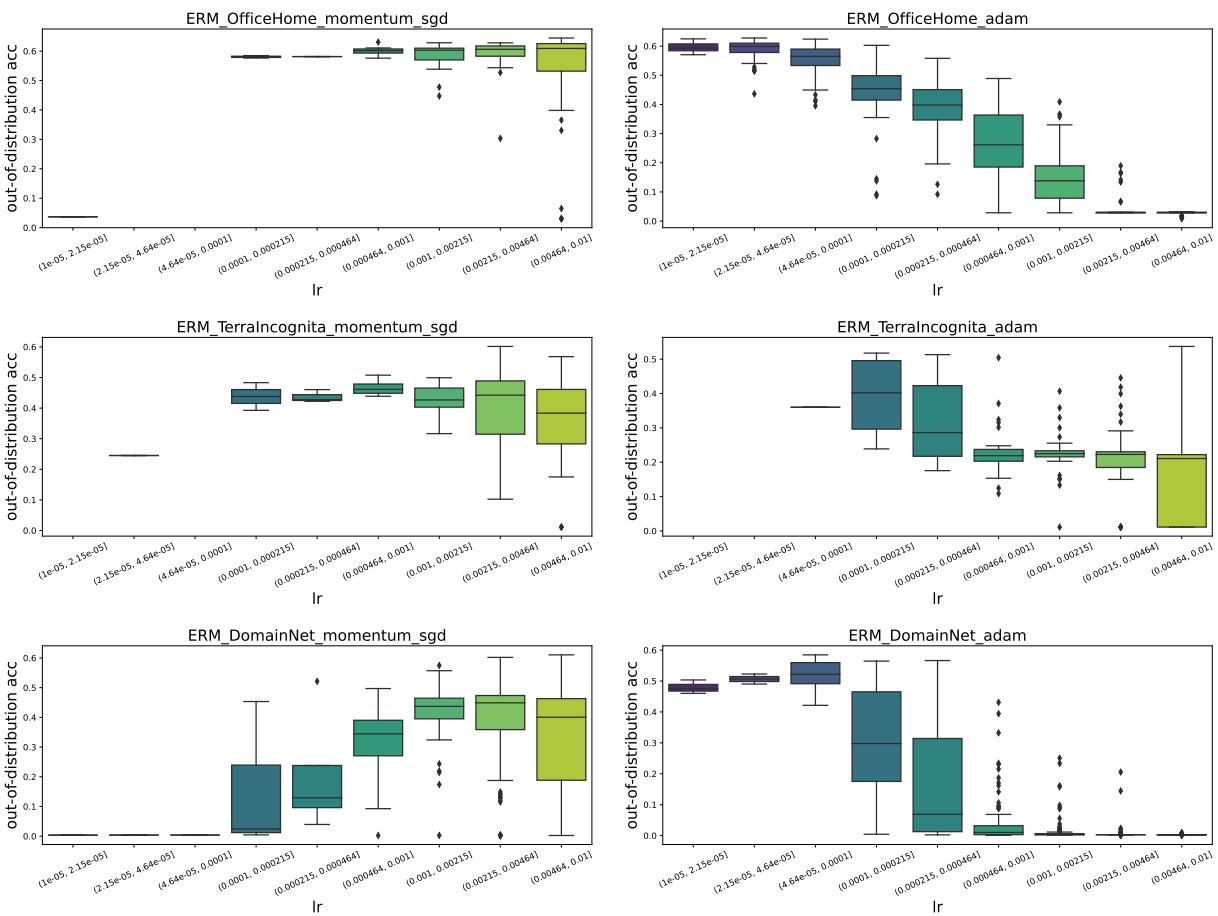

Figure 45. Box-Plot of Out-of-Distribution Accuracy per Log-Scale of Learning Rate: OfficeHome, TerraIncognita, and DomainNet

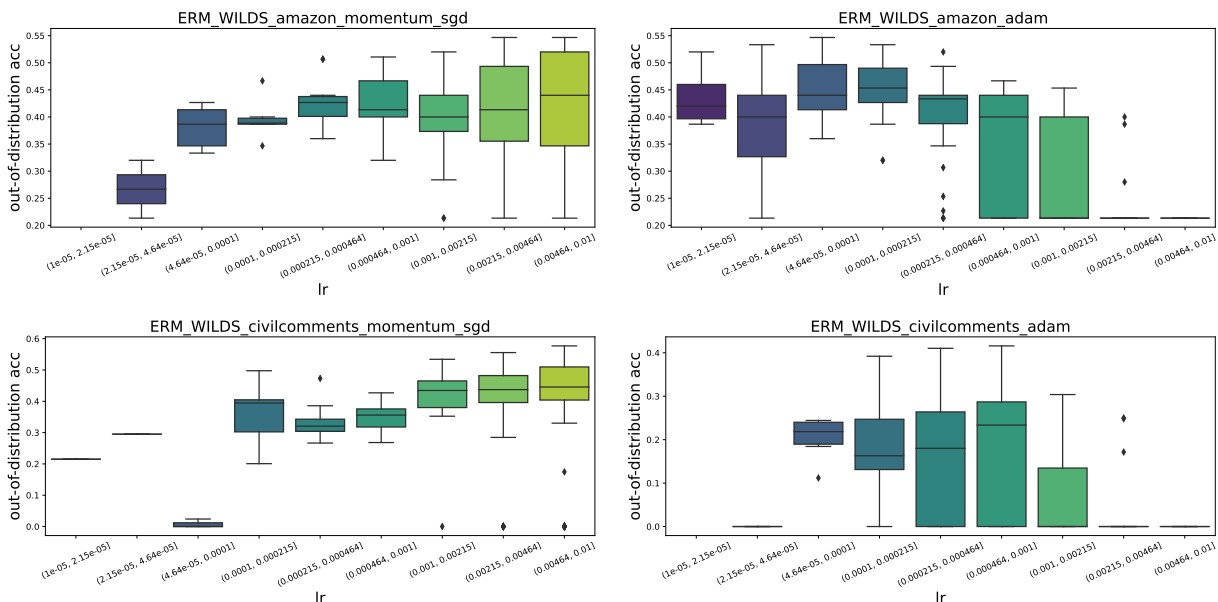

Figure 46. Box-Plot of Out-of-Distribution Accuracy per Log-Scale of Learning Rate: Amazon-WILDS, and CivilComments-WILDS

## H.3 Effect of Initial Configuration on Hyperparameter Optimization

In this section, we investigate how the first hyperparameter combination affected the search in our Bayesian optimization of hyperparameters. We compared Momentum SGD to Adam and chose PACS as our dataset. The experimental results showed that random initialization which we used, given a sufficient number of trials (e.g., more than 200 trials within our experimental protocol), did not differ from the final performance obtained when searching from the default hyperparameters of pytorch.

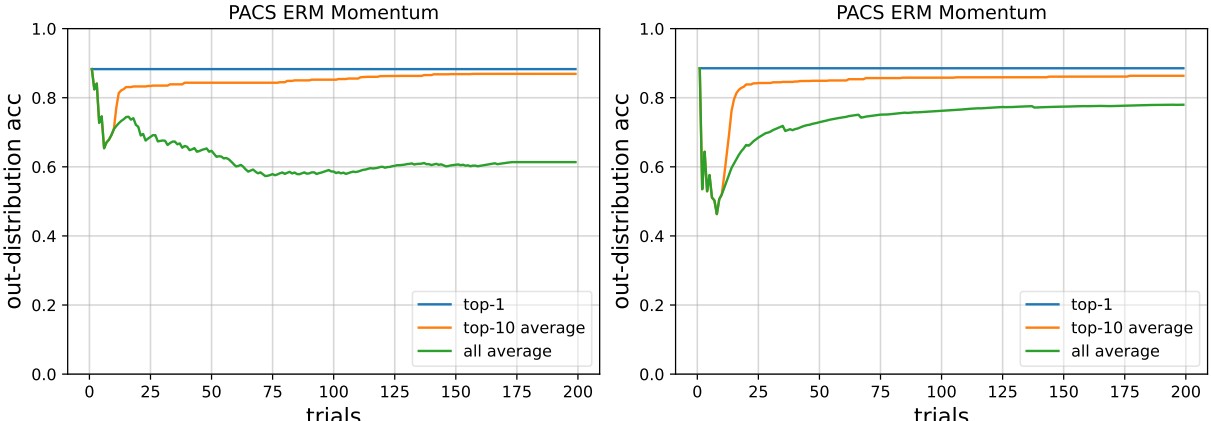

Figure 47. ERM PACS MomentumSGD / Comparison of Initialization (Left: Random initialization, Right: Pytorch default hyperparameter initialization)

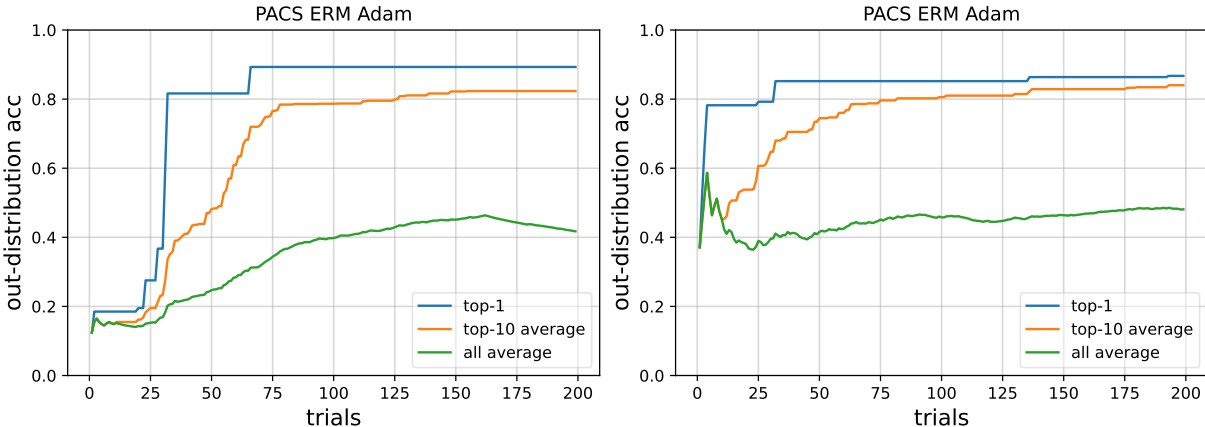

Figure 48. ERM PACS Adam / Comparison of Initialization (Left: Random initialization, Right: Pytorch default hyperparameter initialization)

### H.4 Best OOD Performance Comparison against with Existing Benckmark

In order to confirm the soundness of our experiments, we compared our results with existing benchmarks to see how well they actually performed, in addition to the hyperparameter search ranges in the previous section. In particular, we compared our experimental results with those of DomainBed (Gulrajani & Lopez-Paz, 2021), an existing oracle benchmark that uses Adam.

Table 15: OOD accuracy (%) comparison of our experimental results with the benchmark results reported in DomainBed (Gulrajani & Lopez-Paz, 2021)

| Dataset | OOD Domain | Existing Benchmark Results(Adam) | Our Results(Adam) |
|---|---|---|---|
| ColoredMNIST | 0.9 | $30.0_{\pm 0.3}$ | 73.92 |
| RotatedMNIST | 0 | $96.0_{\pm 0.2}$ | 96.40 |
| VLCS | C | $97.7_{\pm 0.3}$ | 99.36 |
| PACS | A | $87.8_{\pm 0.4}$ | 89.30 |
| OfficeHome | A | $61.2_{\pm 1.4}$ | 63.12 |
| TerraIncognita | L100 | $59.9_{\pm 1.0}$ | 61.35 |
| DomainNet | clipart | $58.4_{\pm 0.3}$ | 58.48 |

# I Additional Study

## I.1 Corruption and Perturbation Shift

In the main body of our paper, we presented experimental results for the seven types of domain shifts included in DomainBed (Gulrajani & Lopez-Paz, 2021), as well as the Background Challenge (Xiao et al., 2021), which deals with background shifts, and WILDS datasets (Koh et al., 2021) which deals with the population shift dataset. In this section, we report the results of our investigation of the corruption and perturbation datasets to investigate a broader range of out-of-distribution generalization. Details of the experiments are described in Appendix C.4, D.4 and E.4. CIFAR10-C and CIFAR10-P (Hendrycks & Dietterich, 2019) were used as the datasets. Momentum SGD and Adam were used as optimization methods for comparison.

The CIFAR10-C results are averaged performance results for 19 different noise types of corruption and are based on Hendrycks & Dietterich (2019) experimental protocol. The higher the performance, the better. The CIFAR10-P experiment shows the variability of inference for noise perturbations, with lower values indicating better performance. Experimental results show that Momentum SGD outperforms Adam in both CIFAR10-C and CIFAR10-P (Table 16).

Table 16: CIFAR-10-C (averaging corruption classification accuracy %) and CIFAR-10-P (top-5 robustness perturbation): Performance comparison between Momentum SGD and Adam with mean and standard deviation.

| Dataset | Momentum | Adam |
|---|---|---|
| CIFAR10-C ($\uparrow$) | $\mathbf{42.89_{\pm 6.66}}$ | $42.06_{\pm 6.79}$ |
| CIFAR10-P ($\downarrow$) | $\mathbf{1.38_{\pm 0.33}}$ | $1.62_{\pm 0.26}$ |

## I.2 Model Architecture

In DomainBed experiments, we followed Gulrajani & Lopez-Paz (2021) and used only ConvNet for the MNIST-based dataset and ResNet-50 for the ImageNet-based dataset. In this section, we investigate the impact of changing the model architecture.

### I.2.1 ResNet-20 for ColoredMNIST

Here are the results of ResNet-20 on the ColorMNIST Task (Figure 17). In ConvNet case, Adam outperformed Momentum SGD, but the results were reversed in ResNet-20.

Table 17: ColoredMNIST: OOD accuracy (%) comparison between Momentum SGD and Adam

| Model Architecture | Momentum | Adam |
|---|---|---|
| ConvNet | $12.86_{\pm 4.66}$ | $\mathbf{16.12_{\pm 7.98}}$ |
| ResNet-20 | $\mathbf{11.00_{\pm 0.52}}$ | $10.09_{\pm 0.15}$ |

### I.2.2 Vision Transformer for PACS

Vision Transformer (ViT)(Dosovitskiy et al., 2020) is a neural network using the recent attention structure. We evaluated the out-of-distribution performance when using Vision Transformer as well as ResNet-50 used in DomainBed. However, due to computational resource constraints, we compared Adam and Momentum SGD only for the task on the PACS dataset. The experimental results are shown in Table 18.

As a result, the experimental results with Vision Transformer show a significant performance improvement over the ResNet-50 case. Furthermore, the result that Momentum SGD outperforms Adam is consistent with the ResNet-50 case.

Table 18:  PACS: OOD accuracy(%) comparison between Momentum SGD and Adam

| Model Architecture | Momentum | Adam |
|---|---|---|
| ResNet-50 | $\mathbf{87.03_{\pm 0.65}}$ | $83.94_{\pm 0.88}$ |
| ViT | $\mathbf{90.28_{\pm 0.54}}$ | $90.05_{\pm 0.29}$ |

### I.3  State-of-the-Arts Optimizers

### I.3.1  Sharpness Aware Minimization (SAM)

Sharpness-Aware Minimization (SAM) (Foret et al., 2020) prevents convergence to high curvature local minima. Its convergence towards smaller curvature solutions results in high validation and test performance on in-distribution (ID) environment. SAM searches for points where the loss is maximized within a neighborhood of $\rho$ and uses the gradient at that point for iterative optimization. The larger the $\rho$, the higher the effect of preventing convergence to high curvature local minima.

We carried out experiments with SAM on PACS and Amazon-WILDS datasets tasks, comparing it to both Momentum SGD and Adam over a range of hyperparameters outlined in Appendices E.3 and E.3. The experimental results indicated competitive performance by SAM, equaling Momentum SGD. In the case of the Amazon-WILDS dataset, SAM proved superior for both in-distribution and out-of-distribution accuracy.

Table 19: PACS: Accuracy (%) comparison of SAM with Momentum SGD and Adam

| Accuracy | Momentum | Adam | SAM |
|---|---|---|---|
| ID accuracy | $96.79_{\pm 0.9}$ | $96.78_{\pm 0.42}$ | $\mathbf{97.54_{\pm 0.07}}$ |
| OOD accuracy | $\mathbf{87.03_{\pm 0.65}}$ | $83.94_{\pm 0.88}$ | $86.65_{\pm 0.9}$ |

Table 20: Amazon-WILDS: Accuracy (%) comparison of SAM with Momentum SGD and Adam

| Accuracy | Momentum | Adam | SAM |
|---|---|---|---|
| ID accuracy | $72.51_{\pm 0.06}$ | $72.02_{\pm 0.07}$ | $\mathbf{73.42_{\pm 0.07}}$ |
| OOD accuracy | $53.33_{\pm 0.0}$ | $52.67_{\pm 0.77}$ | $\mathbf{54.0_{\pm 0.94}}$ |

### I.3.2  Adam with Decoupled Weight Decay (AdamW)

The AdamW optimizer, proposed by Loshchilov & Hutter (2017) is an extension of the popular Adam optimization algorithm. AdamW addresses the shortcomings of the original Adam optimizer concerning weight decay regularization. In the original Adam algorithm, weight decay is directly applied to the adaptive learning rates, causing a discrepancy between the intended effect of weight decay and the actual effect in practice. The AdamW optimizer decouples weight decay from the adaptive learning rates, resulting in a more effective regularization method that is better suited for various deep learning tasks. By incorporating weight decay separately from the update step, the AdamW optimizer exhibits improved convergence properties and generalization performance compared to the original Adam algorithm.

We show the results of our AdamW experiment in Table 21. After tuning learning rate and selecting the model with the best learning rate, we ran the experiment with three different seeds and computed the mean and variance. We found that AdamW performs better than Adam, but not as well as Momentum SGD.

Table 21: Amazon-WILDS: Comparison of AdamW with Momentum SGD and Adam

| Accuracy | Momentum | Adam | AdamW |
|---|---|---|---|
| ID accuracy | $\mathbf{72.51_{\pm 0.06}}$ | $72.02_{\pm 0.07}$ | $72.27_{\pm 0.21}$ |
| OOD accuracy | $\mathbf{53.33_{\pm 0.0}}$ | $52.67_{\pm 0.77}$ | $\mathbf{53.33_{\pm 0.33}}$ |

### I.4 Large $\epsilon$ for Adam

The study in Choi et al. (2019) shows that high in-distribution performance similar to Momentum SGD can be achieved with large $\epsilon$. However, $\epsilon$ is a hyperparameter that has been introduced to prevent zero-percentage, and is specified as $\epsilon = $ 1e-8 in pytorch's default implementation[8]. It is known that as $\epsilon$ increases, Adam approximate Momentum SGD (Choi et al., 2019). While this section of our paper investigated with $\epsilon$ to the extent that it behaves as Adam, in this section we provide experimental results at large $\epsilon$, where Adam is expected to behave more like Momentum SGD.

The results of the experiment are shown in Figure 49. The x marker in the lower left corner shows the results for the default hyperparameters in Adam. The other circle markers are for different $\epsilon$ in Adam. Red stars indicate some of the Momentum results. It can be seen that the larger $\epsilon$ achieved performance closer to Momentum SGD than the default $\epsilon$ in Adam.

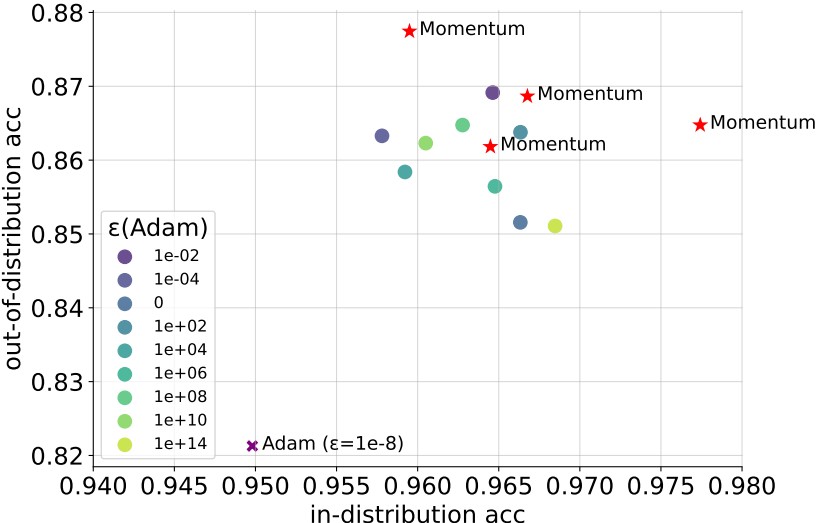

Figure 49. Demonstrating the varying performance of out-of-distribution accuracy according to the value of $\epsilon$ in Adam. As $\epsilon$ becomes larger, it approaches the performance of Momentum.

### I.5 Learning Rate Schedule

Amazon-WILDS task uses linear scheduling without warmups [9]. However, learning rate scheduling can affect performance. To consider this impact, we compared and verified the following four learning rate schedules and eight patterns with and without warmup.

Table 22 shows the results of the Amazon-WILDS experiments. However, the results for the largest out-of-distribution accuracy are shown since no significant differences were found for all experiments. No significant differences were found with or without warmup. It was not clear that introducing warmup is always effective. The Cosine LR Schedule was found to be the most effective in the problem setting of this study.

---

[8]https://pytorch.org/docs/stable/generated/torch.optim.Adam.html
[9]https://github.com/p-lambda/wilds

Table 22: Amazon-WILDS: OOD Accuracy (%) comparison of LR Scheduler and Warmup

| Learning Rate Schedule | Momentum | Adam | AdamW |
|---|---|---|---|
| Constant LR | 53.33 | 53.33 | 52.00 |
| Constant LR + Warmup | 53.33 | 52.00 | 53.33 |
| Cosine LR | **54.67** | **54.67** | **54.67** |
| Cosine LR + Warmup | **54.67** | **54.67** | **54.67** |
| Linear LR (Default) | **54.67** | 52.00 | 52.00 |
| Linear LR + Warmup | **54.67** | **54.67** | 53.33 |
| MultiStep LR | 52.00 | 53.33 | 52.00 |
| MultiStep LR + Warmup | 50.67 | 53.33 | 52.00 |

The CivilComments-WILDS task, akin to Amazon-WILDS, employs linear scheduling without warmups, as elucidated in the official repository[10]. We proceeded to examine the CivilComments-WILDS dataset to identify any analogous trends that may emerge. Our analyses from the CivilComments-WILDS dataset corroborated the findings from our Amazon-WILDS study, thereby reinforcing our preliminary conclusion that non-adaptive optimizers typically exhibit superior performance over their adaptive counterparts. Notably, we observed a significant deviation in the OOD performance when modifying the learning rate scheduler in the CivilComments-WILDS dataset. This observation starkly contrasts with our experiences in the Amazon-WILDS setting. Moreover, despite these modifications to the learning rate scheduler, we could not surpass the performance outcomes achieved with the default learning rate schedule. This reiterates the efficacy of the default setting, and further validates our overall findings.

Table 23: CivilComments-WILDS: OOD Accuracy (%) comparison of LR Scheduler and Warmup

| Learning Rate Schedule | Momentum | Adam |
|---|---|---|
| Constant LR | 47.82 | 44.29 |
| Constant LR + Warmup | 47.54 | 44.44 |
| Cosine LR | 56.98 | 45.40 |
| Cosine LR + Warmup | 56.83 | 45.48 |
| Linear LR (Default) | **57.69** | **46.82** |
| Linear LR + Warmup | 57.14 | 45.00 |
| MultiStep LR | 47.82 | 44.29 |
| MultiStep LR + Warmup | 51.35 | 44.04 |

### I.6 The Effect of Random Seeds

We conducted experiments on the effect of seed, which controls randomness, such as model initialization, on learning, using the PACS and Amazon-WILDS datasets.

In the PACS experiment shown in Table 24, a performance difference of around 6% was observed due to the effect of seed, especially for Adam. In contrast, in Momentum SGD, the effect of seed was not significant. In the Amazon-WILDS experiment shown in Table 25, seed had no significant effect on either Adam or Nesterov Momentum SGD.

Table 24: PACS: OOD Accuracy (%) Different Seed Comparison

| Seed | Momentum | Adam |
|---|---|---|
| 2021 | 86.47 | 81.20 |
| 2022 | 86.47 | 87.06 |
| 2023 | 87.74 | 85.69 |

Table 25: Amazon-WILDS: OOD Accuracy (%) Different Seed Comparison

| Seed | Nesterov | Adam |
|---|---|---|
| 2021 | 53.33 | 52.00 |
| 2022 | 53.73 | 53.33 |
| 2023 | 54.67 | 53.33 |

---

[10] https://github.com/p-lambda/wilds

### I.7   Algorithms (ERM, IRM, VREx and CORAL)

In this study, we focused on ERM and IRM and obtained consistent results that the Non-Adaptive optimizer outperforms the Adaptive optimizer in out-of-distribution performance. We also verified the use of VREx(Krueger et al., 2021) and CORAL(Sun & Saenko, 2016) as the other algorithms. The results of these experiments are shown in Table 26. However, due to limited computing resources, experiments were conducted only for Momentum SGD and Adam for PACS.

Table 26: PACS: OOD Accuracy (%) comparison of algorithms (ERM, IRM, VREx and CORAL)

| Algorithm | Momentum | Adam |
|---|---|---|
| ERM | $\mathbf{87.03}_{\pm \mathbf{0.65}}$ | $83.94_{\pm 0.88}$ |
| IRM | $\mathbf{83.06}_{\pm \mathbf{0.32}}$ | $83.05_{\pm 0.44}$ |
| VREx | $\mathbf{85.70}_{\pm \mathbf{0.24}}$ | $84.85_{\pm 1.88}$ |
| CORAL | $\mathbf{84.25}_{\pm \mathbf{1.35}}$ | $84.10_{\pm 1.13}$ |

