# OpenReview forum: "Empirical Study on Optimizer Selection for Out-of-Distribution Generalization"
_TMLR — Accepted by TMLR_

### Review · Reviewer_Fje8 · 2023-04-07

**Summary Of Contributions:**

The presented work conducts an empirical comparison of 5 different optimizers on several domain generalization benchmarks to assess whether non-adaptive or adaptive optimizers lead to better out-of-distribution generalization performance. For that, they include SGD, Momentum, and Nesterov Momentum as non-adaptive optimizers and Adam, RMSProp as adaptive variants, and tune each optimizer with 200 trials for each benchmarking problem. From their experiments and analysis, they conclude that non-adaptive optimizers perform better than adaptive optimizers for OOD performance.

**Audience:**

Yes

**Claims And Evidence:**

Yes

**Requested Changes:**

I am slightly in favour of publishing, simply because such empirical insights are quite interesting for the broader community. However, adding experiments for the mentioned weaknesses would significantly improve the work and I urge the authors to include some of them, at least in the form of control experiments on a limited set of problems.

**Strengths And Weaknesses:**

### Strengths:

* Good addition to [7] for empirical insights into optimizer performance with special focus on OOD generalization.
* Tuning runs are shared as box-plots, showing how sensitive some of the methods are to tuning.
* Interesting take-away that non-adaptive optimizers are superior in terms of OOD generalisation compared to adaptive methods.
* Good experiments in H.2 to verify that, in the limit, random initialization for hyper parameter search does not differ from initialization with the PyTorch defaults.

### Author-aware Weaknesses:

Here I want to list a few weaknesses that the authors are already aware of but I believe they are still weaknesses that should be addressed to improve the presented work:

* [**Large**]: While I do agree that selecting optimizers to benchmark is a difficult procedure since different readers might have varying expectations given the avalanche of methods (cf. Table 2 in [7]), the fact that no published optimizer after 2015 has been included doesn’t quite sit right with me. For NLP training (a focus of this work according to p.2), the community commonly sticks to an AdamW [1] variant (see e.g. [2], [3], [4]) while in the broader optimization community, sharpness-aware minimization (SAM) [5] has also attracted a lot of recent interest for improved generalization performance. Other works in this field (e.g. [7]), do at least include some more recently proposed optimizers in their studies, albeit the focus on OOD generalization here is slightly different.
* [**Large**]: Building on top of the previous point, the presented work focuses only on first-order methods. Given the recent advances in second-order methods like Distributed Shampoo (see e.g. [6]), it would be interesting for the community to assess OOD generalization for this class of optimization algorithms as well.

### Additional Weaknesses:

* [**Large**]: Given the amount of randomness that comes with benchmarking it would be good to test for the impact of different seeds. For that, we can have a study on one or two of the datasets e.g. WILDSAmazon / WILDSCivilComment where the difference between Adam <> Nesterov is particularly large and re-run the tuning / evaluation on a few more random seeds to verify how large the impact is.
* [**Large**]: Even though H.1 shows the histogram of learning rate, it is hard to verify from these if the tuning range is chosen to be sensible since the OOD generalization performance is not plotted against the corresponding hyperparameter value. Instead of showing a histogram, a box plot for the corresponding OOD performance would be much more useful. Assuming that the bayesian optimization works as expected, there are quite a few Adam learning rates that are sampled frequently towards the lower end of the tuning range (e.g. RotatedMNIST, VLCS, OfficeHome, WILDSAmazon) suggesting that the optimal learning rate falls outside of the search space. I'm expecting similar problems with some of the other hyperparameters that are not shown here.
* [**Minor**]: As discussed, with large $\epsilon$ Adam turns into a non-adaptive method and, as found by Choi et al. (2019), never performs worse than Momentum for in-distribution generalization. In the presented work, the tuning range for $\epsilon$ is too small $[1e-8, 1e-3]$ (compared to Choi et al.) and does not consider this limit of large $\epsilon$ Adam. It would be interesting to see if  large $\epsilon$ Adam also never performs worse than Momentum for OOD generalization.
* [**Minor**]: Only one type of learning rate schedule (linear decay) is considered, albeit with sampling when to decay from and the decay amount. However, there are many more schedules (cf. Table 3 in [7]) where especially warmup + step number decay is popular in NLP. I'm assuming that this will have an impact on the results.

### Minor Comments & Typos:

* Presentation of Figure 1 can be improved, the colour scheme is shared so the legend can be displayed once below all the subfigures. Also, what does the red color indicate? Spacing between top and bottom row should be increased to make it look less cramped.
* Figure 2 Caption: OficeHome → OfficeHome, I suggest creating a macro to avoid typos and unify notation e.g. *\newcommand{\officehome}{\textsc{OfficeHome}\@\xspace}*
* The x-axis label spacing for some of the subfigures can be improved (e.g. Figure 2 or 4). The label for “Momentum” and “Nesterov” basically overlap.
* Liu et al. and [7] have two entries in the bibliography, please consolidate and use the published version.
* I believe the Appendix should be condensed, even though I appreciate the amount of data that is incorporated, I believe some of the plots are redundant and/or could be visualised better to condense them into a few key figures, might be worth considering plots similar to Figure 4 in [7] with shaded areas for tuning runs or another visualization style.
* Figure 37 top row can probably be scaled better, there is nothing to see.
* Eq. 15 is formatted weirdly, there is a lot of space to the main text

---

* [1]: [Decoupled Weight Decay Regularization](https://openreview.net/forum?id=Bkg6RiCqY7) , Loshchilov & Hutter, ICLR 2019
* [2]: [AdamW is default in fairseq as Adam](https://fairseq.readthedocs.io/en/latest/optim.html?highlight=AdamW#fairseq.optim.adam.FairseqAdam) so most papers that use fairseq are using it.
* [3]: [No Language Left Behind: Scaling Human-Centered Machine Translation](https://arxiv.org/abs/2207.04672), NLLB-Team, 2022
* [4]: [LLaMA: Open and Efficient Foundation Language Models](https://arxiv.org/abs/2302.13971), Touvron et al., 2023
* [5]: [Sharpness-aware Minimization for Efficiently Improving Generalization](https://iclr.cc/virtual/2021/spotlight/3497), Foret et al., ICLR 2021
* [6]: [Scalable Second Order Optimization for Deep Learning](https://arxiv.org/abs/2002.09018), Anil et al., 2020
* [7]: [Descending through a Crowded Valley -- Benchmarking Deep Learning Optimizers](https://proceedings.mlr.press/v139/schmidt21a.html), Schmidt et al., ICML 2021

---

### Review · Reviewer_JSuw · 2023-04-19

**Summary Of Contributions:**

This paper presents a comprehensive study on optimizer selection for OOD generalization. The authors compare the IID and OOD performances of various models on multiple datasets from DomainBed, Wilds benchmark, and Backgrounds Challenge, when optimized with both adaptive optimizers (e.g., Adam) and non-adaptive optimizers (e.g., Momentum SGD). The empirical results suggest that, under the same hyperparameter tunning scope and evaluation protocol, the choice of optimizers can have a non-trivial influence on OOD performance, and non-adaptive optimizers achieve better OOD performance than adaptive optimizers in most of the datasets.

**Audience:**

Yes

**Claims And Evidence:**

Yes

**Requested Changes:**

Although I appreciate the great efforts in the empirical comparison, I find the work can still be improved as follows (corresponding to each weakness):

i) One major conclusion from DomainBed is that, the model selection can also have a large influence to the OOD performance. Throughout the paper, all of the models are selected according to the performance in a validation set that has the same distribution as the training set, which may significantly weaken the OOD performance of the selected models, and fail to uncover the full empirical trends. In fact, in practice, an OOD validation set is often available for model selection. Hence, I suggest the authors also present more results that use different validation sets as suggested by DomainBed, to examine whether the same empirical trends can be observed as using the current validation set.

Besides, the model architecture can also have a influence on the optimization. For example, in ColoredMNIST, DomainBed uses a CNN network, which may be more suitable for Adam to coverge and even overfit. It would be further strengthen the reliability of the claims and to differentiate the influence of model architectures to the OOD performance, if the authors would provide additional results using other model architectures for some datasets such as ColoredMNIST.

ii) The experiments regarding the optimization with OOD objectives only consider IRMv1, while there is a rich literature on the OOD objectives. Although the authors find IRMv1 exhibits similar empirical behaviors as ERM, it is unknown whether other OOD objectives have a similar behavior. Since practitioners can use various OOD objectives to train the model, more discussions or empirical results are expected about the influence of OOD objectives on the discoveries of this work.

iii) The related work discussion misses some closely relevant works [1,2,3,4]. [1,2,4] discussed the trade-offs of IID and OOD performance when finetuning pre-trained models. [3] uncovered the optimization dilemma in OOD generalization and the trade-offs in the optimization of the ERM and OOD objectives. [3] proposed to consider both ERM and OOD losses in optimization and model selection, and also conducted experiments on a wide variety and scale of datasets.

iv) From the discussion on Sec. 4.3, i.e., different correlation behaviors between IID and OOD performance, it is unclear how different correlation behaviors can help practitioners better understand and select optimizers. I am also confused about the description of the rightmost part of Figure 5, that “the increase in a large region of the in-distribution saturates the OOD generalization with a small percentage of its e!ect.”, for which I cannot see from the figure. Besides, it seems the key takeaway of the paper is that practitioners should use non-adaptive optimizers. The authors could make the discussion of Figure 5 more informative.


**References**

[1] Kumar et al., Fine-Tuning can Distort Pretrained Features and Underperform Out-of-Distribution, ICLR 2022.

[2] Wortsman et al., Robust ﬁne-tuning of zero-shot models, CVPR 2022.

[3] Chen et al., Pareto Invariant Risk Minimization: Towards Mitigating The Optimization Dilemma in Out-of-Distribution Generalization, ICLR 2023.

[4] Kumar et al., How to Fine-Tune Vision Models with SGD, arXiv 2022.


**Strengths And Weaknesses:**

**Strengths**

i) This work presents a comprehensive empirical study. The authors make great efforts to carefully control the influence of other factors in order to elucidate the inﬂuence of optimizer selection under distributional shifts.

ii) The experiments cover a broad scope of datasets, neural network architectures, and optimizers.

iii) The authors provide plentiful analysis and discussion on the empirical results such as different IID and OOD performance trends.

iv) The empirical trends are aligned with the existing literature.

**Weakness**

i) The evaluation setting may not consider all factors that can have a major influence on the OOD performance, such as model selection criteria.

ii) The experiments regarding the optimization with OOD objectives may not be adequate since only IRMv1 is considered.

iii) Missing discussion with some related works.

iv) It’s unclear how the categorization of IID and OOD performance can help practitioners better understand and select optimizers.

---

### Review · Reviewer_VQ4e · 2023-05-01

**Summary Of Contributions:**

The authors benchmark different optimizers (3 non adaptive optimizers: SGD, Nesterov accelerated gradient, and SGD+Momentum, and 2 adaptive optimizers: Adam and RMSProp) on image classification and NLP datasets, with the goal of understanding which optimizer(s) perform best when evaluated on downstream performance under dataset shift. The authors benchmark CNNs, ResNet 50s, and DistilBERT.

Compared to previous benchmarks of this type, the key differences are the author's type of evaluation metric (performance under dataset shift) and the increased budget allocated to tuning the hyperparameters of each method.

Based on their analysis, the authors draw the following conclusions:
1. Overall, non-adaptive optimizers produce models that are more robust to dataset shift.
2. That the relationship between in-distribution (IND) and out-of-distribution (OOD) performance is either sub-linear, super-linear, or essentially linear, reproducing + expanding upon previous known results (cited in the paper).

**Audience:**

Yes

**Claims And Evidence:**

No

**Requested Changes:**

Critical to acceptance is
- Clarifying the experimental setup (# of experimental replicas, whether results are computed over optimal hyperparameters or over the entire set of evaluated hyperparameters), adding standard deviations to the tables.

Strongly suggested, but not strictly necessary for acceptance:
- Including other datasets commonly used to evaluate robustness under dataset shift. This should be possibly doable for evaluation on the Imagenet corrupted versions described above, assuming the checkpointed models on Imagenet-1K from the Backgrounds challenge are available.


**Strengths And Weaknesses:**

# Strengths
- Well-tuned benchmarks: As the authors point out, their hyperparameter tuning subsumes prior work, in turn leading to more competitive baselines.
- Well-motivated idea: although the importance of benchmarking ML models on dataset shifts is well-recognized in the field, this is (to the extent of my knowledge) the first benchmark of optimizers within this context, providing a valuable contribution.
- The analysis of Colored MNIST (including in the appendix) was really interesting! In general, most of the experimental analysis is very detailed.
- Clarity: Overall, the paper is well written.

# Weaknesses
To me, the major weakness of this work is in the set of benchmarks and in how the results are presented (as such, any author response that addresses or rebuts this weakness would increase my score). My chief concerns are the following:
- Benchmark suite: common benchmarks for dataset shifts include several imagenet variants (Imagenet-C, Imagenet-A, Imagenet-R, Imagenet-V2, etc.), Cifar-10 shifts (Hendrycks & Dietterich, ICLR 2019), or SVHN after training on Cifar-10 (see, e.g., Ovadia et al, NeurIPS 2019). Some of the benchmarks considered in this work seem weak given the type of models that are being benchmarked (e.g., rotated MNIST). I strongly recommend authors consider additional datasets to facilitate comparisons with existing literature on dataset shift benchmarking. However, I do realize that many of this datasets can be expensive to validate against; if others more familiar with optimizer benchmarking support the current set, I will take this into consideration.

- Reported results: some important experimental details were not easily obvious to me. In particular, how many different runs did you use (with the best hyperparameters) for the evaluation? (The text just states "the evaluation of several trained models", and no standard deviations are provided in Table 1.)
- Similarly, what distribution of data was used for the histograms in Figure 1? Are the points from all training runs (including other hyperparameters? or simply from different model initializations with the best hyperparameters?
- I found the barplots of Figure 1 somewhat hard to read; is there a reason not to use a scatterplot?

- Finally, although I understand that this is possibly due to compute limitations, including Transformer-based architectures would strengthen the benchmark and the conclusions that are drawn.

---

### Comment · Action_Editors · 2023-05-11
**Discussion phase - Authors and reviewers**

Dear Reviewers and Authors,

now that we have three reviews in, the discussion phase starts. Authors please check the three reviews, and answer each reviews by commenting below their review. At this time you also have the opportunity to update your draft so as to address the reviews concerns.

Regards, AC

---

### Decision · Action_Editors · 2023-06-04

**Recommendation:** Accept as is

**Comment:**

The paper provides an extensive benchmark of our different optimization methods (adaptive and non-adaptive) performs with respect to Out-of-Distribution Generalization (OOD). The experiments were already extensive, and after requests by all reviewers, the authors have included yet more experimental setups and datasets, making this a useful benchmark for OOD results.

**Audience:**

Both optimisation ML community and experimental DNN could be interested in the benchmarks presented here.

**Claims And Evidence:**

The authors provide and extensive set of experiments to back their claims. Parameter sweeps and settings are also clear.